# A UNIFIED OPTIMIZATION FRAMEWORK OF ANN-SNN CONVERSION: TOWARDS OPTIMAL MAPPING FROM ACTIVATION VALUES TO FIRING RATES

## ABSTRACT

Spiking Neural Networks (SNNs) have attracted great attention as a primary candidate for running large-scale deep artificial neural networks (ANNs) in real-time due to their distinctive properties of energy-efficient and event-driven fast computation. Training an SNN directly from scratch is usually difficult because of the discreteness of spikes. Converting an ANN to an SNN, i.e., ANN-SNN conversion, is an alternative method to obtain deep SNNs. The performance of the converted SNN is determined by both the ANN performance and the conversion error. The existing ANN-SNN conversion methods usually redesign the ANN with a new activation function instead of the regular ReLU, train the tailored ANN and convert it to an SNN. The performance loss between the regular ANN with ReLU and the tailored ANN has never been considered, which will be inherited to the converted SNN. In this work, we formulate the ANN-SNN conversion as a unified optimization problem which considers the performance loss between the regular ANN and the tailored ANN, as well as the conversion error simultaneously. Following the unified optimization framework, we propose the SlipReLU activation function to replace the regular ReLU activation function in the tailored ANN. The SlipReLU is a weighted sum of the threshold-ReLU and the step function, which improves the performance of either as an activation function alone. The SlipReLU method covers a family of activation functions mapping from activation values in source ANNs to firing rates in target SNNs; most of the state-of-the-art optimal ANN-SNN conversion methods are special cases of our proposed SlipReLU method. We demonstrate through two theorems that the expected conversion error between SNNs and ANNs can theoretically be zero on a range of shift values $\delta \in [-\frac{1}{2}, \frac{1}{2}]$ rather than a fixed shift term $\frac{1}{2}$, enabling us to achieve converted SNNs with high accuracy and ultra-low latency. We evaluate our proposed SlipReLU method on CIFAR-10/100 and Tiny-ImageNet datasets, and the results show that the SlipReLU outperforms the state-of-the-art ANN-SNN conversion methods and directly trained SNNs in both accuracy and latency. To our knowledge, this is the first work to explore high-performance ANN-SNN conversion method considering the ANN performance and the conversion error simultaneously, with ultra-low latency, especially for 1 time-step ($T = 1$).

## 1 INTRODUCTION

Spiking neural networks (SNNs) are biologically-inspired neural networks based on biological plausible spiking neuron models to process real-time signals (Hodgkin & Huxley, 1952; Izhikevich, 2003). With the significant advantages of low power consumption and fast inference on neuromorphic hardware (Roy et al., 2019), SNNs are therefore becoming a primary candidate to run large-scale deep artificial neural networks (ANNs) in real-time. The most commonly used neuron model in SNNs is the Integrate-and-Fire (IF) neuron model (Liu & Wang, 2001). Each neuron in the SNNs emits a spike only when its accumulated membrane potential exceeds the threshold voltage, otherwise, it stays inactive in the current time-step. This setting makes SNNs more similar to biological neural networks. Compared to ANNs, event-driven SNNs have binarized/spiking activation values, resulting in low energy consumption when implemented on specialized neuromorphic hardware. Another significant property of SNNs is the pseudo-simultaneity of their inputs and outputs

for making inferences in a spatial-temporal paradigm. Compared to conventional ANNs that present a whole input vector at once, and process layer-by-layer to produce one output value, the forwarding pass in SNN can efficiently process streaming time-varying inputs.

Generally, there are two distinct routes to obtain an SNN: (1) training an SNN from scratch (Wu et al., 2018; Neftci et al., 2019; Zenke & Vogels, 2021), and (2) ANN-SNN conversion (Cao et al., 2015; Diehl et al., 2015; Deng & Gu, 2021), i.e., converting ANNs to SNNs. Training from scratch uses a gradient-based supervised optimization method in back-propagation, pretending that SNNs are specialized ANNs. Due to the non-differentiability of the binary activation function in SNNs, surrogate gradients are usually used (Neftci et al., 2019), but it essentially optimizes different networks in forward and backward passes. This method can only train SNNs on small- and moderate-size datasets (Li et al., 2021). ANN-SNN conversion is an effective method to obtain deep SNNs, with comparable performance as ANNs on large-scale datasets. There are two main types of ANN-SNN conversion mechanism: (1) one-step conversion, which converts the pre-trained ANN to SNN without changing the architecture of the pre-trained ANN, for example Diehl et al. (2015); Li et al. (2021), and (2) two-step conversion, which involves redesigning the ANN, training it and converting it to SNN, for example Cao et al. (2015); Deng & Gu (2021); Bu et al. (2021).

In this work, we investigate the two-step ANN-SNN conversion methods, where we usually redesign the ANN by replacing the regular ReLU activation function to a new activation function, train the tailored ANN and convert it to an SNN. A tailored ANN that deviates too much from the regular ANN will degrade its performance, resulting in a performance loss which will be inherited to the converted SNN. However, the performance degradation between the regular ANN and the tailored ANN has never been considered in the existing ANN-SNN conversion studies. To achieve high-accuracy and low-latency SNNs (e.g., 1 or 2 time-steps), we are the first to consider the performance loss between the regular ANN with ReLU and the tailored ANN, as well as the conversion error simultaneously. Our main contributions are summarized as follows:
(1) We formulate the ANN-SNN conversion as a unified optimization problem which considers the ANN performance as well as the conversion error simultaneously.
(2) We propose to use the SlipReLU activation function in the tailored ANN, in order to minimize the layer-wise conversion error and keep tailored ANN performance as good as the regular ANN.
(3) The SlipReLU method covers a family of activation functions mapping from activation values in source ANNs to firing rates in target SNNs; most of the state-of-the-art optimal ANN-SNN conversion methods are special cases of our proposed SlipReLU method.
(4) We demonstrate through two theorems that the expected conversion error between SNNs and ANNs can theoretically be zero on a range of shift values $\delta \in [-\frac{1}{2}, \frac{1}{2}]$ rather than a fixed shift $\frac{1}{2}$. Experiment results also demonstrate the effectiveness of the proposed SlipReLU method.

## 2 PRELIMINARIES

Given a classification problem on an image dataset $(\boldsymbol{x}, y) \in \mathcal{D}$, where $y \in \{1, \cdots, C\}$ is the true class label for image $\boldsymbol{x} \in \mathbb{R}^m$, we train a neural network $f : \boldsymbol{x} \rightarrow f(\boldsymbol{x})$ in the form of an ANN/SNN, by optimizing the standard cross-entropy (CE) loss, $L_{\mathrm{CE}}(\boldsymbol{y}, \boldsymbol{p}) = -\sum_{j=1}^C \boldsymbol{y}_c \log(\boldsymbol{p}_c)$, where $\boldsymbol{y}_c$ and $\boldsymbol{p}_c$ are the $c$-th elements of the label $\boldsymbol{y}$ and the network prediction $\boldsymbol{p} = f(\boldsymbol{x})$. Since the infrastructures of the source ANN and target SNN are the same, we use the same $f$ notation when it is unambiguous. And $f_{\mathrm{ANN}}$ or $f_{\mathrm{SNN}}$, otherwise. For the notations, refer to Table S1.

**ANN Neuron Model.** In conventional ANN, a whole input vector is presented to the network at one time, and processed layer-by-layer through continuous activation to produce one output value. In ANNs, the forwarding computation of analog neurons is formulated as

$$\boldsymbol{a}^{(\ell)} = \mathcal{F}_{\mathrm{ANN}}(\mathbf{z}^{(\ell)}) = \mathcal{F}_{\mathrm{ANN}}(\mathbf{W}^{(\ell)} \boldsymbol{a}^{(\ell-1)}), \tag{1}$$

where $\mathbf{z}^{(\ell)}$ and $\boldsymbol{a}^{(\ell)}$ are the pre-activation and post-activation vectors of the $\ell$-th layer considered, $\mathbf{W}^{(\ell)}$ denotes the weight matrix, and $\mathcal{F}_{\mathrm{ANN}}(\cdot)$ is the activation function of the ANN.

**SNN Neuron Model.** Compared with ANN, SNN employs binary activation (i.e. spikes) in each layer. To compensate the weak representation capacity of the binary activation, the time dimension (or the latency) is introduced to SNN, where the inputs of the forwarding pass in SNN are presented as streams of events, by repeating the forwarding pass $T$ time-steps to get the final result.

Here we consider the Integrate-and-Fire (IF) neuron model (Cao et al., 2015; Bu et al., 2021; Deng & Gu, 2021) for SNNs. We derive the forward propagation of PSP through layers in the target SNN which is equivalent to the forwarding computation of the analog neurons in the source ANN. Suppose at time-step $t$ the IF neuron in $\ell$-th layer receive its binary input $\mathbf{x}^{(\ell-1)}(t)$ from the previous layer, the IF neuron will temporarily update its membrane potential by

$$\mathbf{u}^{(\ell)}(t) = \mathbf{v}^{(\ell)}(t-1) + \mathbf{W}^{(\ell)}\mathbf{x}^{(\ell-1)}(t) , \tag{2}$$

where $\mathbf{v}^{(\ell)}(t)$ denotes the membrane potential at time step $t$, $\mathbf{u}^{(\ell)}(t)$ denotes the temporary intermediate variable that would be used to determine the update from $\mathbf{v}^{(\ell)}(t-1)$ to $\mathbf{v}^{(\ell)}(t)$. For each IF neuron, if the temporary intermediate potential $u_i^{(\ell)}(t)$ exceeds the membrane threshold $V_{\text{th}}^{(\ell)}$, it would produce a spike output $s_i^{(\ell)}(t)$. Otherwise, it would release no spikes $s_i^{(\ell)}(t) = 0$.

$$s_i^{(\ell)}(t) = H(u_i^{(\ell)}(t) - V_{\text{th}}^{(\ell)}) = \begin{cases} 1, & \text{if } u_i^{(\ell)}(t) \geqslant V_{\text{th}}^{(\ell)}, \\ 0, & \text{otherwise.} \end{cases} \tag{3}$$

The vector $\mathbf{s}^{(\ell)}(t) = \{s_i^{(\ell)}(t)\}$ collects spikes of all neurons of $\ell$-th layer at time $t$. Note that $V_{\text{th}}^{(\ell)}$ can be different in each layer. We update the membrane potential by the reset-by-subtraction mechanism (Rueckauer et al., 2017; Han et al., 2020), which means the temporary membrane potential $u_i^{(\ell)}(t)$ is subtracted by the threshold value $V_{\text{th}}^{(\ell)}$ if the neuron fires, $s_i^{(\ell)}(t) = 1$. That is

$$\mathbf{v}^{(\ell)}(t) = \mathbf{u}^{(\ell)}(t) - \mathbf{s}^{(\ell)}(t)V_{\text{th}}^{(\ell)} . \tag{4}$$

Similar to Deng & Gu (2021); Bu et al. (2021), if the neuron in the current $\ell$ layer fires a spike, then it will release an unweighted PSP (postsynaptic potential) $\mathbf{x}^{(\ell)}(t)$ as input to the next layer,

$$\mathbf{x}^{(\ell)}(t) = \mathbf{s}^{(\ell)}(t)V_{\text{th}}^{(\ell)} .$$

As for the input to the first layer and the output of the last layer of the SNN, we do not employ any spiking mechanism as in Li et al. (2021). We directly encode the static image to temporal dynamic spikes as input to the first layer, which can prevent the undesired information loss introduced by the Poisson encoding. For the last layer output, we only integrate the pre-synaptic input and do not fire any spikes.

# 3 UNIFIED OPTIMIZATION FRAMEWORK OF ANN-SNN CONVERSION

In this section, we formulate the ANN-SNN conversion problem as an optimization problem determined by two terms: one is used to make the tailored ANN not far away from the regular ANN with ReLU, the other one is used to control the ANN-SNN conversion error.

Starting from the first ANN-SNN conversion work (Cao et al., 2015), all the previous ANN-SNN conversion methods (Cao et al., 2015; Diehl et al., 2015; Deng & Gu, 2021) have focused on optimizing the ANN-SNN conversion error. The performance of the converted SNN is determined by both the ANN performance and the conversion error. However, no research work has considered the ANN performance in the two-step ANN-SNN conversions, which is considered in our unified optimization framework. In the two-step conversion method, we need to redesign the new activation function of the regular ANN to get a tailored ANN, train the tailored ANN and convert it to SNN. By considering the performance loss between the tailored ANN and the regular ANN, the new activation function should not deviate too much from the regular ReLU.

## 3.1 ANN-SNN CONVERSION IN A UNIFIED OPTIMIZATION FRAMEWORK

**Definition 1** (Unified Optimization Framework of ANN-SNN Conversion). *The ANN-SNN conversion can be formulated into a unified optimization framework with an implicit variable, $T$,*

$$\min_{\mathcal{F},T} \; \{w\mathbb{E}_{\mathbf{z}}\left(|f(\mathbf{z}; \mathbf{W}, \text{ReLU}) - f(\mathbf{z}; \mathbf{W}, T, \mathcal{F}_{\text{ANN}})|\right) \tag{5}$$

$$+ (1-w)\mathbb{E}_{\mathbf{z}}\left(|f(\mathbf{z}; \mathbf{W}, \mathcal{F}_{\text{ANN}}) - f(\mathbf{z}; \mathbf{W}, T, \mathcal{F}_{\text{SNN}})|\right)\} .$$

*where $w \in [0, 1]$. Specially, if $\mathcal{F}_{\text{ANN}}$ is designed by considering the deviation from the regular ReLU, the layer-wise conversion error becomes*

$$\mathbb{E}_{\mathbf{z}}\left(\left|\text{Err}^{(\ell)}\right|\right) = \mathbb{E}_{\mathbf{z}}\left(\left|\mathcal{F}_{\text{ANN}}(\boldsymbol{a}^{(\ell-1)}; \mathbf{W}^{(\ell)}) - \mathcal{F}_{\text{SNN}}(\bar{\mathbf{x}}^{(\ell-1)}; \mathbf{W}^{(\ell)}, T)\right|\right) . \tag{6}$$

Here $f$ denotes the same neural network infrastructures shared by the source ANN and target SNN (see description in Sect. 2), $f(\mathbf{z}; \mathbf{W}, \mathrm{ReLU})$ denotes the regular ANN with ReLU, $f(\mathbf{z}; \mathbf{W}, T, \mathcal{F}_{\mathrm{ANN}})$ is the tailored ANN with activation function $\mathcal{F}_{\mathrm{ANN}}$, $f(\mathbf{z}; \mathbf{W}, T, \mathcal{F}_{\mathrm{SNN}})$ is the converted SNN, $\mathbf{z}$ is the input to the neural network, $\mathbf{W} = \{\mathbf{W}^{(\ell)}\}$ are the weight matrix trained from the tailored ANN and copied to the target SNN, $\mathcal{F} = \mathcal{F}_{\mathrm{ANN}} \cup \mathcal{F}_{\mathrm{SNN}}$ is the space of activation functions of the tailored ANNs and the target SNNs, and the latency $T$ (or time-steps) is seen as an implicit variable inherently inherited from the target SNNs. Moreover, the latency also allows the flexibility of adjusting $T$ to balance between the latency and the accuracy of the converted SNN for different applications.

**Remark 1.** *(A) An effective activation function $\mathcal{F}_{\mathrm{ANN}}$ of the tailored ANN should address the performance lose caused by the deviation from the regular ReLU. (B) When $\mathcal{F}_{\mathrm{ANN}}$ is designed by considering the deviation from the regular ReLU, the layer-wise error Eq. (6) would come from any mismatch of the three parts: (1) different activation values from source ANNs and target SNNs, i.e. $\boldsymbol{a}^{(\ell)}$ and $\bar{\mathbf{x}}^{(\ell)}$, (2) different activation functions, i.e. $\mathcal{F}_{\mathrm{ANN}}(\cdot)$ and $\mathcal{F}_{\mathrm{SNN}}(\cdot)$, and (3) the latency variable $T$ which implicitly affects both the activation values and activation functions. (C) Whenever the conversion error $\mathbb{E}_{\mathbf{z}}(|\mathrm{Err}^{(\ell)}|)$ achieves its minimum, it is called an "optimal" ANN-SNN conversion. For example, Deng & Gu (2021) achieves optimal minimum error of $\frac{(V_{\mathrm{th}}^{(\ell)})^2}{4T}$, whereas Bu et al. (2021) can theoretically achieve optimal minimum error of $0$.*

## 3.2 ANN-SNN CONVERSION ERROR ANALYSIS

### 3.2.1 FIRING RATES IN SNNs AND ACTIVATION VALUES IN ANNs

To make the layer-wise error as small as possible, ideally, the converted SNN is expected to have approximately the same activation values as the source ANN for each layer, i.e.,

$$\boldsymbol{a}^{(\ell)} \approx \bar{\mathbf{x}}^{(\ell)} = \frac{1}{T} \sum_{t=1}^{T} \mathbf{x}^{(\ell)}(t) = \frac{1}{T} \sum_{t=1}^{T} \mathbf{s}^{(\ell)}(t) V_{\mathrm{th}}^{(\ell)} = V_{\mathrm{th}}^{(\ell)} \bar{\mathbf{s}}^{(\ell)} . \tag{7}$$

Here $\boldsymbol{a}^{(\ell)}$ denotes activation value of the ANN, and $\bar{\mathbf{x}}^{(\ell)}$ is activation value of the SNN which is actually the average postsynaptic potential (i.e. average PSP) released by the $\ell$-th layer as input to the next layer. Note $\bar{\mathbf{s}}^{(\ell)}$ is the firing rate over latency $T$ of $\ell$-th layer. Note that the thresholding $V_{\mathrm{th}}^{(\ell)}$ in SNN can be different from layer to layer, we make it a trainable parameter that can be learned in the source ANN and copied to the target SNN. Any mismatch between the activation values $\boldsymbol{a}^{(\ell)}$ and $\bar{\mathbf{x}}^{(\ell)}$ can lead to conversion error.

### 3.2.2 ACTIVATION FUNCTION IN SNNs

We use the derivation in Deng & Gu (2021); Li et al. (2021) to deduce the SNN activation function $\mathcal{F}_{\mathrm{SNN}}$ which gives the relationship between activation values $\bar{\mathbf{x}}^{(\ell-1)}$ and $\bar{\mathbf{x}}^{(\ell)}$ of successive layers of SNN. By combining Eq. (2) and Eq. (4) and summing up the time-step from 1 to $T$, then we get

$$\mathbf{v}^{(\ell)}(T) - \mathbf{v}^{(\ell)}(0) = \mathbf{W}^{(\ell)} \sum_{t=1}^{T} \mathbf{x}^{(\ell-1)}(t) - \sum_{t=1}^{T} \mathbf{s}^{(\ell)}(t) V_{\mathrm{th}}^{(\ell)} .$$

The accumulated spikes are $\mathbf{m} = \sum_{t=1}^{T} \mathbf{s}^{(\ell)}(t) = \{m_i\}$ where each $m_i \in \{0, 1, 2, \cdots, T\}$ denotes the total number of spikes of neuron $i$. Further assume $\mathbf{v}^{(\ell)}(T) \in [\mathbf{0}, \mathbf{V}_{\mathrm{th}}^{(\ell)})$. Therefore, we have

$$\frac{T\mathbf{W}^{(\ell)}\bar{\mathbf{x}}^{(\ell-1)} - \mathbf{V}_{\mathrm{th}}^{(\ell)}}{V_{\mathrm{th}}^{(\ell)}} + \boldsymbol{\delta} < \mathbf{m} \leqslant \frac{T\mathbf{W}^{(\ell)}\bar{\mathbf{x}}^{(\ell-1)}}{V_{\mathrm{th}}^{(\ell)}} + \boldsymbol{\delta} \quad \text{with shift} \quad \boldsymbol{\delta} = \frac{\mathbf{v}^{(\ell)}(0)}{V_{\mathrm{th}}^{(\ell)}} .$$

Then, we use the clip and floor functions to determine $\mathbf{m}$,

$$\mathbf{m} = \mathrm{clip}\left(\left\lfloor \frac{T\mathbf{W}^{(\ell)}\bar{\mathbf{x}}^{(\ell-1)}}{V_{\mathrm{th}}^{(\ell)}} + \boldsymbol{\delta} \right\rfloor, 0, T\right) .$$

We provide the detailed analysis of the activation function of SNNs in Appendix A. Here the clip$(x, a, b)$ function sets the lower bound $a$ and upper bound $b$. Floor function $\lfloor x \rfloor$ gives the

greatest integer that is less than or equal to $x$. With $\bar{\mathbf{x}}^{(\ell)} = V_{\text{th}}^{(\ell)}\bar{\mathbf{s}}^{(\ell)} = \mathbf{m}V_{\text{th}}^{(\ell)}/T$, finally, the SNN activation function gives the relationship between activation values $\bar{\mathbf{x}}^{(\ell-1)}$ and $\bar{\mathbf{x}}^{(\ell)}$ as follows,

$$\bar{\mathbf{x}}^{(\ell)} = \mathcal{F}_{\text{SNN}}\left(\mathbf{W}^{(\ell)}\bar{\mathbf{x}}^{(\ell-1)}\right) = V_{\text{th}}^{(\ell)}\text{clip}\left(\frac{1}{T}\left\lfloor\frac{T\mathbf{W}^{(\ell)}\bar{\mathbf{x}}^{(\ell-1)}}{V_{\text{th}}^{(\ell)}} + \boldsymbol{\delta}\right\rfloor, 0, 1\right). \tag{8}$$

The SNN activation function $\mathcal{F}_{\text{SNN}}(\cdot)$ is a step function in interval $[0, V_{\text{th}}^{(\ell)}]$ with a step size $\frac{V_{\text{th}}^{(\ell)}}{T}$ (see the green curve in Fig. 1). Since the SNN output is discrete while the ANN output is continuous, there actually would be an intrinsic difference between $\boldsymbol{a}^{(\ell)}$ and $\bar{\mathbf{x}}^{(\ell)}$ as shown in Fig. 1.

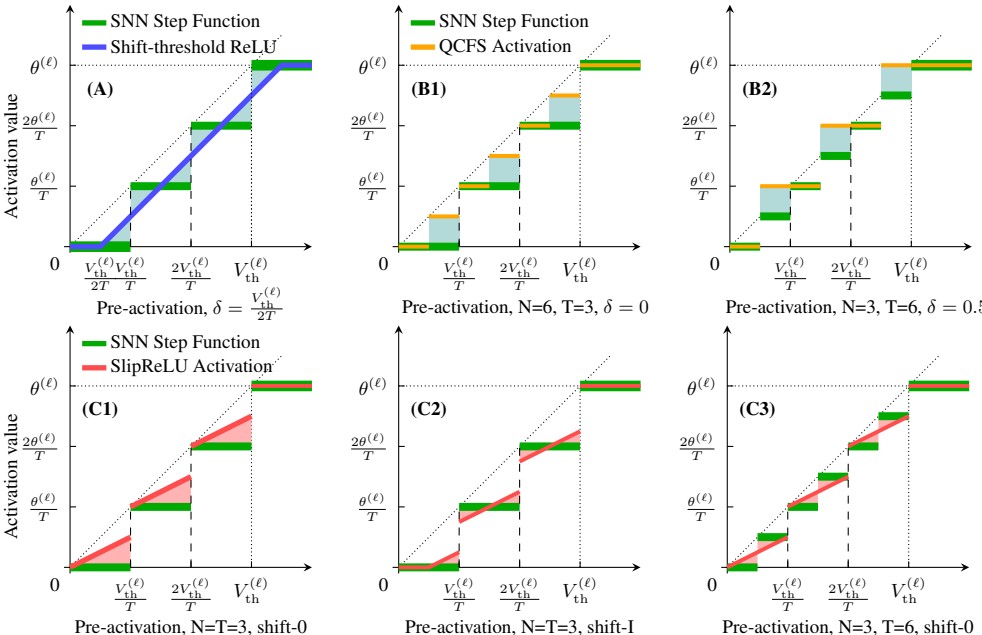

Figure 1: Activation functions of source ANNs, i.e., the shift-threshold-ReLU (blue curve) (Deng & Gu, 2021) in (A), quantization clip-floor-shift (QCFS) activation (orange curve) (Bu et al., 2021) in (B1)-(B2), and our proposed SlipReLU (red curve) in (C1)-(C3). The activation functions of target SNNs is the step function (green curve). The error between the activation function (of ANNs) and the step function (of SNNs) is obtained by summing up of all the shaded area together, which is the ANN-SNN conversion error.

## 4 PROPOSED SLIPRELU: OPTIMAL ANN-SNN CONVERSION

### 4.1 THE SLIPRELU ACTIVATION FUNCTION

In this section, by following our unified optimization framework, we will exploit the two-step conversion mechanism. We redesign the ANN with to get a tailored ANN, train the tailored ANN and convert it to SNN by copying the weights from the tailored ANN to the target SNN. Performance loss will occur if the the new activation function of the tailored ANN deviates too much from the regular ReLU activation function, and we need to minimize the conversion error at the same time. By keeping that in mind, an effective activation function $\mathcal{F}_{\text{ANN}}$ of the tailored ANN should not deviate too much from the regular ReLU and be close to the step function. Therefore, we propose the SlipReLU activation function which is a weighted sum of the threshold-ReLU and the step function to balance the trade-off between the regular ReLU and step function. Assume that both ANN and SNN receive the same input from the previous layer, $\boldsymbol{a}^{(\ell-1)} = \bar{\mathbf{x}}^{(\ell-1)}$. Denote $\mathbf{z}^{(\ell)} = \mathbf{W}^{(\ell)}\bar{\mathbf{x}}^{(\ell-1)} = \mathbf{W}^{(\ell)}\boldsymbol{a}^{(\ell-1)}$.

**Proposed SlipReLU activation function.** Following the unified optimization framework Sect. 3, new activation functions of the tailored ANNs are designed by minimizing the mismatch to the step

function of the target SNNs, and minimizing the deviation from the regular ReLU. We propose the SlipReLU activation function for the tailored ANN (see the red curves in (C1)-(C3) of Fig. 1),

$$\mathcal{F}_{\text{ANN}}(\mathbf{z}^{(\ell)}) = c\theta^{(\ell)}\text{clip}\left(\frac{\mathbf{z}^{(\ell)}}{\theta^{(\ell)}}, 0, 1\right) + (1-c)\theta^{(\ell)}\text{clip}\left(\frac{1}{N}\left\lfloor\frac{N\mathbf{z}^{(\ell)}}{\theta^{(\ell)}}\right\rfloor, 0, 1\right) . \tag{9}$$

From definition in Eq. (9), the SlipReLu activation function is just a weighted sum of the threshold-ReLU (first part) and the step function (second part), with the slope $0 \leqslant c \leqslant 1$ balancing its weight.

With some linear algebra, the SlipReLU can be formulated as a piece-wise linear function with a constant slope $c$ (see the red curves in (C1)-(C3) of Fig. 1 and detailed derivation in Appendix B), $\text{SlipReLU}(\mathbf{z}^{(\ell)}) = c\mathbf{z}^{(\ell)} + (1-c)\frac{k\boldsymbol{\theta}^{(\ell)}}{N}, \ \frac{k\boldsymbol{\theta}^{(\ell)}}{N} \leqslant \mathbf{z}^{(\ell)} < \frac{(k+1)\boldsymbol{\theta}^{(\ell)}}{N}, \ k = 0, 1, \cdots, N-1$. Here $0 \leqslant c \leqslant 1$ is the constant slope of the piece-wise linear function. From the definition and red curves in (C1)-(C3) of Fig. 1, we see that the new proposed function is very similar to a slippery step function with a slope, hence the name "SlipReLU".

**SlipReLU extension**  The SlipReLU extension with shift (see Appendix B) can be formulated as

$$\mathcal{F}_{\text{ANN}}(\mathbf{z}^{(\ell)}) = c\theta^{(\ell)}\text{clip}\left(\frac{\mathbf{z}^{(\ell)}}{\theta^{(\ell)}} + \boldsymbol{\delta}_1, 0, 1\right) + (1-c)\theta^{(\ell)}\text{clip}\left(\frac{1}{N}\left\lfloor\frac{N\mathbf{z}^{(\ell)}}{\theta^{(\ell)}} + \boldsymbol{\delta}\right\rfloor, 0, 1\right) . \tag{10}$$

**An application of the unified optimization framework.**  Recall the step activation function of target SNNs in Eq. (8), by setting $c = 0$ the SlipReLU becomes the step function,

$$\boldsymbol{a}^{(\ell)} = \mathcal{F}_{\text{ANN}}(\mathbf{z}^{(\ell)}) = \theta^{(\ell)}\text{clip}\left(\frac{1}{N}\left\lfloor\frac{N\mathbf{z}^{(\ell)}}{\theta^{(\ell)}} + \boldsymbol{\delta}\right\rfloor, 0, 1\right) . \tag{11}$$

Note that Eq. (11) is exactly the step function in Eq. (8), except $V_{\text{th}}^{(\ell)} \leftarrow \theta^{(\ell)}, T \leftarrow N$. Because the latency $T$ in Eq. (8) is an inherent property of the target SNNs, so it cannot be used in the source ANN. Therefore, instead of $T$, we use quasi-time-steps $N$ in SNN. As mentioned in Sect. 3.2.1, the threshold $V_{\text{th}}^{(\ell)}$ in SNN can be different from layer to layer, and we make it a trainable value $\theta^{(\ell)}$ in the ANN which can be learned and copied to the target SNN. Coincidentally, Bu et al. (2021) uses this function defined in Eq. (11) as the activation function in the source ANNs, and they name it the quantization clip-floor-shift (QCFS) activation function and call $N$ the quantization steps.

**Special cases of SlipReLU**  Here we list some related works which fall in to our proposed unified optimization framework and are special cases of the SlipReLU. (1) When $c = 0, \boldsymbol{\delta} = [\frac{1}{2}]$, the proposed SlipRLU activation function becomes the quantization clip-floor-shift in Bu et al. (2021). It only considers to be close to the SNN step function, but neglects the deviation from the regular ReLU. (2) When $c = 1, \boldsymbol{\delta}_1 = [-\frac{1}{2N}]$, the proposed SlipRLU activation function becomes shift-threshold ReLU in Deng & Gu (2021). It considers the deviation from the regular ReLU but neglects the closeness to the step function of the target SNN. Our proposed SlipReLU balances the trade-off between the regular ReLU and the step function. Refer to Appendix B for the details.

## 4.2 THEOREMS ON THE CONVERSION ERROR

The following two theorems gives the conversion error of the proposed unified method.

**Theorem 1.** *An ANN trained with SlipReLU activation function Eq. (10) is converted to an SNN with the same weights. Let $V_{\text{th}}^{(\ell)} = \theta^{(\ell)}, \mathbf{v}^{(\ell)}(0) = V_{\text{th}}^{(\ell)}\boldsymbol{\delta}, c = 0$. Then for arbitrary $T$ and $N$, the expectation of the conversion error of the proposed unified method reaches $0$, i.e.,*

$$\forall \, T, L \quad \mathbb{E}_{\mathbf{z}}\left(\left|\text{Err}^{(\ell)}\right|\right)\Big|_{\delta \in [-\frac{1}{2}, \frac{1}{2}]} = \mathbf{0} , \tag{12}$$

*holds for any the shift term $\delta$ in the source ANN when $\delta \in \left[-\frac{1}{2}, \frac{1}{2}\right]$.*

Theorem 1 indicates that for $c = 0$ the expected conversion error reaches zero even though $N \neq T$ provided that the shift term $\delta \in [-\frac{1}{2}, \frac{1}{2}]$. The proof is in Appendix C.

**Theorem 2.** *An ANN trained with SlipReLU activation function Eq. (10) is converted to an SNN with the same weights. Let $V_{\mathrm{th}}^{(\ell)} = \theta^{(\ell)}$, $\mathbf{v}^{(\ell)}(0) = V_{\mathrm{th}}^{(\ell)}\boldsymbol{\delta}$, $\boldsymbol{\delta}_1 = [\frac{\delta - 1/2}{T}]$. Then for arbitrary $T$ and $N$ and arbitrary $c \in [0,1]$, the expectation of the conversion error of the proposed unified method reaches the optimal $\frac{c(V_{\mathrm{th}}^{(\ell)})^2}{4T}$, i.e.,*

$$\forall\, T, L \quad \mathbb{E}_{\mathbf{z}} \left( \left| \mathrm{Err}^{(\ell)} \right| \right) \Big|_{\delta \in [-\frac{1}{2}, \frac{1}{2}]} = \frac{c(V_{\mathrm{th}}^{(\ell)})^2}{4T} \,, \tag{13}$$

*holds for any the shift term $\delta$ in the source ANN when $\delta \in [-\frac{1}{2}, \frac{1}{2}]$.*

Theorem 2 indicates that for any $\forall\, c \in [0,1]$, the expectation of the conversion error can reach the minimum $\frac{c(V_{\mathrm{th}}^{(\ell)})^2}{4T}$, provided that the shift term $\delta$ in the source ANN is in the interval $[-\frac{1}{2}, \frac{1}{2}]$, and $\delta_1 = \frac{\delta - 1/2}{T}$. The proof is in Appendix C. These results indicate we can achieve high-performance converted SNN at ultra-low time-steps.

### 4.3 ALGORITHM FOR TRAINING SLIPRELU ACTIVATION FUNCTION IN BACKPROPAGATION

Training an ANN with SlipReLU activation instead of ReLU is also a challenging problem. Although the SlipReLU has a constant slope as its derivative from Eq. (9), however, small slope values $c \in [0,1]$ can cause the gradient vanishing problem. Therefore, inspired by Bu et al. (2021); Bengio et al. (2013), we use the surrogate gradient as the derivative of the floor function $\frac{d\lfloor x \rfloor}{x} = 1$. The overall derivation rule is given as follows,

$$\frac{d\mathcal{F}_{\mathrm{ANN}}(\mathbf{z}^{(\ell)})}{d\mathbf{z}_i^{(\ell)}} = \begin{cases} 1, & \text{if } \mathbf{z}_i^{(\ell)} \in D_1 \cup D_2 \\ 0, & \text{otherwise} \end{cases}$$

where $D_1 = [-\delta_1\theta, \theta - \delta_1\theta]$, $D_2 = [-\delta\theta, \theta - \delta\theta]$, and $\mathbf{z}_i^{(\ell)}$ is the $i$-th element of $\mathbf{z}^{(\ell)}$. Then we can train the ANN with SlipReLU activation using Stochastic Gradient Descent algorithm, and convert it to the SNN. Refer to Appendix D for our proposed ANN-SNN conversion algorithm.

## 5 RELATED WORK

The first study of ANN-SNN conversion is proposed by Cao et al. (2015), which convert the ANNs with the ReLU activation function to SNNs. Afterwards, Diehl et al. (2015) proposed data-based and model-based weight-normalization method to convert a three-layer CNN to an SNN. However, it usually requires hundreds of time-steps for the converted SNN to get accurate results due to the error analyzed in Sect. 3. To address the potential information loss, the "reset-by-subtraction" mechanism (Rueckauer et al., 2017), also called "soft-reset" (Han et al., 2020) rather than "reset-to-zero" is proposed. Recently, many methods and algorithms have been proposed to eliminate the conversion error. Sengupta et al. (2019) proposed a novel weight-normalization technique which considers the actual SNN operation in the conversion step. For direct conversion from a pre-trained ANN to an SNN, Ding et al. (2021) proposed Rate Norm Layer to replace the ReLU activation function in source ANN training, and Li et al. (2021) proposed calibration for weights and biases using quantized fine-tuning to correct the error layer-by-layer. Our work share similarity with Deng & Gu (2021); Bu et al. (2021) which are also on optimal conversion. Deng & Gu (2021) minimized the layer-wise error by shift-threshold ReLU which only considers the deviation from the ReLU in the unified optimization framework in Sect. 3. Bu et al. (2021) proposed to use the quantization clip-floor-shift activation function to train ANNs and the clip-floor-shift activation function only minimizes the conversion error neglecting the performance loss of the tailored ANN with new activation function. They all got the theoretical "optimal" results with some fixed shift term. In comparison, our proposed unified framework gives more flexibility for different application scenarios to covert ANN into SNN with techniques eliminating the conversion error and keeping the ANN performance with less deviation from the regular ANN with ReLU. Our SlipReLU can balance the trade-off between the ANN performance and the conversion error simultaneously.

## 6 EXPERIMENTS

In this section, we compare our SlipReLU method with existing state-of-the-art approaches for image classification task on CIFAR-10 (LeCun et al., 1998) and CIFAR-100 (Krizhevsky & Hinton,

Table 1: Comparison between the proposed SlipReLU method and previous works on CIFAR10.

| Architecture | Method | ANN | T=1 | T=2 | T=4 | T=8 | T=16 | T=32 | T=64 | T=128 |
|---|---|---|---|---|---|---|---|---|---|---|
| ResNet-20 | RTS | 91.46 | 10.00 | 10.00 | 10.00 | 10.00 | 10.00 | 21.10 | 90.34 | 90.54 |
| | SNNC-AP | 97.11 | 55.02 | 66.56 | 82.03 | 89.38 | 94.75 | 96.23 | 96.83 | 97.02 |
| | QCFS | 91.29 | 56.60 | 67.53 | 79.73 | 88.06 | 90.99 | 91.73 | 91.69 | 91.56 |
| | ReLU | 93.71 | 11.58 | 12.54 | 16.05 | 36.47 | 70.84 | 83.47 | 85.93 | 86.46 |
| | **Ours** | **93.37** | **80.30** | **82.80** | **84.69** | **86.12** | **90.45** | **92.95** | **93.49** | **93.52** |
| VGG-16 | RTS | 92.09 | 10.00 | 10.00 | 10.00 | 10.00 | 10.00 | 89.48 | 91.84 | 91.25 |
| | RNL | 86.10 | 10.00 | 10.00 | 10.00 | 10.00 | 10.10 | 17.20 | 38.56 | 60.20 |
| | SNNC-AP | 95.93 | 73.82 | 75.16 | 86.58 | 90.26 | 92.87 | 94.53 | 95.41 | 95.78 |
| | QCFS | 92.69 | 75.51 | 83.81 | 88.58 | 91.47 | 92.50 | 92.83 | 92.83 | 92.90 |
| | ReLU | 95.92 | 10.00 | 10.00 | 11.51 | 70.97 | 88.39 | 93.05 | 94.76 | 95.19 |
| | **Ours** | **95.60** | **85.40** | **86.59** | **88.27** | **89.67** | **95.20** | **95.66** | **95.65** | **95.66** |
| ResNet-18 | RTS | 91.94 | 10.00 | 10.00 | 10.00 | 10.00 | 10.00 | 11.00 | 89.60 | 90.14 |
| | QCFS | 95.84 | 88.30 | 91.52 | 93.89 | 95.02 | 95.51 | 95.72 | 95.70 | 95.69 |
| | ReLU | 96.71 | 11.00 | 25.07 | 55.21 | 73.80 | 88.44 | 94.50 | 96.00 | 96.50 |
| | **Ours** | **96.67** | **93.11** | **93.97** | **94.59** | **94.92** | **95.08** | **96.31** | **96.53** | **96.52** |

2009). Similar to previous works, we utilize VGG-16, ResNet-18, and ResNet-20 network structures for source ANNs. We compare our method with the state-of-the-art ANN-SNN conversion methods, including RNL from Ding et al. (2021), ReLU-Threshold-Shift (RTS) from Deng & Gu (2021), SNN Conversion with Advanced Pipeline (SNNC-AP) from Li et al. (2021), and ANN-SNN conversion with quantization clip-floor-shift activation function (QCFS) from Bu et al. (2021), as well as ANN-SNN conversion with regular ReLU activation function (ReLU). Refer to Appendix E for the network structures and training setups. We use SlipReLU with shift setting $\delta_1 = 0, \delta = 0.5$, refer to Appendix H for ablation studies on SlipReLU and SlipReLU-shift activation.

## 6.1 COMPARISON WITH THE STATE-OF-THE-ART METHODS

Table 1 shows the performance comparison of the proposed SlipReLU with the state-of-the-art ANN-SNN conversion methods on CIFAR-10. For ultra-low latency inference ($T = 1$ or $T = 2$), our proposed SlipReLU has the best performance compared to existing state-of-the-art ANN-SNN conversion methods. Specially, when the latency $T = 1$, our SlipReLU method is able to achieve an accuracy of $93.11\%$ for ResNet-18, with a good margin compared to the next best baseline QCFS ($88.30\%$); the accuracy for VGG-16 is $85.40\%$ with SlipReLU activation, while the next best accuracy is $75.51\%$ with QCFS activation. For ResNet-20, we achieve an accuracy of $82.8\%$ with 2 time-steps. Our proposed SlipReLU method indeed gives the best SNN accuracy for ultra-low latency inference. When considering low-latency inference ($T \leqslant 4$), our model outperforms almost all the other methods with the same time-step setting. Notably, our ultra-low latency performance is comparable with other state-of-the-art supervised training methods, which is shown in Table S2.

The most competitive method of our SlipReLU is the QCFS method, however, it cannot to provide as high performance as our SlipReLU in terms of ANN accuracy, which can be seen from the ANN testing accuracy. Here the ANN accuracy of ReLU activation is the baseline. The results with SlipReLU activation shows that it has higher ANN accuracy than the QCFS (step function). The reason is that QCFS only considers the conversion error but not the ANN performance, while our SlipReLU proposes to consider the conversion error as well as the ANN performance. The highest ANN accuracy can sometimes be achieved by the SNNC-AP method, which is a one-step conversion method, however, the SNNC-AP usually fails to give moderate accuracy for low-latency SNN inference. Considering both ANN accuracy and SNN inference accuracy, the SlipReLU performs the best among all the other state-of-the-art models and it is the closest to the regular ReLU activation function, as our SlipReLU method is designed to consider both the ANN accuracy and the conversion error simultaneously. We further test the performance of our method on the large-scale dataset. Experimental results on CIFAR-100 and Tiny-ImageNet datasets are reported in Table S3 and Table S4 of Appendix G.

## 6.2 EFFECT OF THE SLOPE $c$ AND EFFECT OF THE QUASI-LATENCY $N$

In our SlipReLU method, the slope $c$ balances the weight of the threshold ReLU and the step function, which affects the accuracy of the converted SNN. To analyze the effect of $c$ and better deter-

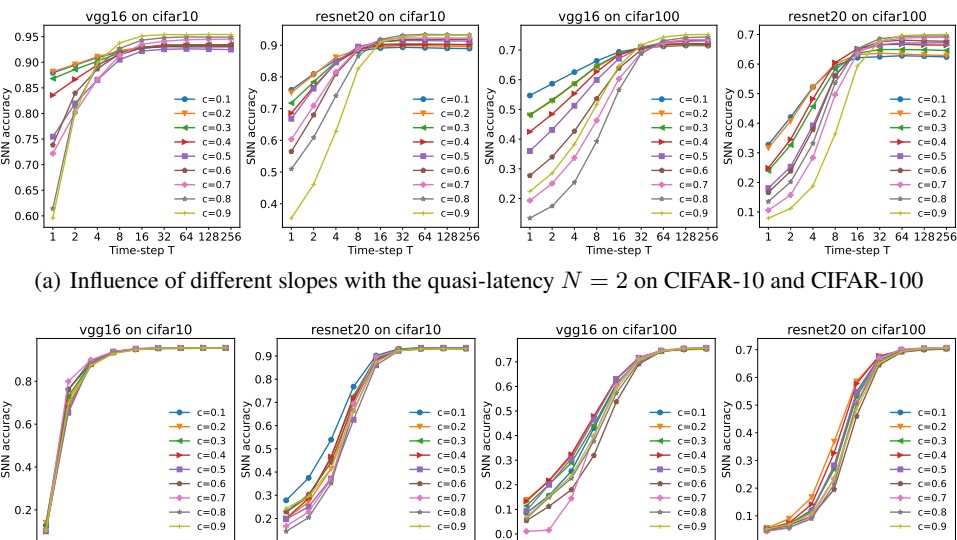

(a) Influence of different slopes with the quasi-latency $N = 2$ on CIFAR-10 and CIFAR-100

(b) Influence of different slopes with the quasi-latency $N = 32$ on CIFAR-10 and CIFAR-100

Figure 2: Effect of different slopes $c$ with different quasi-latency $N$ on CIFAR-10 and CIFAR-100.

mine the optimal value, we train VGG-16 and ResNet-20 networks with quasi-latency $N = 2$ and $N = 32$, and then convert the trained ANNs to SNNs. The experimental results on CIFAR-10 and CIFAR-100 dataset are shown in Fig. 2, where each of the colored curves shows the effect of the slope $c$ on the SNN accuracy over different time-step/latency $T$, under different quasi-latency $N$.

Results in Fig. 2 show that for small values of quasi-latency $N$, the slope $c$ has a large effect on SNN accuracy for ultra-low and low-latency inference. In particular, for small quasi-latency $N$, different slope values $c$ can result in different SNN accuracy when the time-step $T$ is small. But for large values of quasi-latency $N$, the colored curves are close to each other, and different values of slope $c$ give similar results no matter whether the time-step $T$ is small or large. This brings the flexibility to apply our SlipReLU to different scenarios. When we need ultra-low/low- latency inference for the converted SNN, we choose small quasi-latency $N$, but when we do not care about the inference time (the time-step $T$ can be large), we then choose large quasi-latency $N$. Refer to Appendix I for more detailed results.

## 7 DISCUSSION AND CONCLUSION

The performance of the converted SNN is determined by both the ANN performance and the conversion error. The performance loss between the regular ANN with regular ReLU and the tailored ANN has never been considered in the existing ANN-SNN conversion methods, which will be inherited to the converted SNN. In this work, we formulate the ANN-SNN conversion as a unified optimization problem which considers the performance loss between the regular ANN with and the tailored ANN, as well as the conversion error simultaneously. Following the unified optimization framework, we propose the SlipReLU activation function to replace the regular ReLU activation function in the tailored ANN. The SlipReLU is a weighted sum of the shift-threshold-ReLU and the step function, which improves the performance of either as an activation function alone. The SlipReLU method covers a family of activation functions mapping from activation values in source ANNs to firing rates in target SNNs; most of the state-of-the-art optimal ANN-SNN conversion methods are special cases of our proposed SlipReLU method. We demonstrate through two theorems that the expected conversion error between SNNs and ANNs can theoretically be zero on a range of shift values $\delta \in [-\frac{1}{2}, \frac{1}{2}]$ rather than a fixed shift term $\frac{1}{2}$, enabling us to achieve converted SNNs with high accuracy and ultra-low latency. We evaluate our proposed SlipReLU method on CIFAR-10 dataset, and the results show that our proposed SlipReLU outperforms the state-of-the-art ANN-SNN conversion in both accuracy and latency. To our knowledge, this is the first work to explore high-performance ANN-SNN conversion method considering the ANN performance and the conversion error simultaneously, with ultra-low latency, especially for 1 time-step ($T = 1$).

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

## NOTATIONS IN THE APPENDIX

Throughout the paper and this Appendix, we use the following notations in Table S1. Bold-face lower-case letters refer to vectors, and normal-face letters refer to scalars. Note $\mathbf{V}_{\text{th}}^{(\ell)}$ and $\boldsymbol{\theta}^{(\ell)}$ are vectors whose dimensions match the number of neurons in the layer of interest, and denote $\mathbf{V}_{\text{th}}^{(\ell)} = [V_{\text{th}}^{(\ell)}]$ and $\boldsymbol{\theta}^{(\ell)} = [\theta^{(\ell)}]$ respectively. Namely, vector $\mathbf{V}_{\text{th}}^{(\ell)} = [V_{\text{th}}^{(\ell)}]$ means that each element is the same $V_{\text{th}}^{(\ell)}$. Denote $\boldsymbol{\delta} = [\delta]$.

Table S1: Summary of notations in this paper.

| Symbol | Definition | Symbol | Definition |
|---|---|---|---|
| $N$ | Quasi-time-steps of ANNs | $\mathcal{F}_{\text{ANN}}(\cdot)$ | ANN activation function |
| $T$ | Total time-steps of SNNs | $\mathcal{F}_{\text{SNN}}(\cdot)$ | SNN activation function |
| $\boldsymbol{a}^{(\ell)}$ | Activation values of ANNs | $\mathbf{s}^{(\ell)}(t)$ | Spike outputs of SNN |
| $\bar{\mathbf{x}}^{(\ell)}$ | Average PSP of SNNs | $\mathbf{x}^{(\ell)}(t)$ | PSP released by $l$-th layer |
| $\theta^{(\ell)}$ | Trainable threshold in ANNs | $\mathbf{v}^{(\ell)}(t)$ | Membrane potential after firing |
| $V_{th}^{(\ell)}$ | Firing threshold in SNNs | $\mathbf{W}^{(\ell)}$ | Weight matrix |

# A    ANALYSIS OF ACTIVATION FUNCTION IN SNNS

We will derive the activation function of SNN, $\mathcal{F}_{\text{SNN}}(\cdot)$ in this section.

The activation function of SNN gives the relationship between activation values $\bar{\mathbf{x}}^{(\ell-1)}$ and $\bar{\mathbf{x}}^{(\ell)}$ of successive layers of SNN, which defines input-output function mapping for adjacent layers.

Specifically, we can get the potential update equation by combining Eq. (2) and Eq. (4),

$$\mathbf{v}^{(\ell)}(t) = \mathbf{v}^{(\ell)}(t-1) + \mathbf{W}^{(\ell)}\mathbf{x}^{(\ell-1)}(t) - \mathbf{s}^{(\ell)}(t)V_{\text{th}}^{(\ell)} . \tag{A.1}$$

By summing the time-step from time $1$ to $T$, then we get

$$\mathbf{v}^{(\ell)}(T) - \mathbf{v}^{(\ell)}(0) = \mathbf{W}^{(\ell)}\sum_{t=1}^{T}\mathbf{x}^{(\ell-1)}(t) - \sum_{t=1}^{T}\mathbf{s}^{(\ell)}(t)V_{\text{th}}^{(\ell)} . \tag{A.2}$$

Due to the spike-in-spike-out property of the IF neurons in SNN, the output at each time-step can be ether 0 or 1. For each neuron $i$, let $m_i = \sum_{t=1}^{T} s_i^{(\ell)}(t)$, and each $m_i \in \{0, 1, 2, \cdots, T\}$ denotes the total number of spikes of each neuron $i$. Then $\mathbf{m} = \{m_i\}$ is the vector collecting all the number of spikes of all neurons in the $\ell$-th layer. The accumulated spikes $\mathbf{m} = \sum_{t=1}^{T}\mathbf{s}^{(\ell)}(t)$ denotes the total number of spikes. According to the above equations, we have

$$\mathbf{v}^{(\ell)}(T) - \mathbf{v}^{(\ell)}(0) = \mathbf{W}^{(\ell)}T \cdot \bar{\mathbf{x}}^{(\ell-1)} - \mathbf{m}V_{\text{th}}^{(\ell)} . \tag{A.3}$$

Then we get

$$\mathbf{m}V_{\text{th}}^{(\ell)} = T\mathbf{W}^{(\ell)}\bar{\mathbf{x}}^{(\ell-1)} - (\mathbf{v}^{(\ell)}(T) - \mathbf{v}^{(\ell)}(0)) . \tag{A.4}$$

## A.1    ELEMENT-WISE VERSION DERIVATION

Denote

$$\mathbf{z}^{(\ell)} = \mathbf{W}^{(\ell)}\bar{\mathbf{x}}^{(\ell-1)} .$$

We use $z_i^{(\ell)}$, $v_i^{(\ell)}(T)$, $v_i^{(\ell)}(0)$, and $m_i$ to denote the $i$-th element in vector $\mathbf{z}^{(\ell)}$, $\mathbf{v}^{(\ell)}(T)$, $\mathbf{v}^{(\ell)}(0)$, and $\mathbf{m}$ respectively. That is, $\mathbf{z}^{(\ell)} = \{z_i^{(\ell)}\}$, $\mathbf{v}^{(\ell)}(T) = \{v_i^{(\ell)}(T)\}$, $\mathbf{v}^{(\ell)}(0) = \{v_i^{(\ell)}(0)\}$, and $\mathbf{m} = \{m_i\}$.

Then we have

$$\mathbf{m}V_{\text{th}}^{(\ell)} = T\mathbf{z}^{(\ell)} - (\mathbf{v}^{(\ell)}(T) - \mathbf{v}^{(\ell)}(0))$$

$$\iff m_iV_{\text{th}}^{(\ell)} = Tz_i^{(\ell)} - (v_i^{(\ell)}(T) - v_i^{(\ell)}(0)) \text{ (For each neuron } i \text{ with } \mathbf{m} = \{m_i\}, \mathbf{z}^{(\ell)} = \{z_i^{(\ell)}\}) .$$

Note that we assume the terminal membrane potential $v_i^{(\ell)}(T)$ lies within the range $[0, V_{\text{th}}^{(\ell)})$, by further assuming $v_i^{(\ell)}(0) = 0$, we get

$$0 \leqslant v_i^{(\ell)}(T) < V_{\text{th}}^{(\ell)}$$

$$\iff -V_{\text{th}}^{(\ell)} < -v_i^{(\ell)}(T) \leqslant 0 \quad \text{(adding } Tz_i^{(\ell)} \text{ to each term)}$$

$$\iff Tz_i^{(\ell)} - V_{\text{th}}^{(\ell)} < Tz_i^{(\ell)} - v_i^{(\ell)}(T) \leqslant Tz_i^{(\ell)} \quad (m_i = Tz_i^{(\ell)} - v_i^{(\ell)}(T))$$

$$\iff Tz_i^{(\ell)} - V_{\text{th}}^{(\ell)} < m_iV_{\text{th}}^{(\ell)} \leqslant Tz_i^{(\ell)}$$

$$\iff \frac{Tz_i^{(\ell)} - V_{\text{th}}^{(\ell)}}{V_{\text{th}}^{(\ell)}} < m_i \leqslant \frac{Tz_i^{(\ell)}}{V_{\text{th}}^{(\ell)}} .$$

Then we use floor operation and clip operation to determine the totoal number of spikes, $m_i$,

$$m_i = \text{clip}\left(\left\lfloor \frac{Tz_i^{(\ell)}}{V_{\text{th}}^{(\ell)}} \right\rfloor, 0, T\right) \text{ ( and } m_i = T\bar{s}_i^{(\ell)})$$

$$\bar{s}_i^{(\ell)} = \text{clip}\left(\frac{1}{T}\left\lfloor \frac{Tz_i^{(\ell)}}{V_{\text{th}}^{(\ell)}} \right\rfloor, 0, 1\right) \text{ ( and } \bar{x}_i^{(\ell)} = V_{\text{th}}^{(\ell)}\bar{s}_i^{(\ell)})$$

$$\bar{x}_i^{(\ell)} = V_{\text{th}}^{(\ell)}\text{clip}\left(\frac{1}{T}\left\lfloor \frac{Tz_i^{(\ell)}}{V_{\text{th}}^{(\ell)}} \right\rfloor, 0, 1\right) .$$

The assumption $v_i^{(\ell)}(0) = 0$ may be too strong, without it, we will get

$$\Longleftrightarrow Tz_i^{(\ell)} - V_{\text{th}}^{(\ell)} + v_i^{(\ell)}(0) < m_i V_{\text{th}}^{(\ell)} \leqslant Tz_i^{(\ell)} + v_i^{(\ell)}(0)$$

$$\Longleftrightarrow \frac{Tz_i^{(\ell)} - V_{\text{th}}^{(\ell)} + v_i^{(\ell)}(0)}{V_{\text{th}}^{(\ell)}} < m_i \leqslant \frac{Tz_i^{(\ell)} + v_i^{(\ell)}(0)}{V_{\text{th}}^{(\ell)}}$$

$$\Longleftrightarrow \frac{Tz_i^{(\ell)} - V_{\text{th}}^{(\ell)}}{V_{\text{th}}^{(\ell)}} + \delta < m_i \leqslant \frac{Tz_i^{(\ell)}}{V_{\text{th}}^{(\ell)}} + \delta \quad \text{with} \quad \delta = \frac{v_i^{(\ell)}(0)}{V_{\text{th}}^{(\ell)}} \ .$$

Denote $\delta = \frac{v_i^{(\ell)}(0)}{V_{\text{th}}^{(\ell)}}$. Then we have

$$m_i = \text{clip}\left(\left\lfloor \frac{Tz_i^{(\ell)}}{V_{\text{th}}^{(\ell)}} + \delta \right\rfloor, 0, T\right) \ (\text{ and } m_i = T\bar{s}_i^{(\ell)})$$

$$\bar{s}_i^{(\ell)} = \text{clip}\left(\frac{1}{T}\left\lfloor \frac{Tz_i^{(\ell)}}{V_{\text{th}}^{(\ell)}} + \delta \right\rfloor, 0, 1\right) \ (\text{ and } \bar{x}_i^{(\ell)} = V_{\text{th}}^{(\ell)}\bar{s}_i^{(\ell)})$$

$$\bar{x}_i^{(\ell)} = V_{\text{th}}^{(\ell)}\text{clip}\left(\frac{1}{T}\left\lfloor \frac{Tz_i^{(\ell)}}{V_{\text{th}}^{(\ell)}} + \delta \right\rfloor, 0, 1\right) \ .$$

The relationship between activation values $\bar{x}^{(\ell-1)}$ and $\bar{x}^{(\ell)}$ of successive layers of SNN can be formulated as

$$\bar{x}_i^{(\ell)} = V_{\text{th}}^{(\ell)}\text{clip}\left(\frac{1}{T}\left\lfloor \frac{Tz_i^{(\ell)}}{V_{\text{th}}^{(\ell)}} + \delta \right\rfloor, 0, 1\right) \ .$$

### A.2 VECTOR VERSION DERIVATION

The accumulated spikes $\mathbf{m} = \sum_{t=1}^{T} \mathbf{s}^{(\ell)}(t)$ denotes the total number of spikes, and $\mathbf{m} = \{m_i\}$ is the vector collecting all the number of spikes of all neurons in the $\ell$-th layer. Each $m_i \in \{0, 1, 2, \cdots, T\}$ denotes the total number of spikes of each neuron $i$. According to the above equations, we have

$$\mathbf{v}^{(\ell)}(T) - \mathbf{v}^{(\ell)}(0) = \mathbf{W}^{(\ell)}T \cdot \bar{x}^{(\ell-1)} - \mathbf{m}V_{\text{th}}^{(\ell)} \ . \tag{A.5}$$

Then we get

$$\mathbf{m}V_{\text{th}}^{(\ell)} = T\mathbf{W}^{(\ell)}\bar{x}^{(\ell-1)} - (\mathbf{v}^{(\ell)}(T) - \mathbf{v}^{(\ell)}(0)) \ . \tag{A.6}$$

Note that we assume the terminal membrane potential $\mathbf{v}^{(\ell)}(T)$ lies within the range $[\mathbf{0}, \mathbf{V}_{\text{th}}^{(\ell)})$, by further assuming $\mathbf{v}^{(\ell)}(0) = \mathbf{0}$, we get

$$\mathbf{0} \leqslant \mathbf{v}^{(\ell)}(T) < \mathbf{V}_{\text{th}}^{(\ell)}$$

$$\Longleftrightarrow -\mathbf{V}_{\text{th}}^{(\ell)} < -\mathbf{v}^{(\ell)}(T) \leqslant \mathbf{0}$$

$$\Longleftrightarrow T\mathbf{W}^{(\ell)}\bar{x}^{(\ell-1)} - \mathbf{V}_{\text{th}}^{(\ell)} < T\mathbf{W}^{(\ell)}\bar{x}^{(\ell-1)} - \mathbf{v}^{(\ell)}(T) \leqslant T\mathbf{W}^{(\ell)}\bar{x}^{(\ell-1)}$$

$$\Longleftrightarrow T\mathbf{W}^{(\ell)}\bar{x}^{(\ell-1)} - \mathbf{V}_{\text{th}}^{(\ell)} < \mathbf{m}V_{\text{th}}^{(\ell)} \leqslant T\mathbf{W}^{(\ell)}\bar{x}^{(\ell-1)}$$

$$\Longleftrightarrow \frac{T\mathbf{W}^{(\ell)}\bar{x}^{(\ell-1)} - \mathbf{V}_{\text{th}}^{(\ell)}}{V_{\text{th}}^{(\ell)}} < \mathbf{m} \leqslant \frac{T\mathbf{W}^{(\ell)}\bar{x}^{(\ell-1)}}{V_{\text{th}}^{(\ell)}} \ .$$

Then we use floor operation and clip operation to determine the totoal number of spikes, $\mathbf{m}$,

$$\mathbf{m} = \text{clip}\left(\left\lfloor \frac{T\mathbf{W}^{(\ell)}\bar{x}^{(\ell-1)}}{V_{\text{th}}^{(\ell)}} \right\rfloor, 0, T\right) \quad (\text{ and } \mathbf{m} = T\bar{s}^{(\ell)})$$

$$\bar{s}^{(\ell)} = \text{clip}\left(\frac{1}{T}\left\lfloor \frac{T\mathbf{W}^{(\ell)}\bar{x}^{(\ell-1)}}{V_{\text{th}}^{(\ell)}} \right\rfloor, 0, 1\right) \quad (\text{ and } \bar{x}^{(\ell)} = V_{\text{th}}^{(\ell)}\bar{s}^{(\ell)})$$

$$\bar{x}^{(\ell)} = V_{\text{th}}^{(\ell)}\text{clip}\left(\frac{1}{T}\left\lfloor \frac{T\mathbf{W}^{(\ell)}\bar{x}^{(\ell-1)}}{V_{\text{th}}^{(\ell)}} \right\rfloor, 0, 1\right) \ .$$

The assumption $\mathbf{v}^{(\ell)}(0) = \mathbf{0}$ may be too strong, without it, we will get

$$\Longleftrightarrow T\mathbf{W}^{(\ell)}\bar{\mathbf{x}}^{(\ell-1)} - \mathbf{V}_{\mathrm{th}}^{(\ell)} + \mathbf{v}^{(\ell)}(0) < \mathbf{m}V_{\mathrm{th}}^{(\ell)} \leqslant T\mathbf{W}^{(\ell)}\bar{\mathbf{x}}^{(\ell-1)} + \mathbf{v}^{(\ell)}(0)$$

$$\Longleftrightarrow \frac{T\mathbf{W}^{(\ell)}\bar{\mathbf{x}}^{(\ell-1)} - \mathbf{V}_{\mathrm{th}}^{(\ell)} + \mathbf{v}^{(\ell)}(0)}{V_{\mathrm{th}}^{(\ell)}} < \mathbf{m} \leqslant \frac{T\mathbf{W}^{(\ell)}\bar{\mathbf{x}}^{(\ell-1)} + \mathbf{v}^{(\ell)}(0)}{V_{\mathrm{th}}^{(\ell)}}$$

$$\Longleftrightarrow \frac{T\mathbf{W}^{(\ell)}\bar{\mathbf{x}}^{(\ell-1)} - \mathbf{V}_{\mathrm{th}}^{(\ell)}}{V_{\mathrm{th}}^{(\ell)}} + \boldsymbol{\delta} < \mathbf{m} \leqslant \frac{T\mathbf{W}^{(\ell)}\bar{\mathbf{x}}^{(\ell-1)}}{V_{\mathrm{th}}^{(\ell)}} + \boldsymbol{\delta} \quad \text{with} \quad \boldsymbol{\delta} = \frac{\mathbf{v}^{(\ell)}(0)}{V_{\mathrm{th}}^{(\ell)}} .$$

Denote $\boldsymbol{\delta} = \frac{\mathbf{v}^{(\ell)}(0)}{V_{\mathrm{th}}^{(\ell)}}$. Note $\boldsymbol{\delta}$ is a vector whose dimension matches the number of neurons in that layer. Then we have

$$\mathbf{m} = \mathrm{clip}\left(\left\lfloor \frac{T\mathbf{W}^{(\ell)}\bar{\mathbf{x}}^{(\ell-1)}}{V_{\mathrm{th}}^{(\ell)}} + \boldsymbol{\delta} \right\rfloor, 0, T\right) \quad (\text{ and } \mathbf{m} = T\bar{\mathbf{s}}^{(\ell)})$$

$$\bar{\mathbf{s}}^{(\ell)} = \mathrm{clip}\left(\frac{1}{T}\left\lfloor \frac{T\mathbf{W}^{(\ell)}\bar{\mathbf{x}}^{(\ell-1)}}{V_{\mathrm{th}}^{(\ell)}} + \boldsymbol{\delta} \right\rfloor, 0, 1\right) \quad (\text{ and } \bar{\mathbf{x}}^{(\ell)} = V_{\mathrm{th}}^{(\ell)}\bar{\mathbf{s}}^{(\ell)})$$

$$\bar{\mathbf{x}}^{(\ell)} = V_{\mathrm{th}}^{(\ell)}\mathrm{clip}\left(\frac{1}{T}\left\lfloor \frac{T\mathbf{W}^{(\ell)}\bar{\mathbf{x}}^{(\ell-1)}}{V_{\mathrm{th}}^{(\ell)}} + \boldsymbol{\delta} \right\rfloor, 0, 1\right) .$$

The relationship between activation values $\bar{\mathbf{x}}^{(\ell-1)}$ and $\bar{\mathbf{x}}^{(\ell)}$ of successive layers of SNN can be formulated as

$$\bar{\mathbf{x}}^{(\ell)} = V_{\mathrm{th}}^{(\ell)}\mathrm{clip}\left(\frac{1}{T}\left\lfloor \frac{T\mathbf{W}^{(\ell)}\bar{\mathbf{x}}^{(\ell-1)}}{V_{\mathrm{th}}^{(\ell)}} + \boldsymbol{\delta} \right\rfloor, 0, 1\right) .$$

Note $\mathbf{V}_{\mathrm{th}}^{(\ell)}$ is a vector whose dimension matches the number of neurons in that layer, and $\mathbf{V}_{\mathrm{th}}^{(\ell)} = [V_{\mathrm{th}}^{(\ell)}]$ means each element is the same $V_{\mathrm{th}}^{(\ell)}$.

Denote

$$\mathbf{z}^{(\ell)} = \mathbf{W}^{(\ell)}\bar{\mathbf{x}}^{(\ell-1)} .$$

Then

$$\bar{\mathbf{x}}^{(\ell)} = V_{\mathrm{th}}^{(\ell)}\mathrm{clip}\left(\frac{1}{T}\left\lfloor \frac{T\mathbf{z}^{(\ell)}}{V_{\mathrm{th}}^{(\ell)}} + \boldsymbol{\delta} \right\rfloor, 0, 1\right) .$$

## B DERIVATION OF SLIPRELU ACTIVATION FUNCTION

In this section, we will give detailed derivation of the proposed SlipReLU activation function in Eq. (9) and its extension in Eq. (10) with different shift modes. In ANNs, denote

$$\mathbf{z}^{(\ell)} = \mathbf{W}^{(\ell)}\mathbf{x}^{(\ell-1)} .$$

Then the forward propagation of activation values through layers in the ANN is

$$\boldsymbol{a}^{(\ell)} = \mathcal{F}_{\mathrm{ANN}}(\mathbf{z}^{(\ell)}) = \mathcal{F}_{\mathrm{ANN}}(\mathbf{W}^{(\ell)}\mathbf{x}^{(\ell-1)}) .$$

### B.1 DERIVATION OF SLIPRELU ACTIVATION FUNCTION

**Derivation of SlipReLU activation function in Eq. (9).** We start with the initial definition of the SlipReLU function in Eq. (B.1),

$$\mathrm{SlipReLU}(\mathbf{z}^{(\ell)}) = \begin{cases} \mathbf{0} & \text{if } \mathbf{z}^{(\ell)} < 0 \\ c\mathbf{z}^{(\ell)} + (1-c)\frac{k\boldsymbol{\theta}^{(\ell)}}{N} & \text{if } \frac{k\boldsymbol{\theta}^{(\ell)}}{N} \leqslant \mathbf{z}^{(\ell)} < \frac{(k+1)\boldsymbol{\theta}^{(\ell)}}{N} \\ \boldsymbol{\theta}^{(\ell)} & \text{if } \mathbf{z}^{(\ell)} \geqslant \boldsymbol{\theta}^{(\ell)} \end{cases} . \qquad (\text{B.1})$$

Here $k = 0, 1, \cdots, N-1$. Note $\boldsymbol{\theta}^{(\ell)}$ should be a vector whose dimension matches the number of neurons in that layer, $\boldsymbol{\theta}^{(\ell)} = [\theta^{(\ell)}]$.

Then we can rewrite it to

$$\text{SlipReLU}(\mathbf{z}^{(\ell)}) = \mathbf{y}_{\text{temp}} + c \cdot (\mathbf{z}_{\text{temp}} - \mathbf{y}_{\text{temp}}) = c\mathbf{z}_{\text{temp}} + (1-c)\mathbf{y}_{\text{temp}}$$

$$\text{where} \quad \mathbf{z}_{\text{temp}} = \theta^{(\ell)}\text{clip}\left(\frac{\mathbf{z}^{(\ell)}}{\theta^{(\ell)}}, 0, 1\right), \text{and } \mathbf{y}_{\text{temp}} = \frac{\theta^{(\ell)}}{N}\left\lfloor \frac{N \cdot \mathbf{z}_{\text{temp}}}{\theta^{(\ell)}} \right\rfloor.$$

Here

$$\mathbf{y}_{\text{temp}} = \frac{\theta^{(\ell)}}{N}\left\lfloor \frac{N\mathbf{z}_{\text{temp}}}{\theta^{(\ell)}} \right\rfloor$$

$$\Longleftrightarrow \mathbf{y}_{\text{temp}} = \frac{\theta^{(\ell)}}{N}\left\lfloor N \cdot \text{clip}\left(\frac{\mathbf{z}^{(\ell)}}{\theta^{(\ell)}}, 0, 1\right) \right\rfloor$$

$$\Longleftrightarrow \mathbf{y}_{\text{temp}} = \theta^{(\ell)}\text{clip}\left(\frac{1}{N}\left\lfloor N \cdot \frac{\mathbf{z}^{(\ell)}}{\theta^{(\ell)}} \right\rfloor, 0, 1\right)$$

$$\Longleftrightarrow \mathbf{y}_{\text{temp}} = \theta^{(\ell)}\text{clip}\left(\frac{1}{N}\left\lfloor \frac{N\mathbf{z}^{(\ell)}}{\theta^{(\ell)}} \right\rfloor, 0, 1\right).$$

Then Eq. (B.1) can be written as follows,

$$\boldsymbol{a}^{(\ell)} = \text{SlipReLU}(\mathbf{z}^{(\ell)})$$
$$= c\mathbf{z}_{\text{temp}} + (1-c)\mathbf{y}_{\text{temp}}$$
$$= c\theta^{(\ell)}\text{clip}\left(\frac{\mathbf{z}^{(\ell)}}{\theta^{(\ell)}}, 0, 1\right) + (1-c)\theta^{(\ell)}\text{clip}\left(\frac{1}{N}\left\lfloor \frac{N\mathbf{z}^{(\ell)}}{\theta^{(\ell)}} \right\rfloor, 0, 1\right).$$

That is the SlipReLU activation function in Eq. (9),

$$\boldsymbol{a}^{(\ell)} = \mathcal{F}_{\text{ANN}}(\mathbf{z}^{(\ell)}) = c\theta^{(\ell)}\text{clip}\left(\frac{\mathbf{z}^{(\ell)}}{\theta^{(\ell)}}, 0, 1\right) + (1-c)\theta^{(\ell)}\text{clip}\left(\frac{1}{N}\left\lfloor \frac{N\mathbf{z}^{(\ell)}}{\theta^{(\ell)}} \right\rfloor, 0, 1\right).$$

## B.2 SLIPRELU ACTIVATION FUNCTION WITH DIFFERENT SHIFT MODES

**Derivation of SlipReLU extension in Eq. (10) with shift** As mentioned in Sect. 4, the SlipReLU activation function in Eq. (9) in a weighted combination of the threshold-ReLU (first part) and the step function (second part), with the slope $0 \leqslant c \leqslant 1$ balancing the weight, then any shift to these two parts will lead to shift in the SlipReLU activation function. The SlipReLU extension with in Eq. (10) can be formulated as follows,

$$\boldsymbol{a}^{(\ell)} = \mathcal{F}_{\text{ANN}}(\mathbf{z}^{(\ell)}) = c\theta^{(\ell)}\text{clip}\left(\frac{\mathbf{z}^{(\ell)}}{\theta^{(\ell)}} + \boldsymbol{\delta}_1, 0, 1\right) + (1-c)\theta^{(\ell)}\text{clip}\left(\frac{1}{N}\left\lfloor \frac{N\mathbf{z}^{(\ell)}}{\theta^{(\ell)}} + \boldsymbol{\delta} \right\rfloor, 0, 1\right).$$

The shift term $\delta_1 \in [-N, 0]$ and $\delta \in [-\frac{1}{2}, \frac{1}{2}]$ for the source ANNs. And $\boldsymbol{\delta}_1 = [\delta_1]$, $\boldsymbol{\delta} = [\delta]$.

Here we list several examples of the proposed SlipReLU with different shift modes.

Mode 0 We set $\delta_1 = \delta = 0$, then

$$\boldsymbol{a}^{(\ell)} = \mathcal{F}_{\text{ANN}}(\mathbf{z}^{(\ell)}) = c\theta^{(\ell)}\text{clip}\left(\frac{\mathbf{z}^{(\ell)}}{\theta^{(\ell)}}, 0, 1\right) + (1-c)\theta^{(\ell)}\text{clip}\left(\frac{1}{N}\left\lfloor \frac{N\mathbf{z}^{(\ell)}}{\theta^{(\ell)}} \right\rfloor, 0, 1\right).$$

Mode 1 We set $\delta_1 = 0, \delta = \frac{1}{2}$, then

$$\boldsymbol{a}^{(\ell)} = \mathcal{F}_{\text{ANN}}(\mathbf{z}^{(\ell)}) = c\theta^{(\ell)}\text{clip}\left(\frac{\mathbf{z}^{(\ell)}}{\theta^{(\ell)}}, 0, 1\right) + (1-c)\theta^{(\ell)}\text{clip}\left(\frac{1}{N}\left\lfloor \frac{N\mathbf{z}^{(\ell)}}{\theta^{(\ell)}} + [\frac{1}{2}] \right\rfloor, 0, 1\right).$$

## B.3 SPECIAL CASES OF THE SLIPRELU ACTIVATION FUNCTION

Here we list four different special cases of the proposed SlipReLU.

**Threshold-ReLU** When $c = 1$ and $\delta_1 = 0$, the SlipReLU becomes the threshold ReLU activation function which is studied in Deng & Gu (2021).

**Shift-threshold-ReLU** When $c = 1$ and $\delta_1 = -1/(2N)$, the SlipReLU becomes the shift-threshold ReLU activation function which is studied in Deng & Gu (2021).

**Quantization clip-floor (QCF)** When $c = 0$ and $\delta = 0$, the SlipReLU becomes the quantization clip-floor (QCF) activation function which is studied in Bu et al. (2021).

**Quantization clip-floor-shift (QCFS)** When $c = 0$ and $\delta = 1/2$, the SlipReLU becomes the quantization clip-floor-shift (QCFS) activation function which is studied in Bu et al. (2021).

## C  PROOF OF THEOREMS

Before we proof Theorem 1 and Theorem 2, we first introduce an important Lemma.

**Lemma 1.** *If a random variable* $x \in [0, \theta]$ *is uniformly distributed in every small interval* $(m_t, m_{t+1})$ *with* $p_t$ *(*$t = 0, 1, \cdots, T$*), where* $m_0 = 0, m_{T+1} = \theta, m_t = \frac{(2t-1)\theta}{2T}$ *for* $t = 1, 2, \cdots, T$*,* $p_0 = p_T$*. For any value* $\delta \in [-\frac{1}{2}, \frac{1}{2}]$*, then we can conclude that*

$$\mathbb{E}_x \left( \left| x - \frac{\theta}{T} \left\lfloor \frac{Tx}{\theta} + \delta \right\rfloor \right| \right) = 0 . \tag{C.1}$$

*Proof.* We consider $x$ in different small intervals $(m_t, m_{t+1})$.

(1) For $x \in \left(0, \frac{\theta}{2T}\right)$,

$$0 < x < \frac{\theta}{2T} \iff \delta < \frac{Tx}{\theta} + \delta < \frac{1}{2} + \delta \iff \left\lfloor \frac{Tx}{\theta} + \delta \right\rfloor = 0 .$$

(2) For $x \in \left( \frac{(2t-1)\theta}{2T}, \frac{(2t+1)\theta}{2T} \right)$, and $t = 1, 2, \cdots, T-1$

$$\frac{(2t-1)\theta}{2T} < x < \frac{(2t+1)\theta}{2T} \iff t - \frac{1}{2} + \delta < \frac{Tx}{\theta} + \delta < t + \frac{1}{2} + \delta \iff \left\lfloor \frac{Tx}{\theta} + \delta \right\rfloor = t .$$

(3) For $x \in \left( \frac{(2T-1)\theta}{2T}, \theta \right)$,

$$\frac{(2T-1)\theta}{2T} < x < \theta \iff T - \frac{1}{2} + \delta < \frac{Tx}{\theta} + \delta < T + \frac{1}{2} + \delta \iff \left\lfloor \frac{Tx}{\theta} + \delta \right\rfloor = T .$$

Then we have

$$\mathbb{E}_x \left( \left| x - \frac{\theta}{T} \left\lfloor \frac{Tx}{\theta} + \delta \right\rfloor \right| \right)$$

$$= \int_0^{\theta/2T} p_0 \left| x - \frac{\theta}{T} \left\lfloor \frac{Tx}{\theta} + \delta \right\rfloor \right| dx + \sum_{t=1}^{T-1} p_t \int_{(2t-1)\theta/2T}^{(2t+1)\theta/2T} \left| x - \frac{\theta}{T} \left\lfloor \frac{Tx}{\theta} + \delta \right\rfloor \right| dx$$

$$+ \int_{(2T-1)\theta/2T}^{\theta} p_T \left| x - \frac{\theta}{T} \left\lfloor \frac{Tx}{\theta} + \delta \right\rfloor \right| dx$$

$$= p_0 \int_0^{\theta/2T} |x| \, dx + \sum_{t=1}^{T-1} p_t \int_{(2t-1)\theta/2T}^{(2t+1)\theta/2T} \left| x - \frac{t\theta}{T} \right| dx + p_T \int_{(2T-1)\theta/2T}^{\theta} |x - \theta| \, dx$$

$$= p_0 \int_0^{\theta/2T} x \, dx + \sum_{t=1}^{T-1} p_t \int_{(2t-1)\theta/2T}^{(2t+1)\theta/2T} \left( x - \frac{t\theta}{T} \right) dx + p_T \int_{(2T-1)\theta/2T}^{\theta} (x - \theta) \, dx$$

$$= p_0 \frac{\theta^2}{8T^2} + 0 - p_T \frac{\theta^2}{8T^2} = 0 .$$

$\square$

**Lemma 2.** *Let* P *be a probability distribution on* $\mathbb{R}$. *If a random variable* $\mathbf{z} \in \mathbb{R}^m$ *and* $\mathbf{z} \sim$ P, *a function* $\boldsymbol{g} : \mathbf{z} \to \boldsymbol{g}(\mathbf{z}) \in \mathbb{R}^n$ *and* $\boldsymbol{g}(\mathbf{z}) \geqslant \mathbf{0}$ *almost surely for* $\forall\, \mathbf{z} \in D$, *and*

$$\mathbb{E}_{\mathbf{z}} \left| \boldsymbol{g}(\mathbf{z}) \right| = 0 \,,$$

*then we have*

$$\mathbb{E}_{\mathbf{z}} \left\| \boldsymbol{g}(\mathbf{z}) \right\|_2 = 0 \,.$$

*Proof.* By the definition of $L_2$-norm, we have

$$\left\| \boldsymbol{g}(\mathbf{z}) \right\|_2 = \sqrt{g_1^2(z) + g_2^2(z) + \cdots + g_n^2(z)} \leqslant |g_1(z)| + |g_2(z)| + \cdots + |g_n(z)| \,.$$

Then, we can get

$$\begin{aligned}
\mathbb{E}_{\mathbf{z}} \left\| \boldsymbol{g}(\mathbf{z}) \right\|_2 &\leqslant \mathbb{E}_{\mathbf{z}} |g_1(z)| + \mathbb{E}_{\mathbf{z}} |g_2(z)| + \cdots + \mathbb{E}_{\mathbf{z}} |g_n(z)| \\
&= \mathbb{E}_{\mathbf{z}} g_1(z) + \mathbb{E}_{\mathbf{z}} g_2(z) + \cdots + \mathbb{E}_{\mathbf{z}} g_n(z) = 0 \,.
\end{aligned}$$

Then

$$\mathbb{E}_{\mathbf{z}} \left\| \boldsymbol{g}(\mathbf{z}) \right\|_2 = 0 \,.$$

$\square$

## C.1 PROOF OF THEOREM 1

For Theorem 1, we need to prove

$$\forall\, T, L \quad \mathbb{E}_{\mathbf{z}} \left( \left| \text{Err}^{(\ell)} \right| \right) \Big|_{\delta \in [-\frac{1}{2}, \frac{1}{2}]} = \mathbf{0} \,.$$

*Proof.* The activation function of the SNN is

$$\mathcal{F}_{\text{SNN}}(\mathbf{z}^{(\ell)}) = V_{\text{th}}^{(\ell)} \text{clip} \left( \frac{1}{T} \left\lfloor \frac{T\mathbf{z}^{(\ell)} + \mathbf{v}^{(\ell)}(0)}{V_{\text{th}}^{(\ell)}} \right\rfloor, 0, 1 \right) \,.$$

For $c = 0$, the SlipReLU activation function used in the source ANN then becomes

$$\mathcal{F}_{\text{ANN}}(\mathbf{z}^{(\ell)}) = \theta^{(\ell)} \text{clip} \left( \frac{1}{N} \left\lfloor \frac{N\mathbf{z}^{(\ell)}}{\theta^{(\ell)}} + \boldsymbol{\delta} \right\rfloor, 0, 1 \right) \,.$$

With $V_{\text{th}}^{(\ell)} = \theta^{(\ell)}$, then the error becomes

$$\text{Err}^{(\ell)} = \mathcal{F}_{\text{SNN}}(\mathbf{z}^{(\ell)}) - \mathcal{F}_{\text{ANN}}(\mathbf{z}^{(\ell)}) = \frac{\theta^{(\ell)}}{N} \left\lfloor \frac{N\mathbf{z}^{(\ell)}}{\theta^{(\ell)}} + \boldsymbol{\delta} \right\rfloor - \frac{V_{\text{th}}^{(\ell)}}{T} \left\lfloor \frac{T\mathbf{z}^{(\ell)} + \mathbf{v}^{(\ell)}(0)}{V_{\text{th}}^{(\ell)}} \right\rfloor \,.$$

Then

$$\begin{aligned}
&\mathbb{E}_{\mathbf{z}} \left( \left| \text{Err}^{(\ell)} \right| \right) \Big|_{\delta \in [-\frac{1}{2}, \frac{1}{2}]} \\
=&\mathbb{E}_{\mathbf{z}} \left( \left| \frac{\theta^{(\ell)}}{N} \left\lfloor \frac{N\mathbf{z}^{(\ell)}}{\theta^{(\ell)}} + \boldsymbol{\delta} \right\rfloor - \frac{V_{\text{th}}^{(\ell)}}{T} \left\lfloor \frac{T\mathbf{z}^{(\ell)} + \mathbf{v}^{(\ell)}(0)}{V_{\text{th}}^{(\ell)}} \right\rfloor \right| \right) \\
\leqslant& \mathbb{E}_{\mathbf{z}} \left( \left| \frac{\theta^{(\ell)}}{N} \left\lfloor \frac{N\mathbf{z}^{(\ell)}}{\theta^{(\ell)}} + \boldsymbol{\delta} \right\rfloor - \mathbf{z}^{(\ell)} \right| \right) \Big|_{\delta \in [-\frac{1}{2}, \frac{1}{2}]} + \mathbb{E}_{\mathbf{z}} \left( \left| \mathbf{z}^{(\ell)} - \frac{V_{\text{th}}^{(\ell)}}{T} \left\lfloor \frac{T\mathbf{z}^{(\ell)} + \mathbf{v}^{(\ell)}(0)}{V_{\text{th}}^{(\ell)}} \right\rfloor \right| \right) \,.
\end{aligned}$$

Denote $\mathbf{v}_i^{(\ell)}(0)$ and $z_i$ the $i$-th element of vector $\mathbf{v}^{(\ell)}(0)$ and $\mathbf{z}$. Denote $\boldsymbol{\delta} = [\delta]$. Then we need to consider every element of vector $\mathbf{z}$.

$$\begin{aligned}
&\mathbb{E}_{z_i} \left( \left| \frac{\theta^{(\ell)}}{N} \left\lfloor \frac{N z_i^{(\ell)}}{\theta^{(\ell)}} + \delta \right\rfloor - \frac{V_{\text{th}}^{(\ell)}}{T} \left\lfloor \frac{T z_i^{(\ell)} + \mathbf{v}_i^{(\ell)}(0)}{V_{\text{th}}^{(\ell)}} \right\rfloor \right| \right) \\
\leqslant& \mathbb{E}_{z_i} \left( \left| \frac{\theta^{(\ell)}}{N} \left\lfloor \frac{N z_i^{(\ell)}}{\theta^{(\ell)}} + \delta \right\rfloor - z_i^{(\ell)} \right| \right) \Big|_{\delta \in [-\frac{1}{2}, \frac{1}{2}]} + \mathbb{E}_{z_i} \left( \left| z_i^{(\ell)} - \frac{V_{\text{th}}^{(\ell)}}{T} \left\lfloor \frac{T z_i^{(\ell)} + \mathbf{v}_i^{(\ell)}(0)}{V_{\text{th}}^{(\ell)}} \right\rfloor \right| \right) \,.
\end{aligned}$$

(C.2)

Then according to Lemma 1, we have

$$\mathbb{E}_{\mathbf{z}_i}\left(\left|\frac{\theta^{(\ell)}}{N}\left\lfloor\frac{N\mathbf{z}_i^{(\ell)}}{\theta^{(\ell)}}+\delta\right\rfloor-\mathbf{z}_i^{(\ell)}\right|\right)\Big|_{\delta\in[-\frac{1}{2},\frac{1}{2}]}=0$$

$$\mathbb{E}_{\mathbf{z}_i}\left(\left|\mathbf{z}_i^{(\ell)}-\frac{V_{\mathrm{th}}^{(\ell)}}{T}\left\lfloor\frac{T\mathbf{z}_i^{(\ell)}+\mathbf{v}_i^{(\ell)}(0)}{V_{\mathrm{th}}^{(\ell)}}\right\rfloor\right|\right)\Big|_{\mathbf{v}_i^{(\ell)}(0)=\delta V_{\mathrm{th}}^{(\ell)}}=0.$$

This holds for any shift value $\delta$ in the ANNs when $-\frac{1}{2}\leqslant\delta\leqslant\frac{1}{2}$, which gives the conclusion of the Theorem 1.

$$\forall\,T,L\quad\mathbb{E}_{\mathbf{z}}\left(\left|\mathrm{Err}^{(\ell)}\right|\right)\Big|_{\delta\in[-\frac{1}{2},\frac{1}{2}]}=\mathbf{0}.$$

$\square$

## C.2 PROOF OF THEOREM 2

For Theorem 2, we need to prove,

$$\forall\,T,L\quad\mathbb{E}_{\mathbf{z}}\left(\left|\mathrm{Err}^{(\ell)}\right|\right)\Big|_{\delta\in[-\frac{1}{2},\frac{1}{2}]}=\frac{c(V_{\mathrm{th}}^{(\ell)})^2}{4T},\tag{C.3}$$

*Proof.* The activation function of the SNN is

$$\mathcal{F}_{\mathrm{SNN}}(\mathbf{z}^{(\ell)})=V_{\mathrm{th}}^{(\ell)}\mathrm{clip}\left(\frac{1}{T}\left\lfloor\frac{T\mathbf{z}^{(\ell)}+\mathbf{v}^{(\ell)}(0)}{V_{\mathrm{th}}^{(\ell)}}\right\rfloor,0,1\right).$$

For arbitrary $c\in[0,1]$, the SlipReLU activation function used in the source ANN then becomes

$$\mathcal{F}_{\mathrm{ANN}}(\mathbf{z}^{(\ell)})=c\theta^{(\ell)}\mathrm{clip}\left(\frac{\mathbf{z}^{(\ell)}}{\theta^{(\ell)}}+\boldsymbol{\delta}_1,0,1\right)+(1-c)\theta^{(\ell)}\mathrm{clip}\left(\frac{1}{N}\left\lfloor\frac{N\mathbf{z}^{(\ell)}}{\theta^{(\ell)}}+\boldsymbol{\delta}\right\rfloor,0,1\right).$$

With $V_{\mathrm{th}}^{(\ell)}=\theta^{(\ell)}$, then the error becomes,

$$\mathrm{Err}^{(\ell)}=\mathcal{F}_{\mathrm{ANN}}(\mathbf{z}^{(\ell)})-\mathcal{F}_{\mathrm{SNN}}(\mathbf{z}^{(\ell)})$$

$$=c\left\{\theta^{(\ell)}\mathrm{clip}\left(\frac{\mathbf{z}^{(\ell)}}{\theta^{(\ell)}}+\boldsymbol{\delta}_1,0,1\right)-V_{\mathrm{th}}^{(\ell)}\mathrm{clip}\left(\frac{1}{T}\left\lfloor\frac{T\mathbf{z}^{(\ell)}+\mathbf{v}^{(\ell)}(0)}{V_{\mathrm{th}}^{(\ell)}}\right\rfloor,0,1\right)\right\}$$

$$+(1-c)\left\{\theta^{(\ell)}\mathrm{clip}\left(\frac{1}{N}\left\lfloor\frac{N\mathbf{z}^{(\ell)}}{\theta^{(\ell)}}+\boldsymbol{\delta}\right\rfloor,0,1\right)-V_{\mathrm{th}}^{(\ell)}\mathrm{clip}\left(\frac{1}{T}\left\lfloor\frac{T\mathbf{z}^{(\ell)}+\mathbf{v}^{(\ell)}(0)}{V_{\mathrm{th}}^{(\ell)}}\right\rfloor,0,1\right)\right\}$$

$$=c\left\{\mathbf{z}^{(\ell)}+\boldsymbol{\delta}_1\theta^{(\ell)}-\frac{V_{\mathrm{th}}^{(\ell)}}{T}\left\lfloor\frac{T\mathbf{z}^{(\ell)}+\mathbf{v}^{(\ell)}(0)}{V_{\mathrm{th}}^{(\ell)}}\right\rfloor\right\}\quad(\text{with }\mathbf{v}^{(\ell)}(0)=V_{\mathrm{th}}^{(\ell)}\boldsymbol{\delta},V_{\mathrm{th}}^{(\ell)}=\theta^{(\ell)})$$

$$+(1-c)\left\{\frac{\theta^{(\ell)}}{N}\left\lfloor\frac{N\mathbf{z}^{(\ell)}+\mathbf{v}^{(\ell)}(0)}{\theta^{(\ell)}}\right\rfloor-\frac{V_{\mathrm{th}}^{(\ell)}}{T}\left\lfloor\frac{T\mathbf{z}^{(\ell)}+\mathbf{v}^{(\ell)}(0)}{V_{\mathrm{th}}^{(\ell)}}\right\rfloor\right\}$$

$$=c\left\{\mathbf{z}^{(\ell)}+V_{\mathrm{th}}^{(\ell)}\boldsymbol{\delta}_1-\frac{V_{\mathrm{th}}^{(\ell)}}{T}\left\lfloor\frac{T\mathbf{z}^{(\ell)}}{V_{\mathrm{th}}^{(\ell)}}+\boldsymbol{\delta}\right\rfloor\right\}\overset{\Delta}{=}c\cdot\mathrm{Err}_1\tag{C.4}$$

$$+(1-c)\left\{\frac{\theta^{(\ell)}}{N}\left\lfloor\frac{N\mathbf{z}^{(\ell)}}{\theta^{(\ell)}}+\boldsymbol{\delta}\right\rfloor-\frac{V_{\mathrm{th}}^{(\ell)}}{T}\left\lfloor\frac{T\mathbf{z}^{(\ell)}+\mathbf{v}^{(\ell)}(0)}{V_{\mathrm{th}}^{(\ell)}}\right\rfloor\right\}\overset{\Delta}{=}(1-c)\cdot\mathrm{Err}_2\tag{C.5}$$

Then

$$\mathrm{Err}^{(\ell)}\overset{\Delta}{=}c\cdot\mathrm{Err}_1+(1-c)\cdot\mathrm{Err}_2$$

$$\implies\left|\mathrm{Err}^{(\ell)}\right|=|c\cdot\mathrm{Err}_1+(1-c)\cdot\mathrm{Err}_2|\leqslant c\cdot|\mathrm{Err}_1|+(1-c)\cdot|\mathrm{Err}_2|.$$

So we can minimize the whole error by minimized each of the two terms.

Let $\delta_1 = \frac{\phi+\delta}{T}$. For Eq. (C.4), we have

$$|\text{Err}_1| \triangleq \left| \mathbf{z}^{(\ell)} + \frac{V_{\text{th}}^{(\ell)}}{T}(\phi + \delta) - \frac{V_{\text{th}}^{(\ell)}}{T} \left\lfloor \frac{T\mathbf{z}^{(\ell)}}{V_{\text{th}}^{(\ell)}} + \delta \right\rfloor \right|$$

$$= \left| \mathbf{z}^{(\ell)} + \frac{V_{\text{th}}^{(\ell)}}{T}\phi - \frac{V_{\text{th}}^{(\ell)}}{T} \left\lfloor \frac{T\mathbf{z}^{(\ell)}}{V_{\text{th}}^{(\ell)}} \right\rfloor \right| . \tag{C.6}$$

Here

$$\mathbf{z}^{(\ell)} + \frac{V_{\text{th}}^{(\ell)}}{T}\phi \qquad \text{is the activation function of ANN}$$

$$\frac{V_{\text{th}}^{(\ell)}}{T} \left\lfloor \frac{T\mathbf{z}^{(\ell)}}{V_{\text{th}}^{(\ell)}} \right\rfloor \qquad \text{is the step activation function of SNN .}$$

This Eq. (C.6) recovers the loss of the shift-threshold ReLU (with a shift value $\phi$) and the step function, which is the same as Deng & Gu (2021). And as shown in (A) of Fig. 1, the conversion error is the shaded area. The error between the activation function (of ANNs) and the step function (of SNNs) is obtained by summing up of all the shaded area together, which is the ANN-SNN conversion error.

Then the objective becomes minimize

$$\min_\phi \left\{ \mathbb{E}_{\mathbf{z}} |\text{Err}_1| \right\} = \min_\phi \frac{T}{2} \left[ \left( \frac{V_{\text{th}}^{(\ell)}}{T} + \frac{V_{\text{th}}^{(\ell)}}{T}\phi \right)^2 + \left( \frac{V_{\text{th}}^{(\ell)}}{T}\phi \right)^2 \right] = \frac{(V_{\text{th}}^{(\ell)})^2}{4T} \implies \phi = -\frac{1}{2} .$$

Then

$$\delta_1 = \frac{-1/2 + \delta}{T} . \tag{C.7}$$

And the minimum $L_2$-norm of the first error becomes

$$\mathbb{E}_{\mathbf{z}} (|\text{Err}_1|) = \frac{(V_{\text{th}}^{(\ell)})^2}{4T} .$$

For Eq. (C.5), with $\mathbf{v}^{(\ell)}(0) = V_{\text{th}}^{(\ell)}\delta, V_{\text{th}}^{(\ell)} = \theta^{(\ell)}$, we have

$$|\text{Err}_2| \triangleq \left| \frac{\theta^{(\ell)}}{N} \left\lfloor \frac{N\mathbf{z}^{(\ell)}}{\theta^{(\ell)}} + \delta \right\rfloor - \frac{V_{\text{th}}^{(\ell)}}{T} \left\lfloor \frac{T\mathbf{z}^{(\ell)} + \mathbf{v}^{(\ell)}(0)}{V_{\text{th}}^{(\ell)}} \right\rfloor \right|$$

From Lemma 1 and Theorem 1, we have

$$\mathbb{E}_{\mathbf{z}} (|\text{Err}_2|) = \mathbb{E}_{\mathbf{z}} \left( \left| \frac{\theta^{(\ell)}}{N} \left\lfloor \frac{N\mathbf{z}^{(\ell)}}{\theta^{(\ell)}} + \delta \right\rfloor - \frac{V_{\text{th}}^{(\ell)}}{T} \left\lfloor \frac{T\mathbf{z}^{(\ell)} + \mathbf{v}^{(\ell)}(0)}{V_{\text{th}}^{(\ell)}} \right\rfloor \right| \right) \Big|_{\delta \in [-\frac{1}{2}, \frac{1}{2}]}$$

$$= 0 .$$

Then

$$\mathbb{E}_{\mathbf{z}} \left( \left| \text{Err}^{(\ell)} \right| \right) = \mathbb{E}_{\mathbf{z}} \left( \left| \mathcal{F}_{\text{ANN}}(\mathbf{z}^{(\ell)}) - \mathcal{F}_{\text{SNN}}(\mathbf{z}^{(\ell)}) \right| \right) = \mathbb{E}_{\mathbf{z}} \left( |c \cdot \text{Err}_1 + (1-c) \cdot \text{Err}_2| \right)$$

$$\leqslant c \cdot \mathbb{E}_{\mathbf{z}} (|\text{Err}_1|) + (1-c) \cdot \mathbb{E}_{\mathbf{z}} (|\text{Err}_2|)$$

$$= c \cdot \frac{(V_{\text{th}}^{(\ell)})^2}{4T} + (1-c) \cdot 0$$

$$= \frac{c(V_{\text{th}}^{(\ell)})^2}{4T} .$$

This concludes the Theorem 2.

$$\forall T, L \quad \mathbb{E}_{\mathbf{z}} \left( \left| \text{Err}^{(\ell)} \right| \right) \Big|_{\delta \in [-\frac{1}{2}, \frac{1}{2}]} = \frac{c(V_{\text{th}}^{(\ell)})^2}{4T} .$$

$\square$

# D  PSEUDO-CODE FOR THE UNIFIED ANN-SNN CONVERSION ALGORITHM

Here is the pseudo-code for our proposed unified ANN-SNN conversion algorithm.

---

**Algorithm 1:** Algorithm for ANN-SNN conversion.

---

**Input:** ANN model structure $f_{\text{ANN}}(\mathbf{x}; \mathbf{W})$ with initial weights $\mathbf{W} = \{\mathbf{W}^{(\ell)}\}$; Quasi-latency $N$; Shift value $\delta$ from the interval $\delta \in [-\frac{1}{2}, \frac{1}{2}]$; Initial dynamic threshold $\boldsymbol{\theta} = \{\boldsymbol{\theta}^{(\ell)}\}$; Learning rate $\epsilon$.

**Output:** SNN model $f_{\text{SNN}}(\mathbf{x}; \mathbf{W})$

**Data:** Dataset $D$

1 **for** $\ell = 1$ *to* $f_{\text{ANN}}.layers$ **do**
2    **if** *is ReLU activation* **then**
3      Replace $\text{ReLU}(\mathbf{x})$ by $\text{SlipReLU}(\mathbf{x}; N, \theta^{(\ell)})$
4    **if** *is MaxPooling layer* **then**
5      Replace MaxPooling layer by AvgPooling layer

6 **for** $e = 1$ *to epochs* **do**
7    **for** *length of Dataset $D$* **do**
8      Sample minibach $\{(\mathbf{x}^{(0)}, \mathbf{y})\}$ from $D$
9      **for** $\ell = 1$ *to* $f_{\text{ANN}}.layers$ **do**
10       $\mathbf{x}^{(\ell)} = \text{SlipReLU}(\mathbf{W}^{(\ell)}\mathbf{x}^{(\ell-1)}; N, \theta^{(\ell)})$
11    $Loss = CrossEntropy(\mathbf{x}^{(\ell)}, \mathbf{y})$
12    **for** $\ell = 1$ *to* $f_{\text{ANN}}.layers$ **do**
13      $\mathbf{W}^{(\ell)} \leftarrow \mathbf{W}^{(\ell)} - \epsilon \frac{\partial Loss}{\partial \mathbf{W}^{(\ell)}}$
14      $\theta^{(\ell)} \leftarrow \theta^{(\ell)} - \epsilon \frac{\partial Loss}{\partial \theta^{(\ell)}}$

15 **for** $\ell = 1$ *to* $f_{\text{ANN}}.layers$ **do**
16    $f_{\text{SNN}}.\mathbf{W}^{(\ell)} \leftarrow f_{\text{ANN}}.\mathbf{W}^{(\ell)}$
17    $f_{\text{SNN}}.V_{\text{th}}^{(\ell)} \leftarrow f_{\text{ANN}}.\theta^{(\ell)}$
18    $f_{\text{SNN}}.\mathbf{v}^{(\ell)}(0) \leftarrow f_{\text{SNN}}.V_{\text{th}}^{(\ell)} \times \delta$
19 **Return** $f_{\text{SNN}}$

---

# E   EXPERIMENTS DETAILS

## E.1   NETWORK STRUCTURE AND TRAINING SETUPS

There are three steps in our proposed ANN-SNN conversion,
**Step 1:** Tailor the ANN;
**Step 2:** Train the tailored ANN;
**Step 3:** Convert the trained ANN to an SNN.

In the first step, we first replace max-pooling with average-pooling and then replace the ReLU activation with the proposed SlipReLU activation function. The tailored ANN is also called the source ANN. In the second step, we train the tailored ANN. After training the tailored ANN, we copy all weights from the trained-tailored source ANN to the converted SNN, and set the threshold $V_{\text{th}}^{(\ell)}$ in each layer of the converted SNN equal to the threshold value $\theta^{(\ell)}$ of the source ANN in the same layer. Besides, we set the initial membrane potential $\mathbf{v}^{(\ell)}(0)$ in converted SNN as $V_{\text{th}}^{(\ell)}\delta$ to match the optimal shift $\delta$ of the SlipReLU activation in the tailored source ANN, where the optimal shift $\delta$ can be any value in the interval $\delta \in [-\frac{1}{2}, \frac{1}{2}]$.

Common data normalization and some data pre-processing techniques are used in the experiments. For example, we resize the images in the CIFAR-10/CIFAR-100 datasets into $32 \times 32$. Besides, random cropping images, Cutout (DeVries & Taylor, 2017) and AutoAugment (Cubuk et al., 2019) are used for all datasets. The Stochastic Gradient Descent (SGD) optimizer (Bottou, 2012) is used in the experiments with a momentum parameter of 0.9. We set the initial learning rate to $\epsilon = 0.1$ for CIFAR-10 and CIFAR-100. We use a cosine decay scheduler (Loshchilov & Hutter, 2017) to adjust the learning rate with a weight decay $5 \times 10^{-4}$ for CIFAR-10/CIFAR-100 datasets. All models are trained for 300 epochs. When considering small quasi-latency $N = 1$ and $N = 2$, for models that can not be trained properly with learning rate $\epsilon = 0.1$, we set the initial learning rate to 0.05 for CIFAR-10/CIFAR-100. We train all the networks on CIFAR-10/CIFAR-100 dataset with two different settings; we set the quasi-latency $N = 1$ with the slope $c = 0.3, 0.4, 0.5$ for low-latency inference ($T \leqslant 8$), and we set the quasi-latency $N = 4$ with the slope $c = 0.9$ for latency $T > 8$. We set $\delta_1 = 0, \delta = \frac{1}{2}$ for the SlipReLU activation for all the models and all the datasets.

As for the input to the first layer and the output of the last layer of the SNN, we do not employ any spiking mechanism as in Li et al. (2021). We directly encode the static image to temporal dynamic spikes as input to the first layer, which can prevent the undesired information loss introduced by the Poisson encoding. For the last layer output, we only integrate the pre-synaptic input and do not fire any spikes. We use constant input when evaluating the converted SNNs.

## E.2   INTRODUCTION OF DATASETS

**CIFAR-10**: The CIFAR-10 dataset (Krizhevsky & Hinton, 2009) consists of $60,000$ $32 \times 32$ color images in 10 classes of objects such as airplanes, cars, and birds, with $6,000$ images per class. There are $50,000$ samples in the training set and $10,000$ samples in the test set.

**CIFAR-100**: The CIFAR-100 dataset (Krizhevsky & Hinton, 2009) consists of $60,000$ $32 \times 32$ color images in 100 classes with $6,000$ images per class. There are $50,000$ samples in the training set and $10,000$ samples in the test set.

**Tiny-ImageNet**: Tiny-ImageNet (Le & Yang, 2015) is a subset of ImageNet-1k (Russakovsky et al., 2015) with 200 classes. Training data contains a total of $100,000$ images, 500 images from each class. The test data contains a total of $10,000$ images, 50 images per class. The dimension of the images is $64 \times 64$.

# F   COMPARISON WITH THE STATE-OF-THE-ART SUPERVISED TRAINING METHODS ON CIFAR-10 DATASET

Our proposed SlipReLU method is comparable with other state-of-the-art supervised training methods in terms of the ultra-low latency performance. Table S2 reports the results of the proposed models against the state-of-the-art supervised training methods on CIFAR10 dataset. These state-of-the-art supervised training methods include Hybrid-Conversion (HC) from Rathi et al. (2020),

STBP from Wu et al. (2018), TSSL from Zhang & Li (2020) and GDDP from Zheng et al. (2021), which are back-propagation or hybrid training methods.

Our approach for CIFARNet achieves an accuracy of $95.31\%$ with time-step $T = 4$, and the achieved accuracy is higher than any other supervised trained models. Sufficient time-step is required for back-propagation to train the SNN directly. The hybrid training method involves training the converted SNN model using back-propagation as a second step, still it requires 200 time-steps to achieve a good accuracy, which is high compared to the time-step required for our method. For VGG-16, the hybrid training method requires 200 time-steps to obtain $92.03\%$ accuracy, whereas our method achieves $91.08\%$ accuracy with 4 time-steps.

Table S2: Comparison with state-of-the-art supervised training methods on CIFAR-10 dataset.

| Model | Method | Architecture | SNN Accuracy | Time-step T |
|-------|--------|--------------|--------------|-------------|
| HC | Hybrid | VGG-16 | 92.03 | 200 |
| STBP | Backprop | CIFARNet | 85.82 | 12 |
| GDDT | Backprop | CIFARNet | 87.35 | 4 |
| TSSL | Backprop | CIFARNet | 88.23 | 5 |
| Ours | ANN-SNN | VGG-16 | 91.08 | 4 |
| Ours | ANN-SNN | ResNet-18 | 92.86 | 2 |
| Ours | ANN-SNN | CIFARNet | 95.31 | 4 |

## G  RESULTS ON CIFAR-100 DATASET AND TINY-IMAGENET DATASET

We report the results on CIFAR-100 in Table S3, and the results on Tiny-ImageNet in Table S4. From Table S3, we see that our SlipReLU method also outperforms the others both in terms of high accuracy and ultra-low latency. For VGG16, the accuracy of the proposed method can achieve an accuracy of $64.21\%$ which is $29.1\%$ higher than QCFS and $39.98\%$ higher than SNNC-AP when the time-steps is only 1. For ResNet-18, when $T = 1$, we can still achieve an accuracy of $71.51\%$. These results demonstrate that our method outperforms the previous conversion methods.

Training an SNN on large-scale dataset such as Tiny-ImageNet is considered to be one of the challenges in the SNN literature. From Table S4, we can infer our SlipReLU method outperforms other baselines in terms of the SNN accuracy when the time-step T is $\leqslant 4$. When the time-step is 1, for ResNet-34 our SlipReLU method achieves an accuracy of $40.55\%$ which is $6.94\%$ higher than the baseline QCFS ($33.61\%$). For VGG16, our SlipReLU method outperforms other baseline methods with the accuracy of $43.73\%$ when time-step is 1, which is $12.14\%$ better than the baseline QCFS ($31.59\%$) and $32.93\%$ better than the baseline SNNC-AP ($10.80\%$).

## H  COMPARISON OF SLIPRELU AND SLIPRELU-SHIFT ACTIVATION

Here we further conduct ablation studies on SlipReLU and SlipReLU-shift, by comparing the performance of SNNs converted from ANNs with SlipReLU activation and ANN with SlipReLU-shift activation. In Sect. 4, we prove that for arbitrary $T$ and $N$, the expectation of the conversion error reaches 0 with SlipReLU-shift activation function when $c = 0$. We also prove that for arbitrary $T$ and $N$ and arbitrary $c \in [0, 1]$, the expectation of the conversion error of the proposed unified method reaches the optimal $c(V_{\text{th}}^{(} \ell)^2)/(4T)$. To verify these, we set $N = 1, 2, 4, 8, 16, 32$ and train ANNs with SlipReLU activation and SlipReLU-shift activation, respectively.

Fig. S1 shows how the accuracy of converted SNNs changes with respect to the time-step $T$ under different quasi-latency $N$ settings. The accuracy of the converted SNN from ANN with SlipReLU activation (in the first and third columns) first increases or stays flat for time-step $T \leqslant 4$, and then decreases rapidly with the increase of time-steps, because we cannot guarantee that the conversion error is zero when $c \neq 0$. The best performance is still lower than the SlipReLU-shift activation. The non-shifted SlipReLU activation shows no advantage for ultra-low latency inference when $T \leqslant 4$. In contrast, the accuracy of the converted SNN from ANN with SlipReLU-shift activation (in the

Table S3: Comparison between the proposed SlipReLU method and previous works on CIFAR-100.

| Architecture | Method | ANN | T=1 | T=2 | T=4 | T=8 | T=16 | T=32 | T=64 | T=128 |
|---|---|---|---|---|---|---|---|---|---|---|
| ResNet-20 | RTS | 66.05 | 1.00 | 1.00 | 1.00 | 1.00 | 1.00 | 9.54 | 63.22 | 62.49 |
| | SNNC-AP | 81.08 | 16.16 | 25.71 | 49.60 | 65.97 | 75.80 | 76.64 | 80.63 | 80.90 |
| | QCFS | 65.08 | 15.91 | 23.93 | 40.57 | 57.59 | 64.90 | 66.65 | 66.74 | 66.67 |
| | ReLU | 70.18 | 1.28 | 1.16 | 1.76 | 2.91 | 4.03 | 6.17 | 8.95 | 11.66 |
| | **Ours** | 69.45 | 47.08 | 51.34 | 54.51 | 56.00 | 62.26 | 68.71 | 70.06 | 70.29 |
| VGG-16 | RTS | 70.59 | 1.00 | 1.00 | 1.00 | 1.00 | 1.08 | 64.13 | 69.88 | 68.74 |
| | SNNC-AP | 76.17 | 24.23 | 40.16 | 49.86 | 62.97 | 69.89 | 74.36 | 76.51 | 77.63 |
| | QCFS | 71.50 | 35.10 | 43.85 | 53.66 | 62.72 | 68.47 | 70.99 | 72.01 | 72.23 |
| | ReLU | 73.39 | 1.00 | 1.53 | 15.55 | 28.56 | 46.03 | 62.42 | 70.05 | 72.16 |
| | **Ours** | 75.25 | 64.21 | 66.30 | 67.97 | 69.31 | 70.09 | 71.84 | 74.61 | 75.13 |
| ResNet-18 | RTS | 50.01 | 1.00 | 1.00 | 1.00 | 1.00 | 1.00 | 15.70 | 44.64 | 41.17 |
| | RNL | 56.98 | 1.00 | 1.00 | 1.00 | 3.40 | 17.30 | 33.50 | 40.90 | 43.38 |
| | QCFS | 76.68 | 49.19 | 58.65 | 68.31 | 74.46 | 76.70 | 77.24 | 77.37 | 77.43 |
| | ReLU | 77.16 | 1.00 | 1.64 | 4.99 | 11.40 | 34.08 | 60.44 | 71.90 | 75.63 |
| | **Ours** | 78.56 | 71.51 | 73.91 | 74.89 | 75.40 | 75.46 | 77.79 | 78.24 | 78.55 |

Table S4: Comparison between the proposed SlipReLU and other methods on Tiny-ImageNet.

| Architecture | Method | ANN | T=1 | T=2 | T=4 | T=8 | T=16 | T=32 | T=64 | T=128 |
|---|---|---|---|---|---|---|---|---|---|---|
| ResNet-34 | QCFS | 56.87 | 33.61 | 41.11 | 47.95 | 53.98 | 56.76 | 57.65 | 57.53 | 56.94 |
| | **Ours** | 53.32 | 40.55 | 45.95 | 50.64 | 53.28 | 54.38 | 54.61 | 54.08 | 54.54 |
| VGG-16 | SNNC-AP | 57.15 | 10.80 | 18.51 | 28.88 | 41.37 | 49.29 | 53.55 | 55.80 | 56.73 |
| | QCFS | 55.59 | 31.59 | 43.43 | 49.72 | 54.08 | 55.48 | 56.02 | 55.97 | 55.93 |
| | **Ours** | 52.75 | 43.73 | 47.95 | 51.23 | 52.89 | 54.01 | 53.84 | 53.74 | 53.68 |

second and fourth columns) increases with the increase of time-step $T$. It converges to the same accuracy when the time-step is larger than 16. The SlipReLU-shift activation shows advantages for ultra-low latency inference when $T \leqslant 4$.

## I   EFFECT OF THE SLOPE $c$ AND THE QUASI-LATENCY $N$

In our SlipReLU method, the slope $c$ balances the weight of the threshold ReLU and the step function, which affects the accuracy of the converted SNN. To analyze the effect of $c$ and better determine the optimal value, we train VGG-16/ResNet-20 networks with quasi-latency $N = 1, 2, 4, 8, 16, 32$, and then converted the trained networks to SNNs. The experimental results on CIFAR-10/100 dataset are shown in Fig. S2, where each of the colored curves shows the effect of the slope $c$ on the SNN accuracy over different time-step/latency $T$, under different quasi-latency settings. Table S5, Table S6 and Table S7 are the detailed data used to plot the curves.

## J   FUTURE STUDY

**Remark 2.** *Our unified conversion framework exploits both the one-step conversion mechanism and the two-step conversion mechanism. The one-step conversion method uses a pre-trained source ANN, such as Li et al. (2021), however, the two-step conversion method needs to redesign the activation function of the ANN to get a tailored source ANN, train it and convert it to SNN, such as Deng & Gu (2021); Bu et al. (2021).*

**Remark 3.** *Usually, implicit variables of an optimization problem are variables which do not need to be optimized but are used to model feasibility conditions (Mehlitz & Benko, 2021), and they are often interpreted as explicit ones (Mehlitz & Benko, 2021), by using union of image sets associated*

*with given set-valued mappings to make the implicit variables as explicit variables, which can be an interesting future work but not what we are interested in this paper.*

As mentioned in Sect. 3.1, the multi-step output feature of SNN implies that higher-latency output depend on the outputs of all previous time-steps, which can be explored through multi-task learning. Therefore, it is reasonable to use multi-task learning for ANN-SNN conversion where the different time-steps can be seen as different but related tasks.

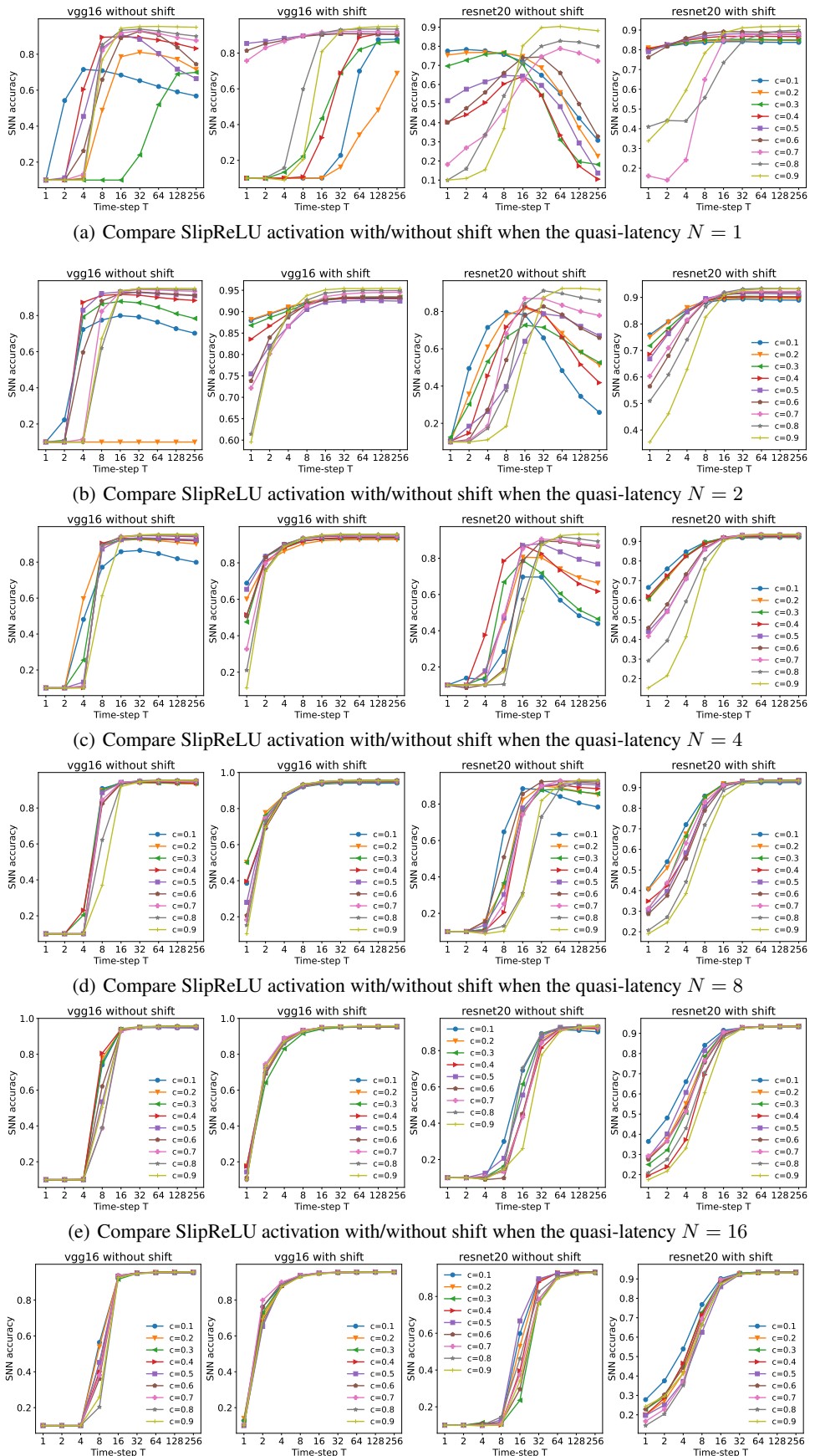

(a) Compare SlipReLU activation with/without shift when the quasi-latency $N = 1$

(b) Compare SlipReLU activation with/without shift when the quasi-latency $N = 2$

(c) Compare SlipReLU activation with/without shift when the quasi-latency $N = 4$

(d) Compare SlipReLU activation with/without shift when the quasi-latency $N = 8$

(e) Compare SlipReLU activation with/without shift when the quasi-latency $N = 16$

(f) Compare SlipReLU activation with/without shift when the quasi-latency $N = 32$

Figure S1: Ablation studies on SlipReLU activation and SlipReLU-shift activation on CIFAR-10 dataset under different slopes $c$ with different quasi-latency $N$.

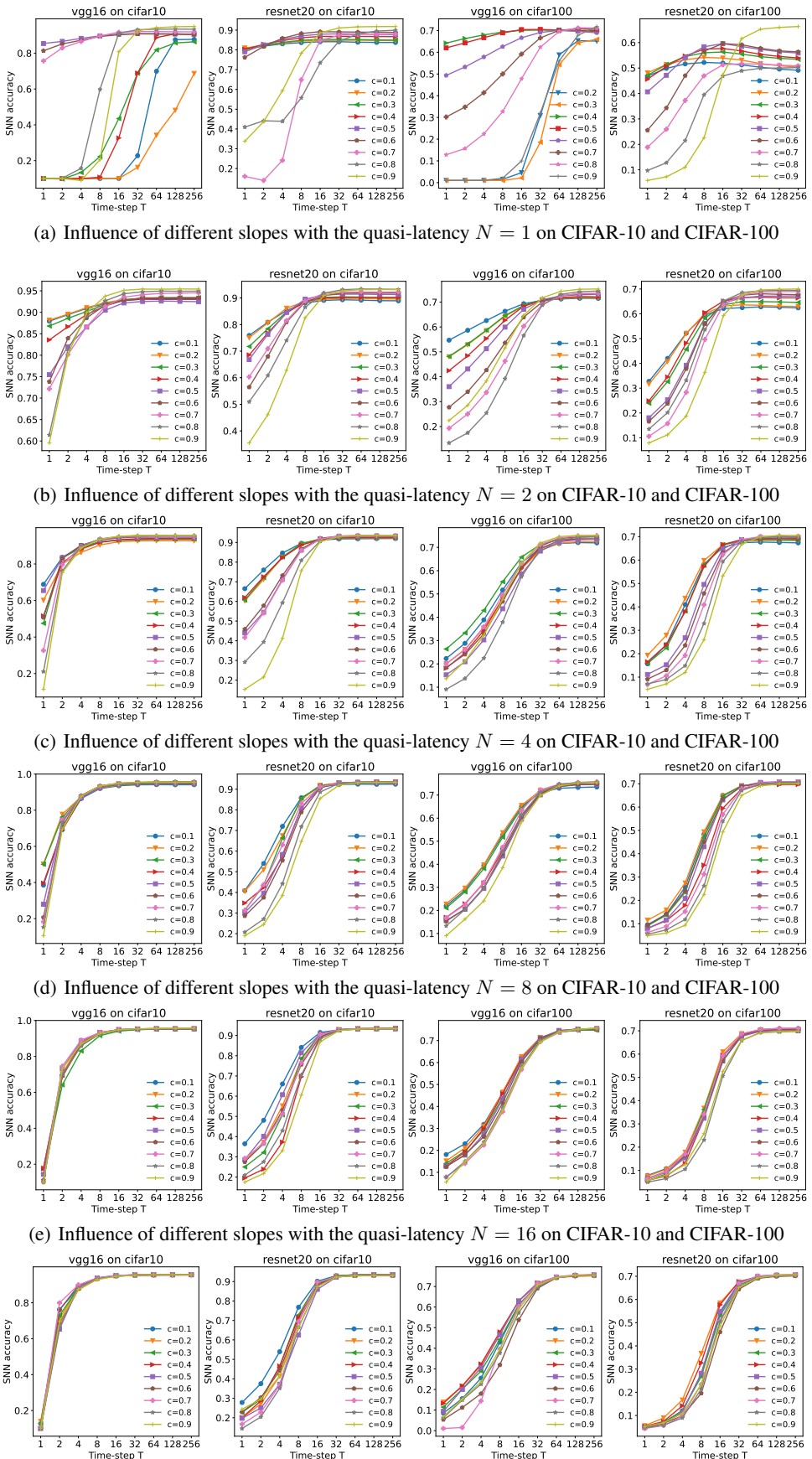

(a) Influence of different slopes with the quasi-latency $N = 1$ on CIFAR-10 and CIFAR-100

(b) Influence of different slopes with the quasi-latency $N = 2$ on CIFAR-10 and CIFAR-100

(c) Influence of different slopes with the quasi-latency $N = 4$ on CIFAR-10 and CIFAR-100

(d) Influence of different slopes with the quasi-latency $N = 8$ on CIFAR-10 and CIFAR-100

(e) Influence of different slopes with the quasi-latency $N = 16$ on CIFAR-10 and CIFAR-100

(f) Influence of different slopes with the quasi-latency $N = 32$ on CIFAR-10 and CIFAR-100

Figure S2: Effect of different slopes $c$ with different quasi-latency $N$ on CIFAR-10 and CIFAR-100.

Table S5: Influence of different slope $c$ with the quasi-latency $N = 1$.

| Slope $c$ | ANN Acc. | T=1 | T=2 | T=4 | T=8 | T=16 | T=32 | T=64 | T=128 | T=256 |
|---|---|---|---|---|---|---|---|---|---|---|
| | | | | **VGG-16 on CIFAR-10** | | | | | | |
| c=0.1 | 90.93 | 10.00 | 10.00 | 10.00 | 10.00 | 10.05 | 22.78 | 69.91 | 87.55 | 87.72 |
| c=0.2 | 88.68 | 10.00 | 10.00 | 10.00 | 10.00 | 10.00 | 16.25 | 34.22 | 48.16 | 68.69 |
| c=0.3 | 86.73 | 10.00 | 9.74 | 13.37 | 22.29 | 43.48 | 68.81 | 81.81 | 85.71 | 86.35 |
| c=0.4 | 91.90 | 10.00 | 10.00 | 10.00 | 10.77 | 32.66 | 68.87 | 88.74 | 90.77 | 90.55 |
| c=0.5 | 90.86 | 85.40 | 86.59 | 88.27 | 89.67 | 90.67 | 90.93 | 90.91 | 90.81 | 90.59 |
| c=0.6 | 90.97 | 81.35 | 85.18 | 87.21 | 89.32 | 90.32 | 90.79 | 90.70 | 90.52 | 90.42 |
| c=0.7 | 92.15 | 75.68 | 82.96 | 86.52 | 89.64 | 91.68 | 92.05 | 92.10 | 91.94 | 91.90 |
| c=0.8 | 93.51 | 10.00 | 10.19 | 15.76 | 59.78 | 91.17 | 93.11 | 93.40 | 93.45 | 93.35 |
| c=0.9 | 94.93 | 10.00 | 10.00 | 8.95 | 20.52 | 80.80 | 92.47 | 94.24 | 94.76 | 94.90 |
| | | | | **ResNet-18 on CIFAR-10** | | | | | | |
| c=0.1 | 89.27 | 10.00 | 10.00 | 10.66 | 26.07 | 75.84 | 89.25 | 89.92 | 89.75 | 89.60 |
| c=0.2 | 93.36 | 92.47 | 92.68 | 93.17 | 93.74 | 93.86 | 93.82 | 93.82 | 93.81 | 93.73 |
| c=0.3 | 94.09 | 92.86 | 93.35 | 94.06 | 94.37 | 94.47 | 94.48 | 94.42 | 94.29 | 94.27 |
| c=0.4 | 94.61 | 93.11 | 93.97 | 94.59 | 94.92 | 95.18 | 95.07 | 94.81 | 94.71 | 94.67 |
| c=0.5 | 94.79 | 68.75 | 10.48 | 10.14 | 47.34 | 89.64 | 93.96 | 94.67 | 94.55 | 94.49 |
| c=0.6 | 94.99 | 87.49 | 88.80 | 89.29 | 90.87 | 92.91 | 94.37 | 94.83 | 94.73 | 94.71 |
| c=0.7 | 95.39 | 45.02 | 50.13 | 60.06 | 80.77 | 91.06 | 94.72 | 95.24 | 95.27 | 95.17 |
| c=0.8 | 95.92 | 10.00 | 10.00 | 10.00 | 41.71 | 92.91 | 94.93 | 95.54 | 95.70 | 95.71 |
| c=0.9 | 96.28 | 9.99 | 10.02 | 19.72 | 59.28 | 78.32 | 90.47 | 94.63 | 95.80 | 95.98 |
| | | | | **ResNet-20 on CIFAR-10** | | | | | | |
| c=0.1 | 81.53 | 80.65 | 81.87 | 83.06 | 83.68 | 84.11 | 84.14 | 83.88 | 83.78 | 83.75 |
| c=0.2 | 82.07 | 80.99 | 82.25 | 83.52 | 84.46 | 84.70 | 84.85 | 84.89 | 84.80 | 84.69 |
| c=0.3 | 83.46 | 80.03 | 82.17 | 83.81 | 84.84 | 85.32 | 85.26 | 85.22 | 85.01 | 84.95 |
| c=0.4 | 84.97 | 80.30 | 82.80 | 84.69 | 86.12 | 86.81 | 86.79 | 86.79 | 86.75 | 86.71 |
| c=0.5 | 86.49 | 79.06 | 82.53 | 85.36 | 87.06 | 87.93 | 88.13 | 88.02 | 87.87 | 87.77 |
| c=0.6 | 88.48 | 76.21 | 81.74 | 85.86 | 88.30 | 89.19 | 89.11 | 88.99 | 88.93 | 88.85 |
| c=0.7 | 89.70 | 16.04 | 13.97 | 24.14 | 64.91 | 84.75 | 87.43 | 87.86 | 88.06 | 88.08 |
| c=0.8 | 91.07 | 40.97 | 44.14 | 43.90 | 55.70 | 73.42 | 84.29 | 88.08 | 89.52 | 89.93 |
| c=0.9 | 92.98 | 33.81 | 43.71 | 59.40 | 78.30 | 88.60 | 91.09 | 91.66 | 91.78 | 91.83 |
| | | | | **VGG-16 on CIFAR-100** | | | | | | |
| c=0.2 | 65.04 | 1.00 | 1.00 | 1.01 | 1.53 | 4.57 | 30.82 | 58.80 | 65.33 | 65.13 |
| c=0.3 | 66.05 | 1.00 | 1.00 | 1.00 | 1.00 | 2.15 | 18.58 | 54.26 | 64.34 | 65.96 |
| c=0.4 | 68.46 | 64.21 | 66.30 | 67.97 | 69.31 | 70.09 | 70.19 | 70.05 | 69.79 | 69.62 |
| c=0.5 | 69.30 | 61.99 | 64.31 | 66.71 | 68.91 | 70.42 | 70.50 | 70.18 | 70.03 | 69.85 |
| c=0.6 | 69.49 | 49.34 | 53.22 | 57.83 | 62.58 | 66.67 | 69.11 | 70.07 | 69.63 | 68.96 |
| c=0.7 | 70.97 | 30.19 | 34.77 | 41.37 | 50.01 | 59.17 | 66.61 | 70.07 | 70.85 | 70.45 |
| c=0.8 | 72.13 | 12.81 | 15.68 | 22.37 | 32.70 | 47.78 | 62.35 | 69.54 | 71.31 | 71.11 |
| c=0.9 | 74.76 | 1.00 | 1.00 | 1.03 | 2.02 | 9.97 | 32.00 | 55.97 | 67.82 | 71.85 |
| | | | | **ResNet-18 on CIFAR-100** | | | | | | |
| c=0.1 | 71.84 | 71.11 | 72.51 | 73.32 | 73.41 | 73.38 | 72.63 | 72.19 | 72.06 | 71.88 |
| c=0.2 | 72.32 | 34.00 | 39.42 | 48.16 | 59.34 | 67.41 | 70.63 | 70.14 | 67.63 | 64.74 |
| c=0.3 | 74.01 | 71.51 | 73.91 | 74.89 | 75.40 | 75.41 | 75.30 | 74.98 | 74.90 | 74.71 |
| c=0.4 | 73.90 | 51.56 | 55.56 | 60.20 | 64.74 | 69.16 | 71.99 | 72.89 | 72.76 | 71.94 |
| c=0.5 | 74.88 | 53.01 | 55.92 | 57.37 | 60.59 | 67.62 | 73.15 | 74.53 | 73.70 | 72.81 |
| c=0.6 | 75.93 | 4.45 | 1.01 | 1.01 | 2.05 | 4.33 | 44.28 | 69.24 | 72.85 | 71.71 |
| c=0.7 | 76.44 | 1.13 | 1.00 | 1.00 | 1.00 | 1.66 | 34.47 | 66.96 | 72.57 | 73.35 |
| c=0.8 | 78.41 | 1.00 | 1.00 | 1.00 | 1.00 | 2.31 | 37.50 | 62.62 | 71.34 | 73.83 |
| c=0.9 | 78.18 | 1.00 | 1.00 | 1.04 | 30.44 | 66.73 | 73.81 | 77.04 | 77.52 | 77.66 |
| | | | | **ResNet-20 on CIFAR-100** | | | | | | |
| c=0.1 | 48.62 | 46.80 | 49.85 | 51.61 | 52.19 | 51.95 | 51.23 | 50.31 | 49.56 | 49.12 |
| c=0.2 | 50.79 | 48.12 | 51.35 | 53.27 | 54.17 | 53.91 | 53.11 | 51.75 | 50.89 | 50.35 |
| c=0.3 | 52.84 | 47.08 | 51.34 | 54.51 | 56.00 | 56.31 | 55.46 | 54.46 | 53.82 | 53.42 |
| c=0.4 | 55.18 | 45.58 | 50.63 | 54.72 | 57.44 | 57.67 | 56.69 | 55.38 | 54.54 | 53.97 |
| c=0.5 | 57.51 | 40.65 | 47.14 | 54.15 | 58.37 | 59.59 | 58.47 | 57.33 | 56.41 | 55.88 |
| c=0.6 | 59.98 | 25.56 | 34.28 | 47.01 | 56.64 | 59.60 | 59.16 | 57.74 | 56.73 | 56.35 |
| c=0.7 | 64.71 | 18.87 | 25.93 | 37.26 | 46.92 | 51.28 | 51.68 | 51.52 | 51.12 | 50.84 |
| c=0.8 | 66.96 | 9.73 | 12.76 | 21.48 | 39.48 | 46.84 | 48.91 | 49.90 | 50.01 | 50.18 |
| c=0.9 | 69.36 | 5.82 | 7.25 | 11.01 | 22.58 | 47.32 | 61.57 | 65.26 | 65.96 | 66.32 |

Table S6: Influence of different slope $c$ with the quasi-latency $N = 2$.

| Slope $c$ | ANN Acc. | T=1 | T=2 | T=4 | T=8 | T=16 | T=32 | T=64 | T=128 | T=256 |
|---|---|---|---|---|---|---|---|---|---|---|
| **VGG-16 on CIFAR-10** | | | | | | | | | | |
| c=0.1 | 92.73 | 87.94 | 89.45 | 90.91 | 92.06 | 92.72 | 93.06 | 93.00 | 93.01 | 93.02 |
| c=0.2 | 93.02 | 88.17 | 89.57 | 91.08 | 92.26 | 92.96 | 93.19 | 93.25 | 93.24 | 93.25 |
| c=0.3 | 93.11 | 86.81 | 88.60 | 90.19 | 91.83 | 92.96 | 93.25 | 93.41 | 93.40 | 93.40 |
| c=0.4 | 93.10 | 83.57 | 86.65 | 89.47 | 91.48 | 92.81 | 93.08 | 93.18 | 93.18 | 93.12 |
| c=0.5 | 92.84 | 75.46 | 81.96 | 86.56 | 90.51 | 92.19 | 92.53 | 92.62 | 92.56 | 92.48 |
| c=0.6 | 93.44 | 73.83 | 83.96 | 88.69 | 91.73 | 92.79 | 93.39 | 93.36 | 93.42 | 93.41 |
| c=0.7 | 94.31 | 72.17 | 80.16 | 86.63 | 91.38 | 93.52 | 94.11 | 94.41 | 94.47 | 94.54 |
| c=0.8 | 94.93 | 61.41 | 81.37 | 89.14 | 92.70 | 94.32 | 94.80 | 94.95 | 94.91 | 94.93 |
| c=0.9 | 95.47 | 59.55 | 80.19 | 89.92 | 93.77 | 95.13 | 95.42 | 95.39 | 95.44 | 95.42 |
| **ResNet-18 on CIFAR-10** | | | | | | | | | | |
| c=0.1 | 94.20 | 89.30 | 91.26 | 92.61 | 93.76 | 94.19 | 94.42 | 94.42 | 94.43 | 94.50 |
| c=0.2 | 95.16 | 90.79 | 92.68 | 94.11 | 95.08 | 95.28 | 95.41 | 95.37 | 95.39 | 95.35 |
| c=0.3 | 95.42 | 89.97 | 92.13 | 93.90 | 95.14 | 95.68 | 95.82 | 95.70 | 95.74 | 95.69 |
| c=0.4 | 95.56 | 90.37 | 92.32 | 93.85 | 94.96 | 95.62 | 95.68 | 95.71 | 95.75 | 95.78 |
| c=0.5 | 95.97 | 90.63 | 92.77 | 94.36 | 95.44 | 96.07 | 96.14 | 96.15 | 96.11 | 96.10 |
| c=0.6 | 95.98 | 86.23 | 90.08 | 93.02 | 94.96 | 95.81 | 96.10 | 96.14 | 96.12 | 96.15 |
| c=0.7 | 96.06 | 85.96 | 89.72 | 92.81 | 94.81 | 95.60 | 95.93 | 95.95 | 96.10 | 96.14 |
| c=0.8 | 96.46 | 82.69 | 88.09 | 92.04 | 94.81 | 95.99 | 96.29 | 96.39 | 96.31 | 96.29 |
| c=0.9 | 96.48 | 68.90 | 77.39 | 86.43 | 92.76 | 95.38 | 96.15 | 96.36 | 96.45 | 96.48 |
| **ResNet-20 on CIFAR-10** | | | | | | | | | | |
| c=0.1 | 87.91 | 75.92 | 80.90 | 85.29 | 88.14 | 89.10 | 89.35 | 89.19 | 89.01 | 88.95 |
| c=0.2 | 88.66 | 75.06 | 80.65 | 86.17 | 88.65 | 89.51 | 89.90 | 89.83 | 89.69 | 89.61 |
| c=0.3 | 89.53 | 71.77 | 78.42 | 84.76 | 88.82 | 90.24 | 90.45 | 90.37 | 90.20 | 90.15 |
| c=0.4 | 89.73 | 68.57 | 76.85 | 84.33 | 88.71 | 90.05 | 90.14 | 90.20 | 90.25 | 90.20 |
| c=0.5 | 90.72 | 66.82 | 76.31 | 84.58 | 89.57 | 91.13 | 91.46 | 91.45 | 91.35 | 91.32 |
| c=0.6 | 91.48 | 56.46 | 67.99 | 80.94 | 88.52 | 90.99 | 91.77 | 91.89 | 91.84 | 91.88 |
| c=0.7 | 92.17 | 60.32 | 70.93 | 81.52 | 88.88 | 91.72 | 92.26 | 92.26 | 92.25 | 92.27 |
| c=0.8 | 92.91 | 50.95 | 60.84 | 74.04 | 86.55 | 91.83 | 93.14 | 93.40 | 93.35 | 93.26 |
| c=0.9 | 93.11 | 35.47 | 46.10 | 62.81 | 82.57 | 90.93 | 92.71 | 93.14 | 93.21 | 93.18 |
| **VGG-16 on CIFAR-100** | | | | | | | | | | |
| c=0.1 | 70.03 | 54.68 | 58.66 | 62.56 | 66.31 | 69.35 | 70.65 | 71.23 | 71.52 | 71.47 |
| c=0.2 | 70.73 | 48.02 | 52.87 | 58.53 | 64.34 | 68.35 | 70.66 | 71.52 | 71.79 | 71.76 |
| c=0.3 | 71.16 | 48.14 | 53.15 | 58.71 | 64.57 | 68.66 | 70.93 | 71.83 | 72.00 | 71.98 |
| c=0.4 | 71.43 | 42.45 | 48.41 | 55.32 | 62.68 | 68.34 | 70.84 | 71.88 | 72.17 | 72.07 |
| c=0.5 | 72.66 | 36.01 | 43.11 | 51.25 | 59.92 | 67.10 | 70.95 | 72.48 | 72.91 | 73.15 |
| c=0.6 | 72.73 | 27.72 | 33.97 | 42.64 | 53.60 | 63.91 | 70.07 | 72.61 | 73.26 | 73.35 |
| c=0.7 | 73.47 | 19.30 | 25.04 | 33.69 | 46.26 | 60.29 | 69.07 | 72.51 | 73.46 | 73.46 |
| c=0.8 | 74.12 | 13.40 | 17.44 | 25.41 | 39.23 | 56.54 | 68.80 | 73.11 | 74.18 | 74.41 |
| c=0.9 | 75.18 | 22.41 | 28.52 | 38.27 | 51.58 | 64.70 | 71.73 | 74.32 | 75.14 | 75.25 |
| **ResNet-18 on CIFAR-100** | | | | | | | | | | |
| c=0.1 | 75.38 | 61.23 | 67.17 | 71.52 | 74.64 | 76.20 | 76.50 | 76.46 | 76.29 | 76.30 |
| c=0.2 | 76.15 | 61.14 | 67.49 | 72.24 | 75.16 | 76.66 | 77.04 | 76.96 | 76.95 | 76.94 |
| c=0.3 | 76.60 | 58.29 | 65.68 | 71.51 | 75.28 | 76.96 | 77.07 | 76.99 | 77.00 | 76.99 |
| c=0.4 | 77.32 | 55.72 | 62.98 | 70.09 | 74.72 | 77.02 | 77.99 | 77.98 | 77.82 | 77.79 |
| c=0.5 | 77.08 | 51.01 | 60.03 | 68.72 | 74.59 | 77.29 | 78.04 | 77.97 | 77.99 | 77.91 |
| c=0.6 | 77.42 | 41.69 | 53.51 | 64.96 | 73.17 | 76.90 | 77.57 | 77.68 | 77.85 | 77.80 |
| c=0.7 | 77.93 | 33.40 | 45.77 | 58.96 | 70.54 | 76.15 | 77.53 | 78.02 | 78.02 | 78.08 |
| c=0.8 | 78.22 | 1.00 | 18.51 | 2.38 | 6.82 | 43.49 | 74.16 | 78.05 | 78.59 | 78.64 |
| c=0.9 | 78.22 | 1.00 | 19.97 | 1.71 | 17.25 | 65.52 | 76.47 | 78.10 | 78.26 | 78.23 |
| **ResNet-20 on CIFAR-100** | | | | | | | | | | |
| c=0.1 | 59.83 | 32.76 | 42.03 | 52.20 | 59.45 | 62.15 | 62.47 | 62.81 | 62.65 | 62.40 |
| c=0.2 | 61.36 | 31.66 | 40.76 | 52.05 | 60.08 | 63.17 | 63.63 | 63.42 | 63.07 | 62.90 |
| c=0.3 | 62.96 | 23.91 | 32.65 | 45.70 | 58.25 | 63.62 | 64.90 | 64.92 | 64.80 | 64.59 |
| c=0.4 | 64.32 | 24.88 | 34.52 | 48.19 | 60.47 | 65.14 | 66.55 | 66.72 | 66.46 | 66.41 |
| c=0.5 | 65.85 | 18.07 | 25.30 | 39.24 | 56.10 | 64.30 | 66.50 | 67.12 | 67.15 | 67.21 |
| c=0.6 | 66.75 | 16.58 | 23.81 | 37.84 | 56.00 | 64.97 | 67.53 | 68.09 | 67.85 | 67.71 |
| c=0.7 | 68.49 | 10.58 | 15.74 | 28.33 | 49.67 | 63.67 | 67.64 | 68.77 | 68.98 | 69.03 |
| c=0.8 | 69.03 | 13.51 | 20.17 | 33.17 | 53.63 | 65.21 | 68.59 | 69.32 | 69.51 | 69.45 |
| c=0.9 | 69.70 | 7.92 | 11.19 | 18.76 | 36.40 | 59.36 | 67.75 | 69.62 | 69.89 | 70.02 |

Table S7: Influence of different slope $c$ with the quasi-latency $N = 4$.

| Slope $c$ | ANN Acc. | T=1 | T=2 | T=4 | T=8 | T=16 | T=32 | T=64 | T=128 | T=256 |
|---|---|---|---|---|---|---|---|---|---|---|
| | | | | | **VGG-16 on CIFAR-10** | | | | | |
| c=0.1 | 93.24 | 68.87 | 83.64 | 89.31 | 92.14 | 93.03 | 93.32 | 93.41 | 93.47 | 93.47 |
| c=0.2 | 92.68 | 60.15 | 80.03 | 86.37 | 90.43 | 92.17 | 92.62 | 92.71 | 92.75 | 92.71 |
| c=0.3 | 93.55 | 47.52 | 83.30 | 88.85 | 92.07 | 93.18 | 93.54 | 93.65 | 93.68 | 93.67 |
| c=0.4 | 93.94 | 51.52 | 80.67 | 88.13 | 91.89 | 93.20 | 93.81 | 93.98 | 94.03 | 94.01 |
| c=0.5 | 94.54 | 65.47 | 83.46 | 90.13 | 93.09 | 94.25 | 94.61 | 94.62 | 94.63 | 94.58 |
| c=0.6 | 94.95 | 50.73 | 83.07 | 90.25 | 93.37 | 94.61 | 94.91 | 95.06 | 95.05 | 95.02 |
| c=0.7 | 95.02 | 32.67 | 79.63 | 89.69 | 93.55 | 94.84 | 95.14 | 95.04 | 95.02 | 95.05 |
| c=0.8 | 95.52 | 21.08 | 76.27 | 89.69 | 93.73 | 94.98 | 95.47 | 95.53 | 95.61 | 95.60 |
| c=0.9 | 95.60 | 11.37 | 75.18 | 88.80 | 93.54 | 95.20 | 95.66 | 95.65 | 95.66 | 95.67 |
| | | | | | **ResNet-18 on CIFAR-10** | | | | | |
| c=0.1 | 96.01 | 88.01 | 90.96 | 93.34 | 95.12 | 95.86 | 96.02 | 96.13 | 96.16 | 96.16 |
| c=0.2 | 96.31 | 86.16 | 89.82 | 93.00 | 95.02 | 95.90 | 96.27 | 96.43 | 96.44 | 96.45 |
| c=0.3 | 96.15 | 86.52 | 90.78 | 93.84 | 95.48 | 96.10 | 96.12 | 96.22 | 96.15 | 96.19 |
| c=0.4 | 96.27 | 87.18 | 90.76 | 93.66 | 95.29 | 95.90 | 96.13 | 96.25 | 96.21 | 96.25 |
| c=0.5 | 96.38 | 84.76 | 89.29 | 92.89 | 94.99 | 95.82 | 96.27 | 96.33 | 96.36 | 96.36 |
| c=0.6 | 96.29 | 79.25 | 85.25 | 90.26 | 94.05 | 95.68 | 96.30 | 96.39 | 96.42 | 96.41 |
| c=0.7 | 96.68 | 74.78 | 82.30 | 89.16 | 93.86 | 95.89 | 96.46 | 96.60 | 96.66 | 96.69 |
| c=0.8 | 96.53 | 73.43 | 80.72 | 88.15 | 93.28 | 95.70 | 96.23 | 96.45 | 96.55 | 96.58 |
| c=0.9 | 96.67 | 56.79 | 68.00 | 81.08 | 90.61 | 95.08 | 96.31 | 96.53 | 96.52 | 96.59 |
| | | | | | **ResNet-20 on CIFAR-10** | | | | | |
| c=0.1 | 91.42 | 66.51 | 75.99 | 84.62 | 89.58 | 91.24 | 91.80 | 91.89 | 91.97 | 92.01 |
| c=0.2 | 91.82 | 60.30 | 71.25 | 82.44 | 89.05 | 91.79 | 92.27 | 92.36 | 92.35 | 92.28 |
| c=0.3 | 91.81 | 60.66 | 72.13 | 82.62 | 89.40 | 91.74 | 92.36 | 92.46 | 92.53 | 92.51 |
| c=0.4 | 92.07 | 61.96 | 72.57 | 82.27 | 88.68 | 91.45 | 92.38 | 92.46 | 92.55 | 92.55 |
| c=0.5 | 92.91 | 44.08 | 54.50 | 71.27 | 86.16 | 91.66 | 93.14 | 93.24 | 93.32 | 93.21 |
| c=0.6 | 92.96 | 45.87 | 57.82 | 73.17 | 86.66 | 92.13 | 93.23 | 93.36 | 93.29 | 93.19 |
| c=0.7 | 93.30 | 41.65 | 54.13 | 70.79 | 85.97 | 91.79 | 93.28 | 93.61 | 93.55 | 93.46 |
| c=0.8 | 93.33 | 29.14 | 39.35 | 59.35 | 80.90 | 90.65 | 92.76 | 93.14 | 93.26 | 93.24 |
| c=0.9 | 93.37 | 15.29 | 21.55 | 41.27 | 75.60 | 90.45 | 92.95 | 93.49 | 93.52 | 93.52 |
| | | | | | **VGG-16 on CIFAR-100** | | | | | |
| c=0.1 | 71.78 | 22.27 | 28.83 | 38.88 | 51.68 | 63.38 | 69.68 | 71.64 | 72.04 | 71.92 |
| c=0.2 | 72.16 | 20.01 | 26.24 | 35.41 | 47.79 | 60.82 | 68.78 | 71.67 | 72.47 | 72.59 |
| c=0.3 | 73.40 | 26.37 | 33.29 | 42.97 | 55.27 | 65.80 | 71.30 | 73.26 | 73.55 | 73.69 |
| c=0.4 | 73.18 | 18.13 | 24.70 | 34.32 | 47.98 | 61.11 | 69.59 | 72.83 | 73.65 | 73.51 |
| c=0.5 | 73.25 | 15.29 | 20.91 | 30.29 | 43.65 | 58.57 | 68.41 | 72.33 | 73.28 | 73.53 |
| c=0.6 | 74.26 | 18.37 | 24.09 | 32.89 | 46.76 | 61.41 | 70.50 | 73.83 | 74.42 | 74.50 |
| c=0.7 | 74.94 | 19.95 | 26.09 | 35.86 | 49.68 | 63.12 | 71.18 | 74.05 | 74.91 | 75.03 |
| c=0.8 | 74.50 | 9.07 | 13.70 | 22.44 | 37.92 | 57.41 | 69.20 | 73.02 | 74.33 | 74.69 |
| c=0.9 | 75.25 | 13.62 | 21.23 | 32.24 | 47.76 | 63.32 | 71.84 | 74.61 | 75.13 | 75.15 |
| | | | | | **ResNet-18 on CIFAR-100** | | | | | |
| c=0.1 | 76.71 | 46.54 | 56.14 | 66.28 | 73.07 | 75.93 | 76.72 | 77.11 | 77.20 | 77.15 |
| c=0.2 | 77.82 | 45.71 | 55.72 | 66.18 | 74.02 | 77.19 | 77.96 | 78.15 | 78.21 | 78.25 |
| c=0.3 | 77.85 | 42.74 | 53.62 | 64.77 | 73.37 | 76.95 | 78.06 | 78.26 | 78.26 | 78.26 |
| c=0.4 | 78.28 | 44.72 | 55.01 | 65.50 | 73.58 | 77.36 | 78.61 | 78.54 | 78.76 | 78.78 |
| c=0.5 | 77.69 | 38.73 | 50.81 | 63.20 | 72.84 | 76.69 | 77.95 | 77.94 | 77.77 | 77.74 |
| c=0.6 | 78.30 | 29.83 | 40.41 | 55.37 | 69.27 | 76.07 | 77.94 | 78.66 | 78.61 | 78.59 |
| c=0.7 | 78.56 | 25.33 | 35.45 | 51.41 | 68.22 | 75.46 | 77.79 | 78.24 | 78.55 | 78.75 |
| c=0.8 | 77.96 | 21.54 | 30.90 | 45.51 | 64.42 | 73.94 | 77.14 | 78.16 | 78.34 | 78.33 |
| c=0.9 | 78.00 | 13.54 | 20.76 | 33.67 | 57.43 | 72.09 | 76.43 | 77.60 | 77.98 | 78.14 |
| | | | | | **ResNet-20 on CIFAR-100** | | | | | |
| c=0.1 | 66.37 | 15.69 | 23.85 | 40.99 | 58.23 | 65.42 | 67.39 | 67.68 | 67.51 | 67.33 |
| c=0.2 | 66.91 | 19.33 | 27.89 | 43.62 | 59.79 | 66.51 | 68.16 | 68.40 | 68.42 | 68.45 |
| c=0.3 | 67.39 | 15.92 | 22.44 | 38.25 | 57.74 | 66.47 | 68.29 | 68.70 | 68.59 | 68.45 |
| c=0.4 | 68.40 | 16.52 | 23.79 | 37.94 | 57.20 | 66.61 | 68.76 | 69.04 | 69.09 | 68.96 |
| c=0.5 | 68.86 | 11.14 | 15.27 | 26.79 | 49.58 | 64.77 | 68.70 | 69.63 | 69.75 | 69.69 |
| c=0.6 | 68.83 | 9.15 | 12.99 | 23.57 | 45.74 | 63.08 | 68.25 | 69.08 | 69.24 | 69.32 |
| c=0.7 | 69.45 | 6.90 | 10.56 | 19.27 | 40.90 | 62.26 | 68.71 | 70.06 | 70.29 | 70.13 |
| c=0.8 | 69.59 | 7.09 | 9.00 | 14.88 | 32.93 | 59.35 | 68.05 | 69.61 | 70.08 | 69.94 |
| c=0.9 | 70.18 | 4.79 | 7.11 | 12.04 | 25.98 | 53.20 | 66.77 | 69.95 | 70.54 | 70.46 |

