# OpenReview forum: "A unified optimization framework of ANN-SNN Conversion: towards optimal mapping from activation values to firing rates"
_ICLR.cc/2023/Conference — Submitted to ICLR 2023_

### Official Review · Reviewer_FPAL · 2022-10-13

**Confidence:** 4
**Correctness:** 3
**Technical Novelty And Significance:** 4
**Empirical Novelty And Significance:** 4
**Recommendation:** 8

**Clarity, Quality, Novelty And Reproducibility:**

Clarity: 6/10

Quality: 8/10

Novelty: 8/10

Reproducibility: 5/10

**Strength And Weaknesses:**

Strengths:
1. The tackled problem is relevant to the scope of ICLR.
2. The contributions look solid.
3. The results look promising.

Weaknesses:
1. It would be interesting to have more details about more intuitions about the design decision made when developing the proposed SlipReLU activation function.
2. The description of the proposed method in Section 4 looks vague and hard to follow. It is recommended to use schemes, and examples to ease the discussion.
3. Please describe the experimental setup and tool flow used to generate the results shown in Section 6.
4. It would be useful to provide the source code for reviewers' inspection during the rebuttal.


**Summary Of The Paper:**

The paper proposes an efficient ANN-SNN conversion mechanism based on the SlipReLU activation function. It replaces the traditional ReLU to improve the accuracy of the converted SNN. The results look promising and competitive compared to the related works.

**Summary Of The Review:**

Good paper where few concerns should be clarified.

---

> ### Author Response · Authors · 2022-11-17
> **Response to Reviewer FPAL**
>
> Q1: More intuitions about the design decision made when developing the proposed SlipReLU activation function
>
> A1: Here are the intuitions about the design decision made when developing the proposed SlipReLU activation function.
> - As we study the two-step ANN-SNN conversion method in this paper, the existing ANN-SNN conversion methods usually redesign the ANN with a new activation function instead of the regular ReLU, train the tailored ANN and convert it to an SNN.
> - A tailored ANN that deviates too much from the regular ANN will degrade its performance, resulting in a performance loss which will be inherited to the
> converted SNN.
> - **The performance loss between the regular ANN with ReLU and the tailored ANN has never been considered**, which will be inherited to the converted SNN.
> - Therefore, **we consider the performance loss between the regular ANN with ReLU and the tailored ANN which has never been considered in ANN-SNN conversion methods.**
> - And we formulate the ANN-SNN conversion as a unified optimization problem which considers the ANN performance as well as the conversion error simultaneously.
> - **We propose to use the SlipReLU activation function in the tailored ANN,** in order to minimize the layer-wise conversion error and keep tailored ANN performance as good as the regular ANN with regular ReLU activation function.
>
>
> Q2: Experimental setups
>
> A2: We have updated the details of experimental setups in **Appendix E** to make the paper reproducible.
>
>
> Q3: The source code
>
> A3: The source code can be found in the appendix.

---

> > ### Comment · Reviewer_FPAL · 2022-11-21
> > **Response to Authors**
> >
> > The efforts made by the authors in answering the reviewers' comments are appreciated.

---

> > > ### Author Response · Authors · 2022-11-23
> > > **Thanks for Reviewer FPAL**
> > >
> > > We greatly appreciate the time you spent on our responses and your valuable suggestions for adding experimental results. We will add experiments in the final version to further support the conclusion that our proposed SlipReLU is better than the baselines.

---

### Official Review · Reviewer_rSjR · 2022-10-13

**Confidence:** 5
**Correctness:** 3
**Technical Novelty And Significance:** 3
**Empirical Novelty And Significance:** 2
**Recommendation:** 6

**Clarity, Quality, Novelty And Reproducibility:**

+ Clarity: This paper is sufficiently clear.

+ Quality: This paper's quality is good.

+ Novelty: An extension from previous work, but it needs more exresults to support its correctness.

+ Reproducibility: Not reproducible at this time.

**Strength And Weaknesses:**

Strength:

+ The structure and presentation of this paper are neat and easy to understand.

+ The perspective of the trade-off between the distance to ANN and the distance to SNN is novel and an interesting activation function is proposed (SlipReLU).

Weakness:

- The major concern in this work is the first term in Eq. 5, which hypothesize that the output between ReLU-based ANN and customized ANN should be considered, is not fully verified in experiments. As far as I interpret Eq. 5, the SlipReLU should have higher ANN accuracy but also higher conversion error than QCFS, because it has less distance to ANN while more distance to SNN compared with QCFS.
And ideal result in SlipReLU is that it shows higher ANN accuracy and despite the higher conversion error, SlipReLU can still achieve higher SNN accuracy than QCFS. However, the results in paper does not verify the tradeoff or did not explicitly discuss the tradeoff in experiments. Ablation studies on the weight of combination, and more analysis on ANN, SNN accuracy tradeoff should be provided.

- Two-step conversion certainly has its pros, but it also has cons. Two step method must train their own ANN models, while one-step models can utilize directly-trained checkpoints in open-sourced projects. Some large-scale datasets and models, are require a lot resource to train the ANN, for example, ImageNet. Thus, ImageNet results are also anticipated.





**Summary Of The Paper:**

In this paper, the authors devise a two-step conversion framework for ANN-SNN conversion. Unlike previous methods which consider the ANN-SNN conversion error, this work considers the error between source ANN and altered ANN as well as altered ANN with SNN. The idea is generally sound and simple to understand. Yet in my view, the two-step conversion naturally has some deficiencies with than one-step conversion (see weakness section for details).




**Summary Of The Review:**

My reasons for the weak rejection are insufficient analysis and the lack of large-scale experiments. Were they addressed by the authors, I can increase my score.

---

> ### Author Response · Authors · 2022-11-17
> **Response to Reviewer rSjR**
>
>
> **Q1: Experiments to interpret Eq. (5)**
>
> A1: We have extended the experimental results section in the main paper, which can be found in **Table 1 in Section 6**, where we **explicitly** discuss the tradeoff in experiments.
> - From Eq.(5), the SlipReLU should have higher ANN accuracy than QCFS. From Eq.(5), despite the conversion error, SlipReLU can still achieve higher SNN accuracy than QCFS.
>
> **For the ANN accuracy**
> - From **Table 1**, the results with SlipReLU activation shows that it has higher ANN accuracy than the QCFS (step function), where the ANN accuracy of ReLU is the baseline.
> - The reason is that QCFS only considers the conversion error but not the ANN performance, while our SlipReLU proposes to consider the conversion error as well as the ANN performance.
>
> **For the SNN accuracy**
>
> - For ultra-low latency inference (T = 1 or T = 2), our proposed SlipReLU has the best performance compared to existing state-of-the-art ANN-SNN conversion methods.
> - Specially, when the latency T = 1, our SlipReLU method is able to achieve an
> accuracy of **93.11%** for ResNet-18, with a good margin compared to the next best baseline QCFS (88.30%).
> - The accuracy for VGG-16 is **85.40%** with SlipReLU activation, while the next best accuracy is 75.51% with QCFS activation.
> - For ResNet-20, we achieve an accuracy of **82.8%** with 2 time-steps.
>
> In, summary, **our proposed SlipReLU method indeed has higher ANN accuracy than the QCFS (step function), and it gives the best SNN accuracy for ultra-low latency inference,** which coincide with the proposed unified framework.
>
>
> **Q2: Ablation studies on the weight of combination and show the trade-off**
>
> A2: We have conducted ablation studies on the weight of combination and show the trade-off in **Figure 2 in Section 6** of the main paper. Detailed analysis can be found in **Section 6** of the main paper, and **Appendix I**.
>
> - It shows that for small values of quasi-latency N, the slope c has a large effect on SNN accuracy for ultra-low and low-latency inference. In particular, for small quasi-latency N, different slope values c can result in different SNN accuracy when the time-step T is small.
> - But for large values of quasi-latency N, the colored curves are close to each other, and different values of slope c give similar results no matter whether the time-step T is small or large.
> - This brings the flexibility to apply our SlipReLU to different scenarios.
> - When we need ultra-low/low- latency inference for the converted SNN, we choose small quasi-latency N.
> - When we do not care about the inference time (the time-step T can be large), we then choose large quasi-latency N.
> - More detailed results can be found in **Fig. S2, Table S5, Table S6 and Table S7 in Appendix I**.
>
>
> **Q3: Experiment results on large-scale dataset**
>
> A3:
> - To support the capability of our proposed SlipReLU model, we have updated our paper with additional experiment on **CIFAR100 and Tiny-ImageNet datasets** and results can be found in **Table S3 and Table S4 of appendix**.
> - We could not include the results on ImageNet because of computational constraints but we included the results of the ImageNet subset which is Tiny-ImageNet. If necessary we will include experiment results on ImageNet in the final version.
> - From **Table S3**, we see that our SlipReLU method outperforms the others both in terms of high accuracy and ultra-low latency.
> - For VGG16, the accuracy of the proposed method can achieve an accuracy of **64.21\%** which is **29.1\%** higher than QCFS and **39.98\%** higher than SNNC-AP when the time-steps is only 1.
> - For ResNet-18, when T = 1, we can still achieve an accuracy of **71.51\%**.

---

> > ### Comment · Reviewer_rSjR · 2022-11-21
> > **Well-supported results, but Table 1 is strongly misleading.**
> >
> > I have checked the results that the authors provided to me. Overall I find the results provided in Table S5 useful. But the results in Table 1 are totally misleading to readers.
> >
> > For example, in the case of VGG-16 on CIFAR-10 from Table 1, the authors claimed the ANN's accuracy is 95.60, this entry is found in Table S7 $c=0.9, N=4$ which is quite close to ReLU. Yet the conversion accuracy of $T=1$ corresponds to the VGG-16 trained with $c=0.5, N=1$, where the ANN accuracy is only **90.86%**. How can you convert an SNN with 95.66% accuracy when ANN accuracy is only 90.86%?
> >
> > Clearly that the authors are using different ANNs for adapting different $T$. This raises a lot of misinterpretation among the readers. **The authors must report the accuracy of SNNs converted from one single ANN**.

---

> > > ### Author Response · Authors · 2022-11-23
> > > **Thanks to Reviewer rSjR for your inspiring question**
> > >
> > > Dear Reviewer rSjR,
> > >
> > > Thanks for your feedbacks.
> > >
> > > A1: We are sorry for not providing clear enough results in Table 1. We agree that we should report the accuracy of SNNs converted from one single ANN.
> > > - According to **the criterion of the Unified Optimization Framework of ANN-SNN Conversion in Eq.(5) in the main paper**,
> > > $$
> > > \min_{F, T}
> > >      w E_{z} \left(
> > >      | f(z; W, {ReLU}) - f(z; W, T, F_{ ANN} ) | \right) + (1-w) E_{z} \left(
> > >      | f(z; W, F_{ANN} ) - f(z; W, T, F_{SNN} ) | \right) ,
> > > $$
> > > we choose the best hyper-parameters (N, c) based on **Eq.(5)**, which **minimizes the following criterion measure**
> > > $$
> > > \text{ criterion measure } = \frac{1}{2}|ReLU - ANN| + \frac{1}{2} |ANN - SNN| .
> > > $$
> > >
> > > - According to **the criterion measure** or Eq.(5) of the main paper, the best hyper-parameter is (N, c)=(2, 0.2). For the baseline QCFS model, N = 4 is the best. It is fair to choose hyper-parameters based on this criterion rather than a rule-of-thumb. Then we **convert this best single model to SNN** and obtain SNN accuracy under different time-steps T.
> > >
> > > - **The results are as follows (VGG-16 on CIFAR-10)**, which should be in **Table 1 in the main paper**:
> > >
> > > | Model  | ANN acc.    | T=1 | T=2 | T=4 | T=8 | T=16 | T=32 | T=64 | T= 128|
> > > | -------|------------| -------------| -----------| ------------| ------------| -----------| ------------| -----------| ------------|
> > > | QCFS | 92.69 | 75.51 | 83.81 | 88.58 | 91.47 | 92.50 | 92.83 | 92.83 | 92.90 |
> > > | ReLU | 95.92 | 10.00 |10.00 | 11.51 |70.97 | 88.39 |  93.05| 94.76 | 95.19|
> > > | Ours (N=2, c=0.2) | 93.02 | 88.17 | 89.57 | 91.08 | 92.26 | 92.96 | 93.19 | 93.25 | 93.24 |
> > >
> > > - Note that the above result is from one row in **Table S6**. **We will make it clear and provide the details in the final version.**
> > >
> > >
> > >
> > > **A2: The details of selecting the best hyper-parameter are as follows**
> > >
> > > For VGG-16 on CIFAR-10, when N=1, N=2, N=4, we have the following results. **We choose the hyper-parameter with the minimum \|ReLU - ANN\| + \|ANN - SNN\|.**
> > >
> > >
> > > | Slope   | ReLU  | ANN  | SNN   | \|ReLU - ANN\| | \| ANN - SNN \| | \|ReLU - ANN\| + \|ANN - SNN\| |
> > > |----------------|-------|-------|-------|--------------|---------------|-----------------------------|
> > > | 0.1  | 95.92 | 90.93 | 10    | 4.99  | 80.93  | 85.92  |
> > > | 0.2  | 95.92 | 88.68 | 10    | 7.24  | 78.68  | 85.92  |
> > > | 0.3  | 95.92 | 86.73 | 10    | 9.19  | 76.73  | 85.92  |
> > > | 0.4  | 95.92 | 91.9  | 10    | 4.02  | 81.9   | 85.92  |
> > > | 0.5  | 95.92 | 90.86 | 85.4  | 5.06  | 5.46   | 10.52  |
> > > | 0.6  | 95.92 | 90.97 | 81.35 | 4.95  | 9.62   | 14.57  |
> > > | 0.7  | 95.92 | 92.15 | 75.68 | 3.77  | 16.47  | 20.24  |
> > > | 0.8  | 95.92 | 93.51 | 10    | 2.41  | 83.51  | 85.92  |
> > > | 0.9  | 95.92 | 94.93 | 10    | 0.99  | 84.93  | 85.92  |
> > > | N=2  |       |       |       |    |     | |
> > > | Slope   | ReLU  | ANN  | SNN   | \|ReLU - ANN\| | \| ANN - SNN \| | \|ReLU - ANN\| + \|ANN - SNN\| |
> > > | 0.1  | 95.92 | 92.73 | 87.94 | 3.19  | 4.79   | 7.98   |
> > > | 0.2  | 95.92 | 93.02 | 88.17 | 2.9   | 4.85   | **7.75**   |
> > > | 0.3  | 95.92 | 93.11 | 86.81 | 2.81  | 6.3 | 9.11   |
> > > | 0.4  | 95.92 | 93.1  | 83.57 | 2.82  | 9.53   | 12.35  |
> > > | 0.5  | 95.92 | 92.84 | 75.46 | 3.08  | 17.38  | 20.46  |
> > > | 0.6  | 95.92 | 93.44 | 73.83 | 2.48  | 19.61  | 22.09  |
> > > | 0.7  | 95.92 | 94.31 | 72.17 | 1.61  | 22.14  | 23.75  |
> > > | 0.8  | 95.92 | 94.93 | 61.41 | 0.99  | 33.52  | 34.51  |
> > > | 0.9  | 95.92 | 95.47 | 59.55 | 0.45  | 35.92  | 36.37  |
> > > | N=4  |       |       |       |    |     | |
> > > | Slope   | ReLU  | ANN  | SNN   | \|ReLU - ANN\| | \| ANN - SNN \| | \|ReLU - ANN\| + \|ANN - SNN\| |
> > > | 0.1  | 95.92 | 93.24 | 68.87 | 2.68  | 24.37  | 27.05  |
> > > | 0.2  | 95.92 | 92.68 | 60.15 | 3.24  | 32.53  | 35.77  |
> > > | 0.3  | 95.92 | 93.55 | 47.52 | 2.37  | 46.03  | 48.4   |
> > > | 0.4  | 95.92 | 93.94 | 51.52 | 1.98  | 42.42  | 44.4   |
> > > | 0.5  | 95.92 | 94.54 | 65.47 | 1.38  | 29.07  | 30.45  |
> > > | 0.6  | 95.92 | 94.95 | 50.73 | 0.97  | 44.22  | 45.19  |
> > > | 0.7  | 95.92 | 95.02 | 32.67 | 0.9   | 62.35  | 63.25  |
> > > | 0.8  | 95.92 | 95.52 | 21.08 | 0.4   | 74.44  | 74.84  |
> > > | 0.9  | 95.92 | 95.6  | 11.37 | 0.32  | 84.23  | 84.55  |

---

> > > > ### Comment · Reviewer_rSjR · 2022-11-27
> > > > **Appreciate it.**
> > > >
> > > > I'd like to thank the authors for providing this result, which is exactly what I expect in this work. I hope they can demonstrate them in their final version clearly.

---

> > > > > ### Author Response · Authors · 2022-11-27
> > > > > **Thanks for Reviewer rSjR.**
> > > > >
> > > > > We greatly appreciate the time you spent on our responses and your valuable suggestions to polish the paper. We will demonstrate them clearly in the final version.

---

### Official Review · Reviewer_7AYG · 2022-10-24

**Confidence:** 5
**Correctness:** 4
**Technical Novelty And Significance:** 3
**Empirical Novelty And Significance:** 3
**Recommendation:** 8

**Clarity, Quality, Novelty And Reproducibility:**

clarity: 9/10
Quality: 8/10
Novelty: 8/10
Reproducibility: 8/10

**Strength And Weaknesses:**

### Strengths
=============

Well written.

Well motivated.

Result is good and inspiring.

Analysis of the various conversion error is present in details. Some are taken from earlier research though.

### Weakness:
===============

The experimental results section is weak. The authors should provide results on larger datasets to show the efficacy.

The author should do an ablation study with SlipReLU and shifted SlipReLU.

It would be interesting to see how such ReLU performs under various noisy inputs as discussed in [1-2], particularly, whether they can also maintain some inherent robustness with such non-linear functions, under two-step conversion based framework.

Please put some light on to the selection of input type (direct, rate etc.) on the performance with  SlipReLU.

lack of reproducibility and missing details of experimental setup.

[1] Inherent adversarial robustness of deep spiking neural networks: Effects of discrete input encoding and non-linear activations, ECCV 2020.

[2] HIRE-SNN: Harnessing the Inherent Robustness of Energy-Efficient Deep Spiking Neural Networks by Training With Crafted Input Noise, ICCV 2021.

**Summary Of The Paper:**

The paper improves upon the conversion error of two step ANN to SNN conversion based technique to yield sota accuracy.  In particular, it presents an efficient ANN-SNN conversion mechanism based on the SlipReLU and shifted SlipReLU activation function replacing the traditional ReLU to improve the accuracy of the converted SNN. The results look promising and competitive compared to the related works. The theoretical analysis of error strengthens the paper.

**Summary Of The Review:**

Overall, this is a well written well motivated paper that requires further experiments to clarify the impact of the proposed non linearity in SNN paradigm.

Post rebuttal: the contributions are new with detailed additional results, major concerns are addressed with results on further complex datasets.

---

> ### Author Response · Authors · 2022-11-17
> **Response to Reviewer 7AYG**
>
> Q1: Comments on experimental results section and experiment results on large-scale datasets
>
> A1:
> - We have extended the experimental results section in the main paper, which can be found in **Table 1 in Section 6**.
> - For ultra-low latency inference (T = 1 or T = 2), our proposed SlipReLU has the best performance compared to existing state-of-the-art ANN-SNN conversion methods.
> - Specially, when the latency T = 1, our SlipReLU method is able to achieve an
> accuracy of **93.11\%** for ResNet-18. The accuracy for VGG-16 is **85.40\%** with SlipReLU activation.
>
> **Experiment results on large-scale datasets**
> - We have also updated the paper with new results on **CIFAR100 and Tiny-ImageNet datasets** and can be found in the **Table S3 and Table S4 of appendix**.
> - From **Table S3**, we see that our SlipReLU method outperforms the others both in terms of high accuracy and ultra-low latency.
> - For VGG16, the accuracy of the proposed method can achieve an accuracy of **64.21\%** which is **29.1\%** higher than QCFS and **39.98\%** higher than SNNC-AP when the time-steps is only 1.
> - For ResNet-18, when T = 1, we can still achieve an accuracy of **71.51\%**.
>
>
> Q2: Ablation study with SlipReLU and shifted SlipReLU
>
> A2: We have conducted ablation studies with SlipReLU and shifted SlipReLU, which can be found in **Appendix H COMPARISON OF SLIPRELU AND SLIPRELU-SHIFT ACTIVATION**.
> - The accuracy of the converted SNN from ANN with SlipReLU activation (in the first and third columns) first increases or stays flat for time-step T <= 4, and then decreases rapidly with the increase of time-steps. The best performance is still lower than the SlipReLU-shift activation.
> - The non-shifted SlipReLU activation shows no advantage for ultra-low latency inference when T <= 4.
> - In contrast, the accuracy of the converted SNN from ANN with SlipReLU-shift activation increases with the increase of time-step T. It converges to the same
> accuracy when the time-step is larger than 16.
> - The SlipReLU-shift activation shows advantages for ultra-low latency inference when T <= 4.
>
> Q3: Suggestion on performance of various noisy inputs
>
> A3: This is a very good and interesting topic which will be further research work followed by this SlipReLU research.
>
>
> Q4: The selection of input type (direct, rate etc.)
>
> A4: As for the input to the first layer and the output of the last layer of the SNN, we do not employ any spiking mechanism as in paper Li et al. (2021). We directly encode the static image to temporal dynamic spikes as input to the first layer, which can prevent the undesired information loss introduced by the Poisson encoding. For the last layer output, we only integrate the pre-synaptic input and do not fire any spikes.
>
>
> Q5: The details of experimental setups
>
> A5: We have updated the details of experimental setups in **Appendix E** to make the paper reproducible.

---

> > ### Comment · Reviewer_7AYG · 2022-11-21
> > **Thanks for the rebuttal**
> >
> > I appreciate the authors' detailed response. Additional results on ultra low latency and complex datasets are useful in demonstrating the advantage of the proposed non-linearity. I am keeping my score unchanged (a 9 might have been more suitable here).

---

> > > ### Author Response · Authors · 2022-11-23
> > > **Thanks for Reviewer 7AYG**
> > >
> > > We greatly appreciate the time you spent on our responses and your valuable suggestions for adding experimental results. We will add experiments in the final version to further support the conclusion that our proposed SlipReLU is better than the baselines.

---

### Official Review · Reviewer_FzYt · 2022-11-01

**Confidence:** 4
**Clarity, Quality, Novelty And Reproducibility:** Not novel at all!
**Correctness:** 2
**Technical Novelty And Significance:** 1
**Empirical Novelty And Significance:** 1
**Recommendation:** 1

**Strength And Weaknesses:**

-Not novel and results are not good. Today, many works are focused on direct training precisely to reduce the overall timestep count in SNNs that can lead to very expensive timestep computations during inference[1]. The authors work still gives best conversion accuracy at T>100. This is not acceptable anymore. I suggest the authors to go over recent conversion works that have strived to created conversion SNNs at T<10 timesteps [2].

[1] Yin, Ruokai, et al. "SATA: Sparsity-Aware Training Accelerator for Spiking Neural Networks." arXiv preprint arXiv:2204.05422 (2022).

[2] Zheng, Hanle, et al. "Going deeper with directly-trained larger spiking neural networks." Proceedings of the AAAI Conference on Artificial Intelligence. Vol. 35. No. 12. 2021.

**Summary Of The Paper:**

The authors propose a conversion algorithm for deploying SNNs.

**Summary Of The Review:**

In summary, this paper's attempt is worthy to be credited but it really lacks good motivation and novelty.

---

> ### Author Response · Authors · 2022-11-17
> **Response to Reviewer FzYt**
>
> Q1: The novelty of the paper
>
> A1: In **General Response**, we emphasize the novelty of our proposed method.
> - As we study the ANN-SNN conversion method in this paper, the existing ANN-SNN conversion methods usually redesign the ANN with a new activation function instead of the regular ReLU, train the tailored ANN and convert it to an SNN.
> - **The performance loss between the regular ANN with ReLU and the tailored ANN has never been considered**, which will be inherited to the converted SNN.
> - **This is the first work to consider the performance loss between the regular ANN with ReLU and the tailored ANN which has never been considered in ANN-SNN conversion methods.**
>
>
>
>
> Q2: Results of SlipReLU method for time-step T < 10
>
> A2: **We humbly disagree with the reviewer on the comment that "results are not good". We conducted experiments on different datasets to show that our proposed model indeed performs well with low latency which is crucial for SNN models during inference.**
>
> - As shown in **Table 1 of the main paper**, our proposed model perform well in terms of SNN test accuracy compared to different baselines on CIFAR10 dataset.
> - From **Table 1**, our proposed SlipReLU method has the best performance compared to existing state-of-the-art ANN-SNN conversion methods, especially for ultra-low latency inference (T=1 or T=2).
> - Specially, when the latency T=1, our SlipReLU method is able to achieve an accuracy of **93.11\%** for ResNet-18 on CIFAR-10. The accuracy for VGG-16 is **85.40\%** with SlipReLU activation. For ResNet-20, we achieve an accuracy of **82.8\%** with 2 time-steps. Our proposed SlipReLU method indeed gives the best SNN accuracy for ultra-low latency inference.
>
>
> **Q3: Comparison on large-scale datasets**
>
> A4:
> - We have also updated the paper with new results on **CIFAR100 and Tiny-ImageNet datasets** and can be found in the **Table S3 and Table S4 of appendix**.
> - From Table S3, we see that our SlipReLU method outperforms the others both in terms of high accuracy and ultra-low latency.
> -  For VGG16, the accuracy of the proposed method can achieve an accuracy of **64.21\%** which is **29.1\%** higher than QCFS and **39.98\%** higher than SNNC-AP when the time-steps is only 1.
> - For ResNet-18, when T = 1, we can still achieve an accuracy of **71.51\%**.
>
>
> Q4: Many works are focused on direct training precisely to reduce the overall timestep count in SNNs.
>
> **A4: Comparison with the recent state-of-the-art supervised SNN training methods**
> - We agree that a lot of effort has been put into developing models which are focused on direct training of SNNs. But, as the existing platforms such as TensorFlow and PyTorch based on CUDA have limited optimization for SNN training, ANN-SNN conversion methods with good SNN test accuracy for lower inference time can be beneficial.
> - We compared the performance the SlipReLU against state-of-the-art supervised models as suggested by the reviewer.
> - We included those results in the **Table S2** of the appendix, which is the same table in **General Response**.
> - Our approach for CIFARNet has achieved an accuracy of **95.31\%** with time-steps 4, the achieved accuracy is higher than any other supervised trained models for low latency value.
>
>
> These state-of-the-art supervised training methods include Hybrid-Conversion (HC) from [1] Rathi et al. (2020), STBP from [2] Wu et al. (2018), TSSL from [3] Zhang & Li (2020) and GDDP from [4] Zheng et al. (2021), which are back-propagation or hybrid training methods.
>
>
> [1] Nitin Rathi, Gopalakrishnan Srinivasan, Priyadarshini Panda, and Kaushik Roy. "Enabling deep spiking neural networks with hybrid conversion and spike timing dependent backpropagation". arXiv preprint arXiv:2005.01807, 2020.
>
> [2] Yujie Wu, Lei Deng, Guoqi Li, Jun Zhu, and Luping Shi. Spatio-temporal backpropagation for training high-performance spiking neural networks. Frontiers in neuroscience, 12:331, 2018.
>
> [3] Wenrui Zhang and Peng Li. Temporal spike sequence learning via backpropagation for deep spiking neural networks. Advances in Neural Information Processing Systems, 33:12022–12033, 2020.
>
> [4] Hanle Zheng, Yujie Wu, Lei Deng, Yifan Hu, and Guoqi Li. Going deeper with directly-trained larger spiking neural networks. In Proceedings of the AAAI Conference on Artificial Intelligence, volume 35, pp. 11062–11070, 2021.

---

> > ### Comment · Reviewer_FzYt · 2022-11-21
> > **Reviewer Response**
> >
> > Thanks to the authors for providing a detailed response. My concern still remains:
> > For e.g., if one uses direct encoding, SNNs can be trained with from-scratch techniques or a mix of conversion and from-scratch (as authors have pointed out as hybrid) to get SNNs with <5 timesteps or even 1 time step as shown here [r2-r8]. I agree that SNNs take longer to be trained on CUDA platforms, but with low-latency training, the training overhead can be minimized. In fact, the authors in [r7,r8] train with just ~5 timestep SNNs with NAS/LTH for getting sparse SNNs with interesting connectivity for higher efficiency.  The author's optimization strategy seems to work but, I feel authors must look at the works from other folks (see below, note may not be an exhaustive list). I agree that the author's results are good, but the overall methodology has been tried before (using a better ReLU alternative in ANNs to get better conversion accuracy in SNN (for e.g. [r4])). And as a result, I will keep my score unchanged.
> >
> > [r2] Chowdhury, Sayeed Shafayet, Nitin Rathi, and Kaushik Roy. "One timestep is all you need: Training spiking neural networks with ultra low latency." arXiv preprint arXiv:2110.05929 (2021).
> >
> > [r3] S. Deng et al. "Optimal Conversion of Conventional Artificial Neural Networks to Spiking Neural Networks." ICLR 2021
> >
> > [r4] T. Bu et al. "Optimal ANN-SNN Conversion for High-accuracy and Ultra-low-latency Spiking Neural Networks." ICLR 2022
> >
> > [r5] Y. Li et al. "Differentiable Spike: Rethinking Gradient-Descent for Training Spiking Neural Networks." NeurIPS 2021
> >
> > [r6] S. Deng et al. "Temporal Efficient Training of Spiking Neural Network via Gradient Re-weighting." ICLR 2022
> >
> > [r7] Y. Kim et al. "Lottery Ticket Hypothesis for Spiking Neural Networks." ECCV 2022
> >
> > [r8] Y. Kim et al. "Neural architecture search for spiking neural networks." ECCV 2022

---

> > > ### Author Response · Authors · 2022-11-23
> > > **Response to Reviewer FzYt (1/3)**
> > >
> > >
> > > Dear Reviewer FzYt
> > >
> > > Thanks for your feedbacks on our paper.
> > >
> > > **Q1: Concern on (1) training from-scratch techniques and (2) ANN-SNN conversion methods**
> > >
> > > A1: **We totally understand your concern on training from-scratch techniques and ANN-SNN conversion methods.** These are generally the **two different routes to obtain an SNN**, *either train an SNN from scratch* or *obtain an SNN from ANN-SNN conversion*.
> > >
> > > - **We agree that a lot of effort has been put into developing models which are focused on direct training of SNNs**. There are many papers investigating methods for directly training SNNs, and some can achieve good result low-latency training.
> > >
> > > - **The ANN-SNN conversion method can be one option to obtain an SNN**. Meanwhile, due to the fact that the existing platforms such as TensorFlow and PyTorch based on CUDA have limited optimization for SNN training, ANN-SNN conversion methods with good SNN test accuracy for lower inference time can be beneficial.
> > >
> > > - **We have added more directly training SNN methods in the following table as suggested by the reviewer** and have compared the performance of state-of-the-art supervised models. We will include those results in the **Table S2 of the appendix** in the final version.
> > >
> > >
> > > **Comparison with the recent state-of-the-art supervised SNN training methods**.
> > > | Model  | Method     | Architecture | SNN Accuracy | Time-step T |
> > > | -------|------------| -------------| -----------| ------------|
> > > | HC | Hybrid | VGG-16 | 92.03 | 200 |
> > > | STBP | Backprop | CIFARNet | 85.82 | 12 |
> > > | GDDT | Backprop | CIFARNet | 87.35 | 4 |
> > > | TSSL | Backprop | CIFARNet | 88.23 | 5 |
> > > | Ours | ANN-SNN  | VGG-16 | 91.08 | 4 |
> > > | Ours | ANN-SNN  | ResNet-18 | 92.86 | 2 |
> > > | Ours | ANN-SNN  | CIFARNet  | 95.31 | 4 |
> > > | IIR-SNN (r2) | Retraining | VGG16 | 92.96 | 5 |
> > > | TET (r6) | Backprop | ResNet19 | 92.98 | 2 |
> > > | LTH (r7) | Pruning | VGG16 | 90.91 | 5|
> > > | NAS (r8) | Neural Architecture Search | SNASNet | 93.47 | 5|
> > >
> > >
> > > - This table reports the results of our model including the models which are trained **through backpropagation, in hybrid mode, using SNN Pruning, employing Neural Architecture Search, through Retraining**. Our approach for CIFARNet has achieved an accuracy of $95.31\%$ with time-steps 4, the achieved accuracy is higher than any other supervised trained models for low latency value.

---

> > > ### Author Response · Authors · 2022-11-23
> > > **Response to Reviewer FzYt (2/3)**
> > >
> > >
> > > **Q2: Comparison with references [r2-r8].**
> > >
> > > **A2: These references [r2-r8] are different aspects studying Spiking Neural Networks (SNNs). There are many papers conducting new methods on SNNs. These different types of methods are not conflicting but complementary in SNNs.**
> > >
> > > - **The paper (denoted as [r2] in the reviewer's comments) investigates the Retraining method using an SNN trained with T=5 time-steps**.
> > > The Iterative Initialization and **Retraining method for SNNs** (IIR-SNN) in [r2] **starts with an SNN trained with T=5 time-steps**. Then at each stage of latency reduction, the network trained at previous stage with higher timestep is utilized as initialization for subsequent training with lower timestep. This acts as a compression method, as the network is gradually shrunk in the temporal domain. It is not surprising that with the Iterative Initialization and Retraining strategy, IIR-SNN method can finally achieve accuracy of 92.96\% of VGG16 on CIFAR-10 by retraining with T=5 until T=1.
> > >
> > > - **The papers (denoted as [r3-r4] in the reviewer's comments) are ANN-SNN conversion methods**, which are **the two baselines in Table 1 in our main paper**.
> > > We have already compared results from **[r3]** (denoted as RTS in our main paper) and **[r4]** (denoted as QCFS in our main paper) in our main paper.
> > >
> > > - **The paper (denoted as [r5] in the reviewer's comments) is an adaptive method** which is used to adaptively estimating gradient during SNN training to find the optimal shape and smoothness for gradient estimation.
> > >
> > > - **The paper (denoted as [r6] in the reviewer's comments) introduces the temporal efficient training (TET) approach** to compensate for the loss of momentum in the gradient descent with surrogate gradient (SG). The TET approach uses **Gradient Re-weighting** to help the training process converge into flatter minima with better generalizability.
> > >
> > >
> > > - **The papers (denoted as [r7-r8] in the reviewer's comments) investigate the sparsity of SNNs**, which propose new methods for finding better SNN architectures or SNN pruning.
> > >
> > > - To be specific, paper [r7] studied the problem of **deep SNN pruning**. It proposed Lottery Ticket Hypothesis (LTH) to scale up a pruning technique towards **deep SNNs**. As mentioned in the paper, the iterative searching process of LTH can bring a huge training computational cost when combined with the **multiple timesteps of SNNs**. Therefore, Early-Time (ET) ticket was proposed. The combination of the proposed ET ticket with common pruning techniques can be used for finding winning tickets, such as Iterative Magnitude Pruning (IMP) and Early-Bird (EB) tickets. Although they applied VGG-16 and ResNet neural networks to SNNs, but we are solving totally different problems using the same neural networks.
> > >
> > > - **We agree that SNN pruning (or finding better SNN architectures) is another efficient way to getting sparse SNNs with interesting connectivity.** Here, in our paper, we proposed another option to obtain an SNN using ANN-SNN conversion method. **These different types of methods are not conflicting but complementary in SNNs.**

---

> > > ### Author Response · Authors · 2022-11-23
> > > **Response to Reviewer FzYt (3/3)**
> > >
> > >
> > > **Q3: Has the SlipReLU methodology has been tried before (for e.g. [r4])?**
> > >
> > > A3: **NO**. **The SlipReLU method is a totally new method, which has never been tried before.**
> > >
> > > - The SlipReLU method is a piece-wise linear method. Using some linear algebra, the new SlipReLU can also be reformulated as a weighted sum of the Threshold-ReLU (method in [r3]) and the step function (method in [r4]).
> > >
> > >
> > > - **The ANN performance loss has never been studied in the existing ANN-SNN conversion studies**. We are **the first to consider the ANN performance loss** in ANN-SNN conversion methods, where the ANN performance loss refers to the performance degradation between the regular ANN and the tailored ANN.
> > >
> > > - By **considering the ANN performance as well as the conversion error simultaneously**, we proposed **the Unified Optimization Framework of ANN-SNN Conversion in Eq.(5)** in the main paper.
> > >
> > > - Based on **the Unified Optimization Framework of ANN-SNN Conversion in Eq.(5)** in the main paper, we then derive **the new SlipReLU activation function**.
> > >
> > > - The new SlipReLU activation function can be seen as a modification of the regular ReLU and it is a weighted sum of the threshold ReLU (method in [r3]) and the step function (method in [r4]) in **Eq.(9)** in the main paper.
> > >
> > >
> > > **Q4: Does the method in [r4] use a better ReLU alternative (for e.g. [r4]))? Is the new proposed SlipReLU a ReLU alternative?**
> > >
> > > A4: We **understand your confusion** about our proposed method with other ReLU alternatives such as the one in [r4]. We will try our best to make our proposed method as clear as possible.
> > >
> > > - As explained in **A3**, **we studied the ANN performance loss which is a new term that has never been studied in the existing ANN-SNN conversion studies.** By considering the ANN performance as well as the conversion error simultaneously, we proposed **the Unified Optimization Framework of ANN-SNN Conversion** which is in Eq.(5) in the main paper. Based on Eq.(5), we then proposed our new SlipReLU activation function.
> > >
> > > - Mathematically speaking, the new SlipReLU activation function can be seen as a modification of the regular ReLU and it is a weighted sum of the threshold ReLU (method in [r3]) and the step function (method in [r4]) in **Eq.(9)** in the main paper.
> > >
> > > - As shown in **Table 1** (in the main paper) and **Table S3, Table S4**, **our new proposed SlipReLU performs better than the method mentioned in [r4].**
> > >
> > >
> > > - **Indeed, our new SlipReLU method is a kind of ReLU alternatives**, and there are also many other ReLU alternatives that are used as activation functions in artificial neural networks, such as Gaussian Error Linear Unit (GELU) [i1], Exponential linear unit (ELU) [i2], Scaled exponential linear unit (SELU) [i3] , Leaky rectified linear unit (Leaky ReLU) [i4], Parametric rectified linear unit (PReLU) [i5] and so on. Our new SlipReLU method happens to be one kind of ReLU alternatives.
> > >
> > >
> > > [i1] Hendrycks, Dan; Gimpel, Kevin (2016). "Gaussian Error Linear Units (GELUs)". arXiv:1606.08415.
> > >
> > > [i2] Clevert, Djork-Arné; Unterthiner, Thomas; Hochreiter, Sepp (2015-11-23). "Fast and Accurate Deep Network Learning by Exponential Linear Units (ELUs)". arXiv:1511.07289.
> > >
> > > [i3] Klambauer, Günter; Unterthiner, Thomas; Mayr, Andreas; Hochreiter, Sepp (2017-06-08). "Self-Normalizing Neural Networks". Advances in Neural Information Processing Systems. 30 (2017). arXiv:1706.02515.
> > >
> > > [i4] Maas, Andrew L.; Hannun, Awni Y.; Ng, Andrew Y. (June 2013). "Rectifier nonlinearities improve neural network acoustic models". Proc. ICML. 30 (1).
> > >
> > > [i5] He, Kaiming; Zhang, Xiangyu; Ren, Shaoqing; Sun, Jian (2015-02-06). "Delving Deep into Rectifiers: Surpassing Human-Level Performance on ImageNet Classification". arXiv:1502.01852.

---

> ### Author Response · Authors · 2022-11-29
> **Would you please read our responses & revision and re-evaluate our work?**
>
>
> **Dear Reviewer FzYt,**
>
> Thank you for providing the insightful comments on our paper. We have tried our best to answer your questions piece by piece, to make clear the main idea of this paper and the novelty. It is very necessary to consider the performance degradation between the regular ANN and the tailored ANN in ANN-SNN conversion. As this performance loss will be inherited to the converted SNN. We are the first one to investigate this in a unified optimization framework of ANN-SNN Conversion, with effective and general solutions.
>
> We sincerely understand your concern on the effectiveness of our SlipReLU method compared to other directly-training SNN methods, and we have shown experiment results comparing directly-training SNN methods and other ANN-SNN models in the revision. We have updated the table results by including more directly-training SNN methods as you mentioned. One thing is certain that our SlipReLU method can always work well, which can be obtained from Theorem 1 & 2 in our paper. We are willing to do more experiments if needed, but the conclusion will remain the same that SlipReLU can always give good results for low latency inference, as our new method is designed to do this. We will revise the manuscript accordingly in the final version.
>
> We really appreciate it if you could re-evaluate our work which, to our best knowledge, is the first effort of considering the ANN performance loss in the process of ANN-SNN conversion.
>
> **Authors of Paper 1895**

---

### Author Response · Authors · 2022-11-17
**General Response to All Reviewers**

Dear reviewers, we thank for your great efforts and valuable comments on our paper. Below, we would like to response your questions generally.

First, we would like to emphasize our contributions again.

The existing ANN-SNN conversion methods usually redesign the ANN with a new activation function instead of the regular ReLU, train the tailored ANN and convert it to an SNN. The performance loss between the regular ANN with ReLU and the tailored ANN has never been considered, which will be inherited to the converted SNN.

- 1) **This is the first work to consider the performance loss between the regular ANN with ReLU and the tailored ANN which has never been considered in ANN-SNN conversion methods.** We formulate the ANN-SNN conversion as a unified optimization problem which considers the ANN performance as well as the conversion error simultaneously.

- 2) **We propose to use the SlipReLU activation function in the tailored ANN,** in order to minimize the layer-wise conversion error and keep tailored ANN performance as good as the regular ANN with regular ReLU activation function.

- 3) **The SlipReLU method covers a family of activation functions** mapping from activation values in source ANNs to firing rates in target SNNs; most of the state-of-the-art optimal ANN-SNN conversion methods are special cases of our proposed SlipReLU method.

- 4) **We give theoretical guarantees through two theorems that the expected conversion error between SNNs and ANNs can theoretically be zero on a range of shift values $\delta \in [-\frac{1}{2},\frac{1}{2}]$ rather than a fixed shift $\frac{1}{2}$.**
Experiment results also demonstrate, the proposed SlipReLU method can achieve high-accuracy and low-latency SNNs (e.g., 1 or 2 time-steps).

**Comment 1: Comparison with direct training SNN methods**

**Response 1:**
We appreciate the suggestion to add another experiment that compares ANN-SNN conversion methods with supervised training SNN methods. The table below shows the comparison with the state-of-the-art supervised training SNN methods. And we have added this table as **Table S2 in the Appendix F** in the revised paper.

Our approach for CIFARNet has achieved an accuracy of **95.31\%** with time-steps 4, the achieved accuracy is higher than any other supervised trained models for low latency value.

**Table S2: Compare with state-of-the-art supervised training methods on CIFAR-10 dataset.**

| Model  | Method     | Architecture | SNN Accuracy | Time-step T |
| -------|------------| -------------| -----------| ------------|
| HC | Hybrid | VGG-16 | 92.03 | 200 |
| STBP | Backprop | CIFARNet | 85.82 | 12 |
| GDDT | Backprop | CIFARNet | 87.35 | 4 |
| TSSL | Backprop | CIFARNet | 88.23 | 5 |
| Ours | ANN-SNN  | VGG-16 | 91.08 | 4 |
| Ours | ANN-SNN  | ResNet-18 | 92.86 | 2 |
| Ours | ANN-SNN  | CIFARNet  | 95.31 | 4 |


These state-of-the-art supervised training methods include Hybrid-Conversion (HC) from [1] Rathi et al. (2020), STBP from [2] Wu et al. (2018), TSSL from [3] Zhang & Li (2020) and GDDP from [4] Zheng et al. (2021), which are back-propagation or hybrid training methods.


[1] Nitin Rathi, et al. Enabling deep spiking neural networks with hybrid conversion and spike timing dependent backpropagation". arXiv preprint arXiv:2005.01807, 2020.

[2] Yujie Wu, et al. Spatio-temporal backpropagation for training high-performance spiking neural networks. Frontiers in neuroscience, 2018.

[3] Wenrui Zhang and Peng Li. Temporal spike sequence learning via backpropagation for deep spiking neural networks. NeurIPS, 2020.

[4] Hanle Zheng, et al. Going deeper with directly-trained larger spiking neural networks. AAAI, 2021.

**Comment 2: Experiment results on large-scale dataset**

**Response 2:**
We appreciate the suggestion to provide results on large-scale datasets to show the efficacy of our proposed method.
- We have also updated the paper with new results on **CIFAR100 and Tiny-ImageNet datasets** and can be found in the **Table S3 and Table S4 of appendix**.
- From Table S3, we see that our SlipReLU method outperforms the others both in terms of high accuracy and ultra-low latency.
- For ResNet-18, when T = 1, we can still achieve an accuracy of **71.51\%**.


**Comment 3: Experiment Settings**

**Response 3:**
- We have extended the experimental results section in the main paper, which can be found in **Table 1 and Figure 2 in Section 6**. We have updated the details of experimental setups in **Appendix E** to make the paper reproducible.
- For ultra-low latency inference (T = 1 or T = 2), our proposed SlipReLU has the best performance compared to existing state-of-the-art ANN-SNN conversion methods.
- Specially, when the latency T = 1, our SlipReLU method is able to achieve an
accuracy of **93.11\%** for ResNet-18. The accuracy for VGG-16 is **85.40\%** with SlipReLU activation.

**Comment 4: Source codes**

**Response 4:** The source codes can be found in the appendix.

---

### Comment · Area_Chair_GdAc · 2022-11-18
**Please respond to author rebuttals**

Dear Reviewers,

The authors have submitted their rebuttals. Please have a look and respond to their efforts. This will be a respect to their hard work. Many thanks!

Area Chair

---

### Comment · Area_Chair_GdAc · 2022-12-02
**About your experimental results**

Dear Authors,
Thank you very much for your great efforts on the rebuttals!
I notice that the accuracies of the same methods you reported in Table 2 of the latest version are much different from those in Table 1 of the original version. The largest differences are even 37.56% (55.55% increased to 93.11% for ResNet-18 (ours)) and 39.90% (48.40% increased to 88.30% for ResNet-18 (QCFS)). Moreover, for  ResNet-18 (QCFS) with T=2, the accuracy reported in (Bu et al. 2021) is 75.44%, not your 61.10%.
Moreover, the accuracy of TET reported in r6 (ICLR 2022) is $94.16\pm 0.03$%, not you wrote 92.98% in your rebuttal.
Can you explain why?

SAC

---

> ### Author Response · Authors · 2022-12-04
> **For experimental results**
>
>
> Dear SAC,
>
> Thank you very much for your time and great efforts on the rebuttals on our paper!
>
> The short explanation would be as follows:
>
> > ### First version: no hyper-parameter fine-tuning, N=16 (as in QCFS), c=0.4 (according to experience)
> > ### Latest version: hyper-parameter fine-tuning, as recommended by the reviewers
>
> Let us explain that piece by piece in details.
>
> > Q1. the accuracies of the same methods you reported in Table 1 of the latest version are much different from those in Table 2 of the original version. The largest differences are even 37.56% (55.55% increased to 93.11% for ResNet-18 (ours)) and 39.90% (48.40% increased to 88.30% for ResNet-18 (QCFS)).
>
> > A1. We have done some modification in the latest version according to the reviewers' comments. In the original version, we did not fine-tune the hyper-parameters, nor perform any ablation study. There are two hyper-parameters, viz, the slope c and the quasi-latency N, in the new proposed SlipReLU method. There is one hyper-parameter N in QCFS (Bu et al. 2021).
>
> > We used the default hyper-parameter N=16 for QCFS from their codes from https://github.com/putshua/SNN_conversion_QCFS, and we used the same N=16 for the proposed SlipReLU, then ran the experiments and reported in the Table 2 in the original (old) version. For the SlipReLU, we chose the hyper-parameter c=0.4 according to experience without hyper-parameter fine-tuning in the original (old) version.
> That is why we had accuracy of 55.55% for ResNet-18 (ours) and accuracy of 48.40% for ResNet-18 (QCFS) in the original (old) version.
>
> > In the rebuttal, we followed the reviewers' advice and conducted thorough studies on the effects of c and N, and we fine-tuned the hyper-parameters for both QCFS and SlipRsLU to obtain their optimal hyperparameters according to principled criterion other than experience/rule-of-thumb. We reported the final results in Table 1 in the latest version.
> That is why we have accuracy of 93.11% for ResNet-18 (ours) and accuracy of 88.30% for ResNet-18 (QCFS) in the latest version.
>
> > As the reviewers suggested, in order to make the paper reproducible, we have updated the details of experimental setups in Appendix E.
>
>
> > Q2. Moreover, for ResNet-18 (QCFS) with T=2, the accuracy reported in (Bu et al. 2021) is 75.44%, not your 61.10%.
>
> > A2. The above results were old results in the original (old) version, which were obtained with the default hyper-parameter from QCFS codes (Bu et al. 2021) with N=16. During the rebuttal, we have conducted thorough studies on the effects of (c, N), fine-tuned the hyper-parameters for both QCFS and SlipRsLU to obtain their optimal hyperparameters according to principled criterion other than experience/rule-of-thumb as the reviewers suggested, and reported the final results in Table 1 in the latest version.
>
> >  In the latest version, we used the QCFS code available on github https://github.com/putshua/SNN_conversion_QCFS to train the model ResNet18 for QCFS, and we followed the same settings in Appendix A.1 in (Bu et al. 2021) by training the model for 300 epochs with learning rate 0.1 with N=4, as mentioned in their paper. We obtained an accuracy of 91.52% on ResNet-18 (QCFS) with T=2, which is much better than the accuracy 75.44% reported in (Bu et al. 2021), and they are reported in Table 1 (latest version).
>
>
> > Q3. Moreover, the accuracy of TET reported in r6 (ICLR 2022) is $94.16 \pm 0.03$ %, not you wrote 92.98% in your rebuttal.
>
> > A3. For the TET method in r6 (ICLR 2022), we conducted experiments on TET method as the reviewer listed more papers. We used their codes available on github https://github.com/Gus-Lab/temporal_efficient_training to train the model ResNet19. They provided distributed pytorch codes using multiple GPUs at a time. We trained the model on 4 GPUs for 300 epochs with learning rate 0.01, for other parameters we followed the settings as defined in their paper, but we ended up with obtaining an accuracy of 92.98% on CIFAR10. We did not recover the exact same results as mentioned in their paper, the possible reason may be difference in the configuration of hardware, and there was a lot of discussion on the reproducibility issue https://openreview.net/forum?id=_XNtisL32jv&noteId=9tdBFCJIIkZ.

---

### Decision · Program_Chairs · 2023-01-20

**Decision:**

Reject

**Justification For Why Not Higher Score:**

The paper is not good enough. The meta-review clearly stated the reasons. But the AC wouldn't mind it be bumped up as three reviewers were positive towards the paper and only one gave a harsh score: strong reject.

**Justification For Why Not Lower Score:**

N/A

**Metareview: Summary, Strengths And Weaknesses:**

The paper proposes a new activation function SlipReLU for ANN-SNN conversion to jointly reduce the conversion error and ANN performance loss. Reviewers have quite diverse opinions on the paper. One reviewer recommends strong reject due to the lack of novelty and not good results, as well as missing comparisons with direct training methods of SNNs. Two reviewers recommend accept for good results and one reviewer recommend marginally above the acceptance threshold after the rebuttal. After reading the paper, the AC found that some concerns should be noticed.

First, there exists many inconsistencies between the results in the paper and those in previous works. For example, the reported results and trends for different network architectures of QCFS (ICLR 2022) are inconsistent with those in the original paper. The authors explained that they adjusted the hyperparameters for each method, according to the suggestions of reviewers, to get better results than the original ones, but for the VGG-16 and ResNet-20 networks, results of QCFS in this paper are much lower than those in the original paper, and for VGG-16, the original QCFS results are even better than those of the proposed method in Table 1 (when T is small). This was not properly explained. Besides, for direct training methods, some of the reported results in the rebuttal and revision are different from those in the original papers. For example, the authors reported 88.23% accuracy for TSSL-BP but in the original paper the result is 91.41%. And for STBP, GDDP, IIR-SNN, LTH, and NAS, the reported results are all inconsistent with the original paper. Since there are so many inconsistencies, some being better and some being worse, the authors' explanation on QCFS and TET did not fully resolve the concern of the AC.

Second, the large-scale ImageNet results are missing. For the conversion and training of SNNs, CIFAR results may not easily generalize to large-scale evaluation, especially with small time steps. These results are important and all previous ANN-SNN works reported results on ImageNet. The AC regreted that the authors could only provide results on Tiny-ImageNet due to computational constraints (but somehow the authors promised to provide the results on ImageNet if the paper is accepted).

Third, for theoretical analysis, a discussion on the assumptions is missing, e.g., the range of $v^{l}(T)$. This assumption is important to only consider quantization error in the analysis. However, in practice the assumption may not hold and there exists “unevenness error” (as stated in QCFS (ICLR 2022)), and recent works reveal that such kind of error is much larger than quantization error for deep networks (Meng et al., Training much deeper spiking neural networks with a small number of time-steps, Neural Networks, 2022). There should be proper discussions for the error analysis and important assumptions. This paper hardly goes beyond existing analysis results and does not theoretically show the advantage of SlipReLU. Moreover, the AC did not see much flavor of "unified optimization" as many of the steps were not deduced from optimization.

Given the above concerns and the overall scores, the AC deemed that the paper does not reach the acceptance threshold in its current form. So the AC recommended rejection, yet thanked the authors for taking great efforts in the rebuttal.


**Summary Of Ac-Reviewer Meeting:**

The AC called for the virtual meeting twice but only one reviewer responded to me throughout the process, despite multiple reminders from SAC and AC. So the AC had to go without virtual meeting.